

# Reconstructing climatic modes of variability from proxy records: sensitivity to the methodological approach

Simon Michel[1], Didier Swingedouw[1], Marie Chavent[2], Pablo Ortega[3], Juliette Mignot[4], and Myriam Khodri[4]

[1]Environnements et Paleoenvironnements Oceaniques et Continentaux (EPOC), UMR CNRS 5805 EPOC-OASU-Universite de Bordeaux, Allee Geoffroy Saint-Hilaire, Pessac 33615, France.
[2]Institut National de la Recherche en Informatique et Automatique (INRIA), CQFD, F-33400 Talence, France.
[3]BSC, Barcelona, Spain.
[4]Sorbonne Universites (UPMC, Univ. Paris 06)-CNRS-IRD-MNHN, LOCEAN Laboratory, 4 place Jussieu, F-75005 Paris, France.

**Correspondence:** Simon Michel (simon.michel@u-bordeaux.fr).

**Abstract.** Modes of climate variability strongly impact our climate and thus human society. Nevertheless, their statistical properties remain poorly known due to the short time frame of instrumental measurements. Reconstructing these modes further back in time using statistical learning methods applied to proxy records is a useful way to improve our understanding of their behaviours and meteorological impacts. For doing so, several statistical reconstruction methods exist, among which the Principal Component Regression is one of the most widely used. Additional predictive, and then reconstructive, statistical methods have been developed recently, following the advent of big data. Here, we provide to the climate community a multi-statistical toolbox, based on four statistical learning methods and cross validation algorithms, that enables systematic reconstruction of any climate mode of variability as long as there are proxy records that overlap in time with the observed variations of the considered mode. The efficiency of the methods can vary, depending on the statistical properties of the mode and the learning set, thereby allowing to assess sensitivity related to the reconstruction techniques. This toolbox is modular in the sense that it allows different inputs like the proxy database or the chosen variability mode. As an example, the toolbox is here applied to the reconstruction of the North Atlantic Oscillation by using Pages 2K database. In order to identify the most reliable reconstruction among those given by the different methods, we also investigate the sensitivity to the methodological setup to other properties such as the number and the nature of the proxy records used as predictors or the reconstruction period targeted. The best reconstruction of the NAO that we thus obtain shows significant correlation with former reconstructions, but exhibits better validation scores.

## 1 Introduction

The climate system is composed of interdependent subsystems, such as the atmosphere that can vary at relatively fast timescales as compared to the ocean or the cryosphere. As a result of the interactions between those components, the climate variability spectra is very large and ranges from hourly to multidecadal timescales [Mitchell et al. (1966)]. In the absence of any modulations of the external forcings, such variability is still present, as evidenced in preindustrial control simulations with





global coupled climate models. This variability is frequently referred to as internal variability [Hawkins and Sutton (2009)]. The variations and dynamics of the climatic system are also influenced by external factors such as volcanic aerosols [Mignot et al. (2011); Swingedouw et al. (2015); Khodri et al. (2017)], solar irradiance [Swingedouw et al. (2011); Seidenglanz et al. (2012)], anthropogenic aerosols [Evan et al. (2009); Evan et al. (2011); Booth et al. (2012)] and greenhouse gas concentrations

[Stocker et al. (2013)], which alter the Earth's radiation balance, and hence, deflect the mean climate state. By only considering internal variability and the impact of external forcings not due the human activity, one explores the so-called natural climate variability.

An unequivocal rise in both the greenhouse gas composition in the atmosphere and the global mean temperature has been observed in instrumental measurements [Bradley (2003); Stocker et al. (2013)]. However, the non-stationary variability around

this trend from a decade to another [Kosaka and Xie (2013); Santer et al. (2014); Swingedouw et al. (2017)] asks the question of the role of anthropogenic forcing relatively to that of natural variability for decadal to multidecadal climate variations. Thereby, improving our knowledge about natural climate variability should allow improving our knowledge and better evaluate the changes in climate in the near term future (decades, e.g. Hawkins and Sutton (2009)).

The physics driving the climate system induces large-scale variations, organised around recurring climate patterns with spe-

cific regional impacts and temporal properties. These variations are known as climate modes of variability, and their evolution is usually quantified by an index that can be calculated from a specific observed climate variable. These indices provide an evaluation of the corresponding climate variations and their regional impacts [Hurrell (1995); Neelin et al. (1998); Trenberth and Shea (2006)].

As an example, the North Atlantic Oscillation (NAO), is the leading mode of atmospheric variability in the North Atlantic

basin [Hurrell et al. (2003)]. Generally defined as the sea level pressure (SLP) gradient between the Azores high and the Icelandic low, the NAO describes large-scale changes in winter atmospheric circulation in the Northern hemisphere and controls the strength and direction of westerly winds and storm tracks across the Atlantic [Hurrell (1995)]. A stronger than normal SLP gradient between the two centers of action induces a northward shift of the eddy-driven jet-stream. Such large-scale changes in atmospheric circulation lead to precipitation and temperature variations in various regions (North Africa, Eurasia,

North America and Greenland [Casado et al. (2013)]). Moreover, these meteorological impacts have major influences on many ecological processes, including marine biology [Drinkwater et al. (2003)] as well as terrestrial ecosystems [Mysterud et al. (2001)]. This mode also affects the oceanic convection in the Labrador Sea and the Greenland-Iceland-Norwegian Seas through changes in atmospheric heat, freshwater and momentum fluxes [Dickson et al. (1996); Visbeck et al. (2003)]. These changes may lead to modifications in the Atlantic Meridional Overturning Circulation (AMOC) which in turn affects the poleward heat

transport and the related SST pattern over the Atlantic [Trenberth and Fasullo (2017)].

The dynamics of these modes are still not fully understood due to the small duration of the instrumental records, which is preventing robust statistical evaluation of their properties (spectrum, stability of teleconnections, underlying mechanisms ...). To partly overcome this limitation, numerous studies have reconstructed climate variations well beyond the period of direct





measurements of climate variables (since around 1870), and use proxy records to do so [Cook et al. (2002); Mann et al. (2009); Ortega et al. (2015); Luterbacher et al. (2016); Wang et al. (2017)]. Proxy records provide indirect observations of local or regional climate in the past, using natural archives coming for instance from sediment cores, speleothems, ice cores or tree rings. The different records have their own characteristics and limitations, which need to be considered when combined

together to perform the reconstructions. For example, each proxy record has a specific temporal resolution, from years to millennia, and then covers a specific period: from hundreds to millions of years. New proxy records are continuously gathered extending the available datasets and allowing paleoclimatologists to build increasingly consistent reconstructions [Pages 2K Consortium (2013); Pages 2K Consortium (2017)]. Hence, the last millennium is a period extensively investigated as it contains the densest network of high-resolution proxy records. [Mann et al. (2009); Luterbacher et al. (2016)].

The last millenium is of a great interest to put in perspective and understand the recent climate variations. Indeed, before the early $19^{th}$ century, the anthropogenic radiative forcing was negligible [Hegerl et al. (2007); Hawkins et al. (2017)], so that the climate variation was mainly natural. Moreover, proxy records reveal two contrasting climatic periods during that millennium, as identified by Lamb (1965). These periods are known as the Medieval Climate Anomaly (MCA) and the Little Ice Age (LIA) [Mann et al. (2009)], which correspond to an anomalously warm and cool period mean temperature in the Northern

hemisphere, respectively.

Modes of climate variability can have diverse worldwide impacts (usually known as climate fingerprints), which can be recorded by different proxy records. These records can be thus combined to make reconstructions of their variability. The selected proxy records need to cover, at least partially, the observational period. That is an important requisite to make a robust calibration. Based on this assumption, several studies have used statistical predictive methods to reconstruct different climatic

modes on longer timescales [Cook et al. (2002); Gray et al. (2004); Ortega et al. (2015); Wang et al. (2017)].

For instance, for the NAO, Cook et al. (2002) firstly proposed a complete methodology of nested Principal Component Regressions (PCRs) using annually resolved proxy records bounding the North Atlantic to reconstruct its variability back to 1400. Several new proxy records have been documented since this study [Pages 2K Consortium (2013)] and the NAO reconstruction could probably be largely improved if it was updated to include these new data. More recently, Ortega et al.

(2015) performed a NAO reconstruction from 1073 to 1969 based on the PCR, using 48 proxy records that were significantly correlated with the historical NAO index on their common time window. Instead of nesting reconstructions of different sizes, which leads to inhomogeneities between time windows using different proxy selections, Ortega et al. (2015) used several random calibration/validation samplings of the overlap period of the NAO index and the proxy records to perform individual reconstructions on the same time frame. By repeating numerous times that sampling, several reconstructions were obtained

through the different PCR results. This ensemble approach brings two advantages. The first is that since validation/calibration periods are not fixed, the validation/calibration skills do not depend on the particular way these periods are split. The second advantage is that the different reconstructions obtained can be aggregated by averaging each of them to isolate the coherent features among them. The standard deviation between the individual reconstructions is thereby reduced, as only the most





emergent patterns are kept. Such kind of ensemble reconstruction, using nested PCR as in Cook et al. (2002), have been recently made by Wang et al. (2017), but for reconstructing the Atlantic Multidecadal Variability (AMV), a climate variability index characterising large-scale variations in North Atlantic SST [Trenberth and Shea (2006)].

The recent increasing amount of data is not specific to the paleoclimatology field. Indeed, since the past four decades, the
advent of internet and technological innovation has allowed to store and manage exponentially growing data from various sources [Wang et al. (2009)]. Hence, the capacity of decision making through data analysis in several fields has been largely developed, using many predictive algorithms for all kind of data [Tibshirani (1996); Breiman (2001); Zou and Hastie (2005)]. That field of science, often referred as "big data", is based on several statistical and probability theories and is named Statistical Learning or Machine Learning which is a subpart of Artificial Intelligence [Vapnik (2000)]. Combined with cross validation
algorithms, the PCR is one of the most efficient statistical learning regression methods [Hotelling (1957)]. It is still considered as a performant method in many fields, such as paleoclimatology. However, more recent algorithms provide alternative methods that can also be used to reconstruct climate modes, and may possibly further improve the quality and the robustness of these reconstructions.

In this paper, we provide a toolbox, using multiple statistical approaches, for reconstructing climate modes indices. It is
based on four regression methods: the PCR, the Partial Least Squares regression (PLS), the Elastic-net regression (Enet) and the Random Forest (RF). The aim is to propose a systematic reconstruction approach through a computer device. This toolbox communicates with a large proxy database. This database contains various types of proxy records distributed all over the Earth, and associated with different climate variables. Therefore, this toolbox allows reconstructing any climatic mode in the past (Fig. 1). The confidence we have in the reconstruction is then evaluated through training-testing techniques. Some general
statistical learning tools, such as the cross validation, are first presented. The reconstruction methods, are then described in a mathematical formalism. We then compare these methods by reconstructing the NAO index over the last millenium. Finally, we investigate the reconstruction sensitivity to methodological choices, such as the method used, the reconstruction period targeted, the proxy predictors selection and the size of the training samples.

## 2    Data, notations and methodologies

### 2.1    Data

The assessment of our reconstruction techniques is investigated for the NAO index, as it is the mode of variability that has been observed for the longest time period. Indeed, this index is relatively simple to calculate from instrumental records because it only needs two instrumental record locations for SLP: one within the center of action of the Azores anticyclone and one within the Icelandic low. Thus, because of this simplicity, the NAO index covers a longer instrumentally observed period than any
other indices. The reference NAO index calculated from SLP records in Gibraltar and Reykjavik starts in 1856 [Jones et al. (1997)]. An extension to 1823 has been proposed, using new SLP series from Cadiz and San Fernando, approximately 100 kilometers from Gibraltar [Vinther et al. (2003)]. This extended index is chosen as our historical NAO index in this paper.





Our statistical toolbox is based on a set of proxy predictors essentially composed of the Pages 2k 2014 version database [Pages 2K Consortium (2013)]. However, some proxy records (`Arc_38` to `Arc_59`, following PAGES encoding) have been removed because their resolution is longer than ten years, which may have an impact on the interpretation of annual to sub-decadal climate processes in the reconstruction. All the proxy records with a greater than annual resolution are then linearly

interpolated to that resolution. We also added to this database 69 proxy records used in the Wang et al. (2017) and Ortega et al. (2015) studies. All of the North American tree ring series in Pages 2K database have been truncated to 1200 as this is their oldest common year. 15 of these series extend further back in time and have been considered here in their full length. These series are encoded as `NAm-TR_7, 13, 14, 15, 21, 28, 29, 30, 62, 76, 81, 109, 110, 127, 128` in the Pages 2K database 2014 version [Pages 2K Consortium (2013)]. We end up with 539 worldwide distributed proxy records,

which can potentially allow to reconstruct any mode of variability. All of the proxy records which are not in the Pages 2K 2014 version are presented in Supplementary table 1. For the other proxy records, the reader can refer to the Pages 2K 2014 version database. We attribute an ID to each proxy records to make them recognizable by the users of the statistical tool (see Supplementary table 1). Among the 539 proxy records, only those completely overlapping the reconstruction period are kept. The statistical tool that we propose adjust the proxy dataset depending on the reconstruction period targeted.

## 2.2 Methodology

The reconstruction procedure follows 10 steps, all already implemented in the statistical toolbox. These are applied sequentially as follows (Fig. 1):

1. An observational time series of the mode of variability is chosen to be used as the predictand

2. A target time period $\mathcal{T}$ for the reconstruction is selected

3. The statistical reconstruction method to be applied is selected

4. The proxy records that overlap with the selected reconstruction period are extracted to be used as predictors

5. The common period between the observed climate index and the selected proxy records is extracted for fitting the reconstruction

6. This common period is randomly split into two parts, one for training the model (training period), and one for testing it
(testing period). This is repeated $R$ times to generate an ensemble

7. For each member of the ensemble, the reconstruction is calibrated over the training period for all the different statistical parameters for a given method, and the best one is identified

8. The corresponding optimal setup is then applied to extend the reconstruction over the target period $\mathcal{T}$ for each ensemble member





9. A validation score is computed for each member by comparing the true testing series and each individual reconstruction over the corresponding testing period

10. The final reconstruction is calculated as the average of all the individual $R$ reconstructions

Thus the toolbox provides the mean reconstruction and a vector with $R$ validation scores as final outputs.

The number of proxy records and the reconstruction period are here fixed for the different training/testing period sections, in contrast with some previous studies which used nested approaches [Cook et al. (2002); Wang et al. (2017)]. Indeed, we argue that as the weight of each proxy record is unknown before performing the reconstruction, the nested approaches may attribute unrealistic weights to the proxy records that bear the longest temporal coverage. In addition, as we want to perform several reconstructions by changing the set of proxy records employed or the reconstruction period considered, using a nested
approach would have a simultaneous impact on both factors, and may hinder the interpretation of the validation scores.

### 2.3    Mathematical formalism of empirical data

To facilitate the mathematical notation, we make the assumption that the proxy record selection and truncation have already been made (see section 2.2, steps 4 and 5). It is important that all proxy records are truncated on the same time window to make them mergeable in the same matrix. Each record has to cover at least the chosen reconstruction time window $\mathcal{T}$ (section
2.2, step 2). Following these steps, the proxy record matrix does not contain missing values.

Fig. 2 illustrates how the proxy records data are organised in the input matrix $X$. We denote $X^1 = (X_t^1)_{t \in \mathcal{T}}, \ldots, X^p = (X_t^p)_{t \in \mathcal{T}}$, where $t$ stands for the time (with $N$ annual time steps), and $p$ is the number of proxy records on the same period $\mathcal{T}$. $X$ is thus a $N \times p$ matrix where all these vectors are merged: $X = [X^1, \ldots, X^p]$. $Y = (Y_t)_{t \in T}$ is the target mode of variability, defined on the historical time window $T$, containing $n$ annual time steps. The period where $Y$ is not known is denoted $\tau$,
containing $m$ annual time steps (Fig. 2). Thus $\mathcal{T} = T \cup \tau$ is the entire reconstruction period, which contains $N$ annual time steps. With these notations, the dimensions of the different matrices and vectors are: $X \in \mathbb{R}^{N \times p}$; $X_{(T)} \in \mathbb{R}^{n \times p}$; $X_{(\tau)} \in \mathbb{R}^{m \times p}$; $Y \in \mathbb{R}^n$. The period $T$, on which all the predictors and the predictand are known and the training/testing splits are performed, is called the learning period. The period $\mathcal{T} = T \cup \tau$, covered by the predictors, is called the reconstruction period. The learning set is then $\{X_{(T)}, Y\}$, and the reconstruction set is $\{X_{(\mathcal{T})}\}$.

### 2.4    Terms and notations of learning theory

To build and validate the reconstruction of $Y$, the dataset of predictors $X$ is split in two independent subsets, one for training the statistical model (usually called training set), and another on which the statistical model is tested (called testing dataset or first seen data).

Building a model consists in estimating all the parameters needed to reconstruct $Y$ given the predictors $X^1, \ldots, X^p$. As an
example, building a PCR model consists in determining the Principal Component of the predictor matrix $X$ and finding the





best linear combination of them to reconstruct $Y$ over the training period. Then, the reconstruction consists in projecting the first seen data on the orthogonal basis built, and applying the estimated regression coefficients to reconstruct $Y$ over the whole time window $\mathcal{T}$.

We denote the chosen reconstruction method by $\mathcal{M}$. Each method is defined by a specific number of parameters $q$, contained in the vector $\theta$. As an example, the Principal Component Regression has a single parameter that is the number of Principal Component used as regressor (Cook et al. (2002); Gray et al. (2004); Ortega et al. (2015); Wang et al. (2017)). We can denote the function $\mathcal{M}$ as a function of: $(i)$ a set on which the model is built ($\{X, Y\}$), $(ii)$ observations of the predictors on the reconstruction period ($X_{(rec)}$), and $(iii)$ a parameter vector ($\theta$):

$$\mathcal{M} : (\{X, Y\}, X_{(rec)}, \theta) \rightarrow \hat{Y}_\theta \tag{1}$$

$$(\{\mathbb{R}^{n \times p}, \mathbb{R}^n\}, \mathbb{R}^{m \times p}, \mathbb{R}^q) \rightarrow \mathbb{R}^m \qquad\qquad n, p, m, q \in \mathbb{N} \text{ (not fixed)} \tag{2}$$

Hence, the $\mathcal{M}$ function gives an entire reconstruction of size $m \in \mathbb{N}$, depending on $\theta$ for given training/testing periods.

We introduce $S$ as the score function. This function is an indicator that estimates the quality of a prediction $\hat{Y}$ in comparison to the observed values $Y_{(obs)}$:

$$S : (Y_{(obs)}, \hat{Y}) \rightarrow s \tag{3}$$

$$(\mathbb{R}^m, \mathbb{R}^m) \rightarrow \mathbb{R} \tag{4}$$

In this paper, two kind of score functions will be considered. The first is a correlation function, and the second is a root mean squared error (RMSE) function:

$$S_{cor}(Y_{(obs)}, \hat{Y}) = Cor(Y_{(obs)}, \hat{Y}) \tag{5}$$

$$S_{RMSE}(Y_{(obs)}, \hat{Y}) = \|Y_{(obs)} - \hat{Y}\| = \sqrt{\sum_{i=1}^{m}(Y_{i\,(obs)} - \hat{Y}_i)^2} \tag{6}$$

The first will be used to validate the reconstruction methods over the testing period, and the second will allow to determine the optimal parameters ($\theta$) for the reconstruction over the training period.

### 2.5 Parameter tuning and model comparisons

#### 2.5.1 Parameter tuning by leave-one-out cross validation

To estimate the optimal set of parameters $\theta_{opt}$ on a given training set $\{X_{train}, Y_{train}\}$, we use the leave-one-out cross validation (LOOCV; section 2.2, step 7 and 8) [Stone (1974); Geisser (1975)]. Cross Validation (CV) methods, are in general, widely used as parametrization and model validation techniques [Kohavi (1995); Browne (2000); Homrighausen and McDonald (2014); Zhang and Yang (2015)]. As presented in Fig. 3, the particularity of the LOOCV is that it use a single observation





for verification and the $n-1$ other observations as calibration set [Stone (1974)]. Here it is used to determine an empirically optimal set of parameters for $\theta$. $\forall 1 \leq i \leq n$, we denote $\{X_{(i)}, Y_{(i)}\}$, containing only information for the $i^{th}$ time step. Then, $\{X_{(-i)}, Y_{(-i)}\}$ is the set containing all the initial observations, except the $i^{th}$. For all possible values of $\theta$ contained in $\Theta$, we scan the $n$ models based on the sets $\{X_{(-i)}, Y_{(-i)}\}_{1 \leq i \leq n}$. The empirical optimal set of parameters is obtained by minimizing

the averaged $S_{RMSE}$ functions on the $n$ splits regarding all possible combinations of $\theta$ [Stone (1974)]. Mathematically, the optimal LOOCV set of parameters $\theta_{LOO}$ is determined by:

$$\theta_{LOO} = \arg\min_{\theta \in \Theta} \frac{1}{n} \sum_{i=1}^{n} S_{RMSE}(Y_{(i)}, \mathcal{M}(\{X_{(-i)}, Y_{(-i)}\}, X_{(i)}, \theta)) \tag{7}$$

Using this approach, we retain the empirical estimation of the optimal set of parameters $\hat{\theta}_{opt} = \theta_{LOO}$ for the given method $\mathcal{M}$ and a given learning set $\{X, Y\}$.

### 2.5.2   Final reconstructions and validation correlations

In order to find the most performant method for a given dataset, we split the initial learning period $T$ in $R$ partitions of two subsets: $\{T_{(train)}^{(r)}, T_{(test)}^{(r)}\}, \forall 1 \leq r \leq R$. For all the methods, $R$ reconstructions are build on the $R$ training periods. $R$ is arbitrarily chosen, but larger $R$ tends to produce reliable ensemble reconstruction by decreasing the variance of the $R$ individual reconstructions made on the training samples [Browne (2000)]. $\forall 1 \leq r \leq R$, we denote $\{X_{(train)}^{(r)}, Y_{(train)}^{(r)}\}$ the training set,

and $\{X_{(test)}^{(r)}, Y_{(test)}^{(r)}\}$ the test set.

LOOCV is applied to build a unique optimized reconstruction for every training sets and any given method. Then, for all the corresponding and independent testing periods, the associated testing series $Y_{(test)}^{(r)}$ are compared to the individual reconstructions using the $S_{cor}$ function. This way, $R$ validation correlations are obtained for the four methods. In section 4, the distributions of the validation correlations will be used as a metric to compare different reconstructions. Fig. 4 shows the

whole procedure to get the validation correlation vectors for a given method $\mathcal{M}$.

## 3   Statistical learning methods

We present each method in two steps: model fitting (training) and reconstruction (testing). We also identify the number of parameters and their mathematical meaning. For each method the proxy predictor set is denoted as $X \in \mathbb{R}^{n \times p}$ the proxy predictor set and the target index as $Y \in \mathbb{R}^n$. In this section, $X_{(rec)} \in \mathbb{R}^{m \times p}$ is the testing dataset on which a $\mathbb{R}^m$ reconstruction

vector is evaluated on the testing period. $Y$ and each column of $X$ are here normalized on their own time period.





### 3.1 Principal Component Regression (PCR)

#### 3.1.1 Modeling

The Principal Component Regression [Hotelling (1957)] method consists in finding the best linear combination between $Y$ and the Principal Component of $X$. The Principal Component Analysis (PCA) consists in applying an orthogonal transformation of

5 an initial set of variables, potentially correlated between them, into another set of linearly uncorrelated variables: the Principal Component [Pearson (1901); Hotelling (1933)].

The first step consists in building an orthogonal basis where $X$ will be projected. We define $S \in \mathbb{R}^{p \times p}$, as the empirical estimator of the covariance matrix of $X$:

$$S = \frac{1}{n} X^T X \in \mathbb{R}^{p \times p} \tag{8}$$

The idea is to calculate the orthogonal basis formed by the vectors $v_1, \ldots, v_p$ by diagonalizing $S$:

$$v_1 = \arg \max_{\substack{v \in \mathbb{R}^p \\ \|v\|=1}} v^T S v \tag{9}$$

$$v_2 = \arg \max_{\substack{v \in \mathbb{R}^p \\ \|v\|=1 \\ \langle v^T v_1 \rangle = 0}} v^T S v \tag{10}$$

$$\ldots \tag{11}$$

$$v_p = \arg \max_{\substack{v \in \mathbb{R}^p \\ \|v\|=1 \\ \langle v^T v_1 \rangle = 0 \\ \ldots \\ \langle v^T v_{p-1} \rangle = 0}} v^T S v \tag{12}$$

$$\tag{13}$$

where $\|v\| = \sqrt{\sum_{j=1}^p (v^j)^2}, \forall v \in \mathbb{R}^p$. This procedure is equivalent to maximizing step by step the empirical variance of the projection of $X$ on each orthogonal axis. Indeed, $\forall v \in \mathbb{R}^p$ :

$$v^T S v = \frac{1}{n-1} v^T X^T X v = \frac{1}{n-1} (Xv)^T (Xv) = Var_{emp}(Xv) \tag{14}$$

The vectors $(v_k)_{1 \le k \le p}$ are called the Empirical Orthogonal Functions (EOFs). Since the columns of $X$ represent the proxy

records, it means that each EOF, which corresponds to the eigenvectors of the covariance matrix, contains a certain part of the spatial variability of the dataset. Hence, we attribute them eigenvalues $(\lambda_k)_{1 \le k \le p}$, which corresponds to the initial variance of $X$ translated by each orthogonal projection in the new basis:

$$\lambda_k = Var(Xv_k) = v_k^T S v_k \qquad\qquad \forall 1 \le k \le p \tag{15}$$





The Principal Component $(u_1, \ldots, u_p)$ are then the projections of $X$ on the EOFs. We denote $V = (v_1, \ldots, v_p)$. We then calculate the Principal Component matrix $U = (u_1, \ldots, u_p)$, defined as:

$$U = XV \in \mathbb{R}^{n \times p} \tag{16}$$

Now, we regress $Y$ on the $q \leq p$ (see subsection 3.1.3) first Principal Component. These $q$ Principal Component are merged in a submatrix of $U$: $\mathcal{U} = (u_k)_{1 \leq k \leq q}$. The model is given by:

$$Y = \mathcal{U}\beta + \epsilon \tag{17}$$

Where $\epsilon$ is a white noise vector of size n.

The best estimator for $\beta = (\beta_1, \ldots, \beta_q)$, is given by the Ordinary Least Squares (OLS) estimator which minimizes $\|\hat{\epsilon}\| = \|Y - \hat{Y}\|$:

$$\hat{\beta}_{OLS} = (\mathcal{U}^T \mathcal{U})^{-1} \mathcal{U}^T Y \tag{18}$$

### 3.1.2 Reconstruction

Using the testing data matrix $X_{(rec)}$ (see section 2.4), we project the former on the pre-calculated orthogonal basis $V$:

$$U_{(rec)} = X_{(rec)} V \in \mathbb{R}^{m \times p} \tag{19}$$

We then obtain the prediction by applying the estimated coefficient vector on the sub-matrix $\mathcal{U}_{(rec)} = (U^1_{(rec)}, \ldots, U^q_{(rec)}) \in \mathbb{R}^{m \times q}$:

$$\hat{Y}_q = \mathcal{U}_{(rec)} \hat{\beta}_{OLS} \in \mathbb{R}^m \tag{20}$$

### 3.1.3 Parameters

Here, $q$ is the unique tuning parameter. The choice of that parameter clearly affects the reconstruction and then the validation correlations. Here the parameter vector $\theta$ is unidimensional and takes its values in the discrete set $\{i\}_{1 \leq i \leq p}$.

To our knowledge, this is the first time that a PCR uses the LOOCV method to tune the number of Principal Component used at each split in paleoclimatological reconstruction. Previous studies used different criteria to define the number $q$ of Principal Component $U_1, \ldots, U_q$ to be kept. For example, Gray et al. (2004) retained all Principal Component for which the cumulated eigenvalues weights just exceeds $66\%$ of the initial variance. Wang et al. (2017), selected the $q$ Principal Component for which $\lambda_k > 1, \forall k \in \{1, \ldots, p\}$. Also, Ortega et al. (2015) used the Preisendorfer's rule N [Preisendorfer (1988)]. In our case, the use of LOOCV as our parameter selection method is preferred, as it is also valid for the other reconstruction techniques.



### 3.2 The Partial Least Squares Regression

The Principal Component Analysis keeps most of the initial variance in $X$ in a lower number of vectors. The major problem of the PCR in a predictive or reconstructive purpose, is that the EOFs $v_1, \ldots, v_p$ are constructed without taking into account any information about the predictand $Y$. Another possible approach is thus to determine the orthogonal basis in which the empirical

covariance between $Y$ and the projection of $X$ on that former is maximized. This is the Partial Lest Squares regression (PLSr) method [Zou and Hastie (2005)].

The first latent variable (LV), denoted $\xi_1 = \sum_{j=1}^{p} v_{1,j} X^j = Xv_1$, where $X \in \mathbb{R}^{n \times p}$ and $v_1 \in \mathbb{R}^p$ is the linear combination of the initial variables $X^1, \ldots, X^p$ such as:

$$v_1 = \arg \max_{\substack{u \in \mathbb{R}^p \\ \|v\|=1}} Cov(Y, Xv), \tag{21}$$

In a similar approach to the PCR, the second LV is $\xi_2 = \sum_{j=1}^{p} v_{2,j} X^j = Xv_2$, orthogonal to $\xi_1$, such as:

$$v_2 = \arg \max_{\substack{v \in \mathbb{R}^p \\ \|v\|=1 \\ \langle \xi^1, Xv \rangle = 0}} Cov(Y, Xv) \tag{22}$$

And so on, until we have $r \leq p$ LVs. The LV matrix is denoted $\Xi = [\xi_1, \ldots, \xi_p]$. Here, $v_1, \ldots, v_p \in \mathbb{R}^p$, are analogous to the EOFs in PCA, and are called loadings. The latent variables $\xi_1, \ldots, \xi_r$ respectively correspond to the projection of $X$ on the $r$ loadings.

Finding the loadings is not as trivial as for PCR. This is due to the fact that the empirical covariance matrix is not necessary definite positive and thus cannot be inversed. We solve this problem by using the algorithm 1 named PLS1. Analogously to the PCR, the method provides various alternative reconstructions depending on the value $r$, which corresponds to the number of LVs kept as regressors.

---

**Algorithm 1** : PLS1

---

$X_0 \leftarrow X$

*for* $h = 1, \ldots, r$

$v_h \leftarrow \dfrac{X_{h-1}^T Y}{\|X_{h-1}^T Y\|^2}$

$\xi_h \leftarrow X_{h-1} v_h$

$X_h = X_{h-1} - \dfrac{\xi_h \xi_h^T}{\|\xi_h\|^2} X_{h-1}$ *(deflation phase)*

---

Now we regress $Y$ on the $r \leq p$ first LVs. These $r$ LVs are merged in a submatrix of $\Xi$: $\Psi = (\xi_k)_{1 \leq k \leq r}$. The model is given

by:

$$Y = \Psi \beta + \epsilon \tag{23}$$

Where $\epsilon$ is a white noise vector of size n.





The best estimator for $\beta = (\beta_1, \ldots, \beta_q)$, is given by the Ordinary Least Squares (OLS) estimator which minimizes $\|\hat{\epsilon}\| = \|Y - \hat{Y}_{q_{LOO}}\|$:

$$\hat{\beta}_{OLS} = (\Psi^T \Psi)^{-1} \Psi^T Y \tag{24}$$

### 3.2.1 Reconstruction

The prediction is done in the same way as for PCR. Using the first seen data matrix $X_{(rec)}$ (section 2.4), we project the latter on the pre-calculated orthogonal basis $V$:

$$\Xi_{(val)} = X_{(val)} V \in \mathbb{R}^{m \times p} \tag{25}$$

We then obtain the prediction by applying the estimated coefficient vector on the sub-matrix $\Psi_{(rec)} = (\xi^1_{(rec)}, \ldots, \xi^r_{(rec)}) \in \mathbb{R}^{m \times r}$:

$$\hat{Y}_r = \Psi_{(rec)} \hat{\beta}_{OLS} \in \mathbb{R}^m \tag{26}$$

### 3.2.2 Parameters

For the PLSr method, $r$ is the unique tuning parameter. Analogously to the Principal Component Analysis, the tuning of that latter is obtained by LOOCV.

## 3.3 The Elastic Net regression

### 3.3.1 Modeling

Without using orthogonal transformation of the initial variables as in PCR and PLSr, the most simple predictive model is the multiple linear regression model:

$$Y = X^1 \beta_1 + \cdots + X^p \beta_p + \epsilon \tag{27}$$

Where $\epsilon \sim \mathcal{N}(0, \sigma^2)$ and $Cov(\epsilon_i, \epsilon_j) = 0$ if $i \neq j$.

The prediction of $Y$, given $p$ proxy records $X^1, \ldots, X^p$ is obtained by the equation:

$$\hat{Y} = X^1 \hat{\beta}_1 + \cdots + X^p \hat{\beta}_p \tag{28}$$

$\hat{\beta} = (\hat{\beta}_1, \ldots, \hat{\beta}_p)$ are the regression coefficients, which are obtained by the OLS predictor. However, this usual regression model is known to present frequently a poor prediction accuracy due to the several assumptions made on the original data [Poole and O'Farrell (1971)], which are often not verified: such as homoscedasticity and errors normality. Several studies developed 25 regularized (or penalized) regression methods to overcome the OLS defaults. Here we focus on the Elastic Net regression [Zou and Hastie (2005)], which is a combination of the Ridge regression [Hoerl and Kennard (1970)] and the Lasso regression





[Tibshirani (1996)]. All these methods have been developed to avoid the high variability of the OLS predictor when the number of predictors is relatively high. The Ridge regression shrinks towards zero the estimated coefficients associated to predictors unlinked to the predictand. No predictor selection is made by this method, but the shrunken estimated coefficients modulate the importance of these in the model. By contrast, the lasso also reduces the variability of the estimates, but in this case by

5    shrinking to zero the estimated coefficients associated to unreliable variables. Hence, a selection is made by rejecting variables associated to coefficients shrunk to zero.

The idea of a regularized (or penalized) regression is to add a threshold constraint using the $l_k$ norm of $\beta$: $\|\beta\|_k^k = \sum_{j=1}^{k} |beta_j|^k$. With $k = 1$ in Lasso regression, and $k = 2$ in Ridge regression. The penalized loss functions are given by:

$$L^{ridge}(\beta) = \|Y - \sum_{j=1}^{p} \beta_j X^j\|^2 + \lambda_2 \sum_{j=1}^{p} \beta_j^2 \tag{29}$$

$$L^{lasso}(\beta) = \|Y - \sum_{j=1}^{p} \beta_j X^j\|^2 + \lambda_1 \sum_{j=1}^{p} |\beta_j| \tag{30}$$

$$L^{enet}(\beta) = \|Y - \sum_{j=1}^{p} \beta_j X^j\|^2 + \lambda_1 \sum_{j=1}^{p} |\beta_j| + \lambda_2 \sum_{j=1}^{p} \beta_j^2 \tag{31}$$

$\lambda_1$ penalizes the sum of the absolute values of the regression coefficients while $\lambda_2$ penalizes their summed squares. Here, $\lambda_1, \lambda_2 > 0$.

Let $w = (w_j)_{1 \leq j \leq p} = (sgn(\beta_j))_{1 \leq j \leq p}$, where $sgn$ is the sign function. The loss functions can then be denoted as:

$$L^{ridge} = \|Y - X\beta\|^2 + \lambda_2 \beta^T \beta \tag{32}$$

$$L^{lasso} = \|Y - X\beta\|^2 + \lambda_1 w^T \beta \tag{33}$$

$$L^{enet} = \|Y - X\beta\|^2 + \lambda_1 w^T \beta + \lambda_2 \beta^T \beta \tag{34}$$

The estimated regression coefficients obtained by minimizing the Lasso and the Ridge loss functions are:

$$\hat{\beta}^{lasso} = (X^T X)^{-1}(X^T Y - \frac{\lambda_1}{2} w) \tag{35}$$

$$\hat{\beta}^{ridge} = (X^T X + \lambda_2 I)^{-1} X^T Y \tag{36}$$

The Elastic Net regression coefficients are then estimated by minimizing $L^{enet}$:

$$\hat{\beta}^{enet} = (X^T X + \lambda_2 I)^{-1}(X^T Y - \frac{\lambda_1}{2} w) \tag{37}$$

An alternative way to write this equation as a linear combination of $\hat{\beta}^{lasso}$ and $\hat{\beta}^{ridge}$ is:

$$\hat{\beta}^{enet} = (X^T X + (1 - \alpha)\lambda I)^{-1}(X^T Y - \frac{\alpha\lambda}{2} w) \tag{38}$$

25    where $\alpha \in [0, 1]$. If $\alpha = 1$, a Rigde regression is applied, and if $\alpha = 0$, we apply a Lasso regression.





### 3.3.2 Reconstruction

The prediction is obtained by applying the estimated regression coefficients $\hat{\beta}^{enet}$ on the validation variables $X_{val}^1, \ldots, X_{val}^p$:

$$\hat{Y}_{\lambda,\alpha} = \sum_{j=1}^{p} X_{(val)}^j \hat{\beta}_j^{enet} \qquad (39)$$

### 3.3.3 Parameters

For Enet method, the tuning parameters are $\lambda$ and $\alpha$. The latter controls the relative balance between the lasso and ridge regularization, while the former controls the overall intensity of regularization as $\lambda_1$ (resp. $\lambda_2$) in lasso (resp. ridge regularization). A high $\alpha$ suggests a dense model with many but small non-zero coefficients. A low $\alpha$ suggests a sparse model with many zero coefficients. In our case, since we want a general methodology performant for each random split, we apply two simultaneous LOOCV to find the best estimated pair $(\hat{\lambda}, \hat{\alpha})$.

Since $\lambda$ and $\alpha$ take respectively their values in the continuous sets $\mathbb{R}^p$ and $[0,1]$, we have to discretize their respective intervals for the parameter estimation. The finer these discretizations are, the more reliable the parameters will be, but the longer the required computational time will be.

## 3.4 Random Forest regression

The random forest has been introduced by Breiman (2001) as a learning method for regression. The method relies on using
randomization to minimize the prediction uncertainty given by regression trees. Random forests encompass a large variety of regression methods [Breiman (2001)]. Here, we present the most classical kind of random forests known as random-input random forests [Breiman (2001)].

### 3.4.1 Modeling

First we have to define regression trees. We denote each set of predictand/predictors by $\{Y_i, X_i)_{1 \le i \le n}\}$ where $X_i = (X_i^1, \ldots, X_i^p)$,
is the ensemble of proxy records for the $i^{th}$ time step, and $Y_i$ the corresponding values of the climate index at the same time step, $\forall 1 \le i \le p$. All the observations, $\{Y_i, X_i)_{1 \le i \le n}\}$, $\forall 1 \le i \le p$, are put on the root of the tree. The first step consists in cutting that root in two child nodes. A cut is defined as:

$$\left\{ X^j \le d \right\} \cup \left\{ X^j \ge d \right\} \qquad (40)$$

where $j = \{1, \ldots, p\}$ and $d \in \mathbb{R}$. Cutting a node with $\left\{ X^j \le d \right\} \cup \left\{ X^j \ge d \right\}$ means that all observations with a $j^{th}$ variable
lower than $d$ are placed in the left child node. Hence, all observations with a $j^{th}$ variable greater than $d$ are placed in the right child node. The method selects the best pair $(j, d)$ which minimize a loss function. Here, we aim at minimizing the variance of the child nodes. The variance of a given node $t$ is defined as:

$$\sum_{i: X_i \in t} (Y_i - \bar{Y}_t)^2 \qquad (41)$$

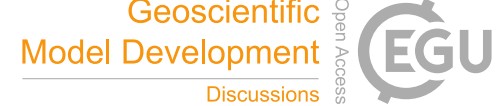



where $\bar{Y}_t$ is the averaged $Y_i$ in the node $t$.

The same procedure is then applied recursively to the child nodes using the same variables until a stop criterion is reached. The procedure automatically stops if each node contains a unique observation. Hence, the maximal depth of a regression tree is $n-1$. An illustration of such tree is presented in Fig. 5.

A random-input regression tree is used here. This is a particular case of regression trees, in which a set of $m < p$ variables is randomly preselected before applying the regression tree. A large number $K$ of random-input trees is computed. For each tree, we randomly select $m < p$ variables with probability $\frac{1}{p}$ and we apply the method until it reaches its maximal depth.

### 3.4.2 Reconstruction

The prediction is obtained by splitting each testing series in the different trees previously constructed. In each tree, the estima-
tion attributed to an observation is the empirical average of $Y$ inside the node where the corresponding observation ends up, given the cut made on the corresponding predictors. For each testing series, the $K$ reconstructions are averaged to give the final prediction.

### 3.4.3 Parameters

A priori, this method requires the optimization of two parameters: the number of trees $K$ and the number of variables selected
for each tree $m$. In practice $K$ does not require to be tuned, as long as the number of trees is sufficiently high given p, which guarantees convergent results for any value of $m$ [Breiman (2001)]. $m$ is then the only parameter to optimize. The LOOCV is then applied on $m$ with a high $K$ (here set to 1000), to select empirically the most efficient model.

## 4   Results

### 4.1  Methodological sources of uncertainty in the reconstruction

We apply the former methods to the reconstruction of the NAO. In the following, each reconstruction is obtained by averaging $R = 50$ individual reconstructions performed for $R$ training/testing random draws. Validation scores (based on correlations over the testing periods) are also produced, and stored in a vector of R elements. This vector will thus be used as a quality metric to characterize the methodological uncertainty in the reconstruction. The following actions were undertaken to minimize the reconstruction uncertainty, and estimate its sensitivity:

1. Pre-selecting the most relevant proxy records

    2. Choosing the most appropriate training/testing window length

    3. Selecting the best learning period

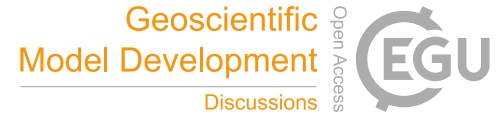

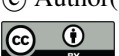

### 4.1.1 Proxy pre-selection

In order to investigate the sensitivity related to the selection of the initial set of predictors, we set the reconstruction period to $\mathcal{T} = \{1000, 1970\}$, and the learning period to $T = \{1823, 1970\}$, with $n = 148$. In addition, the training window length is set at $n_{train} = 111$, which gives $n_{test} = 37$. Only 122 of the 539 proxy records of the initial dataset are covering this reconstruction

period.

We run 4 different reconstructions, for each method, each based on a different proxy group chosen according to a correlation significance test with the original NAO index on the period $T$. The first group contains all the available proxy records on the period $\mathcal{T}$ (122 proxy records). The three other groups respectively contain the proxy records significantly correlated with the NAO index at the confidence levels $80\%$ (61 proxy records), $90\%$ (35 proxy records) and $95\%$ (18 proxy records). The

proxy records, and their respective correlation significance level with the NAO index are presented in Fig. 6. Fig. 7 gives the validation scores related to each reconstruction and each proxy selection.

First, it appears that for each method, the validation scores are improved when we use the most significantly correlated proxy records with the NAO index over the historical period (Fig. 7). In addition, not all the methods have the same sensitivity to the proxy pre-selection. Indeed, Enet, PCR and RF methods have better validation results than PLS when all of the available proxy

records are used as predictors.

Our results suggest that enhancing the spatial coverage of the proxy records is not a necessary condition to improve the reconstruction. Indeed, we showed that using the densest proxy network (i.e., all of the available proxy records on $\mathcal{T}$) does not lead to better validation scores, due to the noise introduced by predictors that covary weakly with the target index (Fig. 6 and 7). Among the previous reconstruction studies, this kind of investigation have often been overlooked at the expense of increasing

the spatial density of the proxy records [Cook et al. (2002); Gray et al. (2004); Wang et al. (2017)]. Ortega et al. (2015) already showed the advantage of subsampling the proxy records more significantly correlated (i.e. $90\%$) with the NAO. The validation correlations obtained in their study are weaker than those we obtained here by using PCR on the 35 proxy records significantly correlated with the NAO index at the $90\%$ confidence level, from which 19 are the same in both studies.

Here, the best score ($\bar{s} \simeq 0.46$ on average) are obtained for the PLS method when only the proxy records significantly

correlated with the NAO index at the $95\%$ confidence level are kept (16 proxy records). These results are better than those obtained by Ortega et al. (2015), for the calibration constrained reconstruction ($r_{val} \in [-0.14; 0.58]; \bar{s} \simeq 0.24$) as well as for the model constrained reconstruction ($r \in [0.14; 0.64]; \bar{s} \simeq 0.42$) (see Ortega et al. (2015)). Nevertheless, it should be noted that these results have been obtained for a particular length of the training and the testing windows of (111 and 37, respectively). The sensitivity to this length will be assessed in the next section.



### 4.1.2 Sensitivity to the length of training and testing periods

To estimate the sensitivity to the length of the training and the testing window, we set again the reconstruction period to $\mathcal{T} = \{1000, \dots, 1970\}$, and the learning period to $T = \{1823, \dots, 1970\}$, with $n = 148$. Based on the findings of the previous section, we only keep the proxy records which are significantly correlated with the NAO index at the 95% confidence level

(18 proxy records, see section 4.1.1 and Fig. 6). We run $R$ reconstructions with different window sampling for each method by gradually increasing the length of the training window: from 5% to 95% of the initial size of the learning period, with a step of 5%. Fig. 8 shows the validation correlations obtained for these simulations. Small training windows length, may leads to an overlook of the general information in the data, which translates into negative and non-significant validation correlations (Fig. 8). On the contrary, using a very long training window gives very high validation correlations close to 1, but it also give

negative ones (Fig. 8), i.e. a very wide range of validation scores, suggesting that the testing period is too short to robustly validate the reconstruction.

Between these two extremes we find a large window where validation scores are relatively similar (from around 30% to 70%). To assess the best reconstruction, we search the score vector which has the highest validation scores on average among the vectors that own only significant and positive testing correlations. Following this rule, the optimal window split is 75%

of the total for the training ($n_{train} = 111; n_{test} = 37$) for PLS . For PCR, we find an optimal split for by using training samples with a length of 70% of the length of the training period ($n_{train} = 104; n_{test} = 44$). For RF, the optimal split is 45% ($n_{train} = 67; n_{test} = 81$), while for Enet, the optimal split is 65% ($n_{train} = 96; n_{test} = 52$) . Overall method which gives the highest validation correlations on average is the PLS, closely followed by PCR and Enet (Fig. 8).

We now address the degree of uncertainty associated to the way the training/testing windows are partitioned. Fig. 9 shows

the correlation between the reconstructions in the optimal window split (identified above), and the other alternative partitions. All correlation values thus obtained are particularly high, especially for training windows length representing at least a 45% of the total period, for which correlations are greater than 0.96, regardless of the method, except RF, for a training window length of correlations of 85% of the length of the initial periods . This suggests that the choice of the training period is not a crucial methodological source of uncertainty for the reconstruction, although it is worth to optimize it.

### 25  4.1.3 Sensitivity to the reconstruction period

In this section, we focus on the most efficient method (PLS) with the optimal training/testing windows length ($n_{train} = 104, n_{test} = 44$, see section 4.1.2) and we explore the impact of the reconstruction period, and hence, the learning period and the proxy set. Changing this period affects the final reconstruction in two different ways, both related with the final proxy selection. Firstly, it modifies the initial set of proxy records considered (as they need to cover the whole reconstruction period).

Secondly, by changing the period of overlap with the observations, it leads to different correlations between the proxy records and the NAO index, which would affect their significances and therefore the final proxy selection. Indeed, a proxy record significantly correlated with the NAO index at a given confidence level on a given time window, can be non-significantly





correlated with the NAO index with the same confidence level, but on another time window. This may be induced by physical processes that modifies the stationarity of the NAO and its teleconnections.

We run the reconstruction on 36 periods $\mathcal{T}$: from 1000-1965 to 1000-2000, with an increment of one year. By doing so, the number of available proxy records is not the same for each of the periods. Each reconstruction is performed by using only
proxy records significantly correlated at the $95\%$ confidence level with the NAO on the corresponding learning period.

Fig. 10 shows the evolution of the proxy predictor set and the validation correlations obtained for the different reconstructions and learning periods. Using the validation correlations as a quality metric, we find that the best reconstruction time window is 1000-1967 (19 proxy records used; Fig. 10). Indeed, the associated validation correlations ($\bar{s} = 0.48; r \in [0.11, 0.68]$) are on average significantly greater than all of the others (at the $95\%$ confidence level). In addition, we observe two significant drops
in validation correlations at the $95\%$ confidence level, depending on the size of the reconstruction period: One from 1978 to 1979 and one from 1994 to 1995 (Fig. 10). Both can be associated to important changes in the number and the nature of proxy predictor sets (Fig. 10). For the other methods, we found that the optimal reconstruction period for Enet and RF is 1000-1973 (not shown), while the optimal reconstruction period for PCR is 1000-1970 (not shown).

In contrast with the length of the training periods, the choice of the reconstruction period appears as an important source of
reconstruction uncertainty. This parameter strongly affects the reconstruction by modifying directly or indirectly the predictors. Thus, we recommend to determine this period carefully with numerous simulations on different time windows, following the approach we presented here.

### 4.2   Reconstructions assessment

We now compare the best reconstructions obtained for each of the methods. The four optimized reconstructions are obtained
by maximizing the validation correlations on the training/testing period (see section 4.1.2) and the total reconstruction period (PLS: see section 4.1.3; other methods: not shown), using the proxy records significantly correlated at the $95\%$ confidence level with the NAO on the corresponding learning period (section 4.1.1 and 4.1.3).

#### 4.2.1   Comparison with previous work

Fig. 11 shows the different reconstructions of the NAO, including the Ortega et al. (2015) calibration constrained reconstruction
(only proxy-based), and Tab. 1 exhibits the paired correlations between the 5 reconstructions. All the reconstructions are significantly correlated with each other at the $99\%$ confidence level on their overlap periods even if they were performed with different proxy groups and learning periods (Tab. 1). As they also have been optimised for multiple sources of sensitivity for the reconstruction, these results strongly support the fact that the reconstructions we propose are reliable to translate the variations of the NAO index over the past millenium.

According to the validation scores, the best reconstruction that we found has been obtained using the PLS method on the reconstruction period 1000-1967, using the 19 proxy records significantly correlated with the NAO index on this period. The



averaged validation scores attributed to each of the best reconstruction for each method are: $\bar{s}_{PCR} = 0.41, \bar{s}_{RF} = 0.41, \bar{s}_{RF} = 0.43, \bar{s}_{PLS} = 0.48$. The correlation coefficient between the original NAO index and the PLS reconstruction is about 0.63 (p<0.01) on the time window 1823-1967 while its correlation with the Ortega et al. (2015) reconstruction is about 0.45 (p<0.01) on the time window 1823-1969. Furthermore, the PLS validation correlations are greater than those from Ortega et al. (2015), on average. This is true both for the calibration constrained reconstruction ($r_{val} \in [-0.14; 0.58]; \bar{s} = 0.24$) and the model constrained reconstruction ($r \in [0.14; 0.64]; \bar{s} = 0.42$) (see Ortega et al. (2015)). To understand this difference it is important to note that the best performing reconstruction in Ortega et al. (2015) has a substantially weaker correlation with the observed NAO in their overlap period (r=0.45, p<0.01) that all the NAO reconstructions discussed here for the different methods ($r \in [0.56, 0.63]$, $p < 0.01$). The 5 reconstructions, including Ortega et al. (2015) do not show a predominant positive NAO phase during the MCA, contrary to the hypothesis formulated by Trouet et al. (2009). This means that NAO may had not been directly involved in the onset of the MCA and LIA as suggested Trouet et al. (2009).

The different optimizations performed on the different methods allowed us to find the optimal reconstruction. Hence, we statistically verified that the reconstruction from this study is more robust and reliable than those in Ortega et al. (2015). This improvement in performance may also arise from the inclusion of new relevant proxy records into the reconstruction, but also the use of a new statistical regression method. The PLS reconstruction uses 19 different proxy records, 12 of them have been used in the NAO reconstruction from Ortega et al. (2015) (see Fig. 12). Among the 7 proxy records we added, there is a tree ring proxy record from Asia, with a medium negative weight in the reconstruction (Fig. 12). This proxy record (`Asi_221`) belongs to the Pages2K database 2014 version [Pages 2K Consortium (2013)], but no associated reference is provided. The six other proxy records come from ice cores and are located in the Arctic area: three of them have been recorded in Greenland [Vinther et al. (2010)], two have been recorded in North Canada [Vinther et al. (2008), Meeker and Mayewski (2002)] , and the last one has been record in Northern Sweden [Young et al. (2012)] (Fig. 12). For the other proxy records, the weight we attributed to them have the same signs than those found in Ortega et al. (2015).

### 4.2.2 Response to external forcing

We now focus on the response of the NAO to external forcing: volcanic aerosols, Total Solar Irradiance (TSI), and $CO_2$ concentration. Indeed, Ortega et al. (2015) suggested that a positive NAO phase is triggered after strong volcanic eruptions, a response that is not reproduced over the last millennium by model simulations [Swingedouw et al. (2017)]. By applying composite analysis of the NAO response to the 10 strongest volcanic eruptions (see Supplementary table 2) which occurred during the last millenium, and using dates from 4 different reconstructions of the last millenium volcanic activity [Gao et al. (2008); Crowley and Unterman (2013); Sigl (2014); Ortega et al. (2015); Supplementary table 2], we obtained consistent results with Ortega et al. (2015) for the four regression methods developed here: a positive NAO response 2 years following the eruption onset (Fig. 13). Contrary to the findings of Ortega et al. (2015), we did not find a second significant NAO response after 4 years. By using a Monte-Carlo approach as in Ortega et al. (2015), the 2-years lagged NAO response we obtain has significance levels greater than 90% for all methods, all volcanic reconstructions and all composites, except for the composite





RF based on the volcanic activity reconstruction from Gao et al. (2008) (Fig. 13). The RF reconstruction is the less reliable among our four reconstructions, since it has the worst validation scores on average between our four reconstructions (section 4.2.1), so that the NAO response two years after the eruption is a robust result.

On the other hand, we did not find any significant correlation of the NAO with any available reconstructgions of the TSI
[Crowley (2000); Vieira et al. (2011)]. Moreover, none of the reconstructions (including Ortega et al. (2015)) shows clear negative phases during the Maunder and the Sporer minima as some model simulations were suggesting [Shindell et al. (2004)]. In addition, no significant correlation on the pre-industrial era has been found with a $CO_2$ reconstruction based on a Law Dome (East Antarctica) ice core [Etheridge et al. (1996)], indicating that the NAO is not linearly associated with $CO_2$ variations.

## 5   Conclusions

We have proposed and described four statistical methods for reconstructing any modes of climate variability and have compared them for a particular example: the reconstruction of the NAO. By investigating and minimizing the sources of reconstruction uncertainty, due to the method used (sections 3, 4.1.2 and 4.2.1), the time frame considered (section 4.1.3) and the proxy selection (sections 4.1.1 and 4.1.3), we found the optimal NAO reconstructions, all providing better validation and calibration results than previous studies (section 4.2.1). All the reconstructions show a positive NAO response the year 2 following volcanic
eruptions, in agreement with Ortega et al. (2015). Moreover they also presents low-frequency negative phases at the multi-decadal scale (section 4.2.1), which may induce anomalously cold winter conditions in Europe during these periods (e.g. $11^{th}, 12^{th}, 15^{th}$ and $18^{th}$ centuries).

We have showed that using proxy records with a strong correlation with the index to be reconstructed over the overlapping period is a good means for improving the validation scores, and hence allow more reliable reconstructions. Among the 539
available proxy records collected, containing the PAGES 2K database 2014 version [Pages 2K Consortium (2013)], which is a well-verified high resolution proxy collection, only 19 covers the reconstruction period 1000-1967 and are significantly correlated with the NAO index (at the $95\%$ confidence level) on the period 1823-1967. Gathering new proxy records, significantly correlated with the NAO, may be a reliable source of reconstruction improvement. The toolbox we developed in this paper should allow to perform such new reconstructions, thanks to a device made available to the community (cf. code availability
and section 2).

In order to extract the most robust reconstruction, numerous simulations are needed. To facilitate it, the statistical tool we developed performs a reconstruction by considering several entries: an index of the climate mode, the reconstruction period, the length of the training window (in proportion of the total length of the learning window), the number of training/testing period samplings, and a threshold confidence level for the correlation between the proxy records and the target index (appendix
1). This modular statistical tool is an opportunity to reconstruct quickly, and with quantified reliability, several climate modes. This may allow us to improve our understanding of the last millennium large-scale climate variations, such as the MCA and the LIA, as well as the interactions between different climatic modes, which will be analysed in future studies.



*Code and data availability.* The statistical toolbox code and the proxy records database are available at the link: https://zenodo.org/record/1403146#.W4UMUGaB2qA.





## Appendix A: Supplementary table 1 : Proxy records not in Pages 2K

| N° | Code | Location | Longitude (°E) | Latitude (°N) | First year | Laste year | Archive | Proxy type | Related variable | Seasonality | Ref. |
|---|---|---|---|---|---|---|---|---|---|---|---|
| 1 | accr-b18-yr | B18 | -36.40 | 76.60 | 1000 | 1992 | Ice core | Snow accumulation | Precip. | Annual | Miller and Schwager (2004) |
| 2 | accr-crete-yr | Crete | -38.50 | 71.00 | 1000 | 1973 | Ice core | Snow accumulation | Precip. | Annual | Andersen et al. (2006) |
| 3 | accr-GISP2-yr | GISP2 | -38.50 | 72.60 | 1000 | 1988 | Ice core | Snow accumulation | Precip. | Annual | Cuffey et al. (1995) |
| 4 | d18O-agass-79 | Agassiz | -77.00 | 80.70 | 1000 | 1972 | Ice core | $\delta^{18}O$ | SAT | Annual | Fisher et al. (1995) |
| 5 | d18O-dasuopu-yr | Dasuopu | 85.00 | 28.00 | 1010 | 1997 | Ice core | $\delta^{18}O$ | SAT | JJAS | Thompson et al. (2003) |
| 6 | icecore-GISP2-ssNA | GISP2 | -39.00 | 73.00 | 1000 | 1986 | Ice core | Sea salt Na | SLP | DJF | Meeker and Mayewski (2002) |
| 7 | lake-allos-flood | Allos Lake | 5.00 | 44.23 | 1000 | 2009 | Lake sediment | Flood deposit thickness | Precip. | SON | Wilhelm et al. (2012) |
| 8 | lake-bigrou-vthick | Big round Lake | -71.00 | 69.87 | 1000 | 1995 | Lake sediment | Varve thickness | SAT | JAS | Thomas and Briner (2008) |
| 9 | lake-braya-temp | Braya Lake | -51.03 | 66.99 | 1001 | 2005 | Lake sediment | Alkenone UK37 | SAT | Annual | von Gunten et al. (2012) |
| 10 | lake-castor-lime | Castor Lake and Lime Lake | -118.45 | 48.71 | 1000 | 2000 | Lake sediment | $\delta^{18}O$ | Precip. | NDJF | Steinman et al. (2012) |
| 11 | lake-donard-varves | Donard Lake | -61.35 | 66.67 | 1000 | 1995 | Lake sediment | Varve thickness | SAT | JJA | Moore et al. (2001) |
| 12 | lake-hvitar-icel | Hvtarvatn Lake | -19.80 | 64.60 | 1000 | 2000 | Lake sediment | Varve thickness | SAT | JJAS | Larsen et al. (2011) |
| 13 | lake-itilliq-bsi | Itilliq Lake | 67.10 | 69.90 | 1000 | 2005 | Lake sediment | Organic matter and insect assemblages | SAT | Annual | Thomas et al. (2011) |
| 14 | lake-lowmur-maccum | Lower Murray Lake | -69.50 | 81.33 | 1000 | 1969 | Lake sediment | Mass accumulation | SAT | JJA | Cook et al. (2008) |
| 15 | lake-ximencuo-toc | Ximencuo Lake | 101.11 | 33.38 | 1005 | 2004 | Lake sediment | $\delta^{13}C$ | Precip. | Annual | Pu et al. (2013) |
| 16 | ocsed-capeghir-sst | Cape Ghir | -10.09 | 30.84 | 1000 | 1971 | Ocean sediment | UK37 | SST | Annual | McGregor et al. (2007) |
| 17 | ocsed-md99-2275 | MD992275 | -17.70 | 66.60 | 1000 | 2001 | Ocean sediment | UK37 | SST | Annual | Sicre et al. (2008),Sicre et al. (2011) |
| 18 | ocsed-subnatlan-Ausst | Subpolar North Atlantic | -27.91 | 57.45 | 1000 | 2004 | Ocean sediment | Diatom | SST | August | Miettinen et al. (2012) |
| 19 | speleo-crystal-d18O | Crystal cave | -121.00 | 36.90 | 1000 | 2007 | Speleothem | $\delta^{18}O$ | SAT | Annual | McCabe-Glynn et al. (2013) |
| 20 | speleo-juxtla-rain | Juxtlahuaca cave | -99.20 | 17.40 | 1000 | 1999 | Speleothem | $\delta^{18}O$ | Precip | MAM | Lachniet et al. (2012) |
| 21 | speleo-so-1 | Sofular cave | 31.93 | 41.42 | 1000 | 2006 | Speleothem | $\delta^{18}O$ | precip | annual | Fleitmann et al. (2009) |
| 22 | speleo-su-96-7 | Uamh an Tartair cave | -4.93 | 58.14 | 1000 | 1995 | Speleothem | Bandwidth | Precip. | DJFM | Baker et al. (2002) |
| 23 | tree-alps-Tjjas | European Alps | 9.00 | 46.00 | 1000 | 2004 | Tree ring | Tree ring MXD | SAT | JJAS | Büntgen et al. (2012) |
| 24 | tree-AR050-stah | Black Swamp | -91.30 | 35.15 | 1019 | 1980 | Tree ring | Tree ring width | SAT | Annual | Stahle (1996a) |
| 25 | tree-AR052-stah | Mayberry Slough | -89.00 | 35.50 | 1000 | 1990 | Tree ring | Tree ring width | SAT | Annual | Stahle and Cleaveland (2005a) |
| 26 | tree-CA051-tosh | San Gorgonio | -116.82 | 33.40 | 1000 | 1970 | Tree ring | Tree ring width | SAT | Annual | Tosh (1994) |
| 27 | tree-CA528-grayb | Flower lake | -115.70 | 39.90 | 1000 | 1987 | Tree ring | Tree ring width | Precip. | NDJFM | Bunn et al. (2005) |
| 28 | tree-CA529-grayb | Timber Gap Upper | -117.00 | 37.30 | 1000 | 1987 | Tree ring | Tree ring width | Precip. | NDJFM | Bunn et al. (2005) |
| 29 | tree-CA530-grayb | irque Peak | -117.50 | 35.00 | 1000 | 1987 | Tree ring | Tree ring width | Precip | NDJFM | Bunn et al. (2005) |
| 30 | tree-ca605-king | Mammoth Peak | -119.50 | 41.00 | 1000 | 1996 | Tree ring | Tree ring width | Precip. | NDJFM | Bunn et al. (2005) |
| 31 | tree-CA636-graum | Boreal Plateau | -119.50 | 36.45 | 1000 | 1992 | Tree ring | Tree ring width | Precip. | NDJFM | Bunn et al. (2005) |
| 32 | tree-CA637-graum | Upper Wright Lakes | -115.60 | 35.50 | 1000 | 1992 | Tree ring | Tree ring width | Precip. | NDJFM | Bunn et al. (2005) |
| 33 | tree-CA640-graum | Hamilton | -118.92 | 39.00 | 1000 | 1988 | Tree ring | Tree ring width | Precip. | NDJFM | Bunn et al. (2005) |
| 34 | tree-co572-woodho | Lily Lake | -105.60 | 40.30 | 1000 | 1998 | Tree ring | Tree ring width | SAT | Annual | Woodhouse and Brown (2006) |
| 35 | tree-firth-anchuk | Firth River | -141.63 | 68.65 | 1073 | 2002 | Tree ring | Tree ring MXD | SAT | JJA | Anchukaitis et al. (2013) |
| 36 | tree-FL001-stah | Choctawhatchee River | -85.92 | 30.45 | 1000 | 1992 | Tree ring | Tree ring width | SAT | Annual | Stahle and Cleaveland (2005b) |
| 37 | tree-forfjo-cloud | Forfjorddalen | 15.73 | 68.80 | 1000 | 2001 | Tree ring | Tree ring $\delta^{13}C$ | Cloud % | JJA | Young et al. (2012) |
| 38 | tree-LA001-stah | Big Cypress | -92.97 | 32.25 | 1000 | 1988 | Tree ring | Tree ring width | SAT | Annual | Stahle (1996b) |
| 39 | tree-mor-pdsi | Morocco | -5.00 | 33.75 | 1049 | 2001 | Tree ring | Tree ring width | SPI | FMAMJ | Esper et al. (2007) |
| 40 | tree-mt112-king | Yellow Mountain Ridge I | -112.00 | 45.30 | 1000 | 1998 | Tree ring | Tree ring width | Precip. | NDJFM | Graumlich et al. (2003) |
| 41 | tree-mt113-wagon | Yellow Mountain Ridge (Entire Bark Trees) | -109.80 | 45.60 | 1000 | 1998 | Tree ring | Tree ring width | Precip. | NDJFM | Graumlich et al. (2003) |
| 42 | tree-NM584-touch | Mesa Alta | -106.60 | 36.20 | 1000 | 2007 | Tree ring | Tree ring width | Precip. | ONDJFMAMJ | Touchan et al. (2011) |
| 43 | tree-NV516-grayb | Hill 10842 | -114.20 | 38.90 | 1000 | 1984 | Tree ring | Tree ring width | SAT | Annual | Graybill (1994a) |
| 44 | tree-nv517-grayb | Spring Mountains Lower | -114.70 | 34.30 | 1000 | 1984 | Tree ring | Tree ring width | SAT | Annual | Graybill (1994a) |
| 45 | tree-SCpla-precip | S. Colorado Plateau I | -109.30 | 37.50 | 1000 | 1987 | Tree ring | Tree ring width | Precip. | October-July | Salzer and Kipfmueller (2005) |
| 46 | tree-SCpla-temp | S. Colorado Plateau II | -110.70 | 36.50 | 1000 | 1996 | Tree ring | Tree ring width | SAT | Annual | Salzer and Kipfmueller (2005) |
| 47 | tree-sc004-stahle | Four Holes Swamp | -80.42 | 33.18 | 1001 | 1985 | Tree ring | Tree ring width | Precip. | MAMJ | Stahle and Cleaveland (2005a) |
| 48 | tree-siber-temp | Taimyr-Putoran | 103.00 | 71.29 | 1000 | 1996 | Tree ring | Tree ring width | SAT | Annual | Naurzbaev et al. (2002) |
| 49 | tree-swit177-schwein | Lauenen + div. Stao | 6.50 | 46.42 | 1000 | 1976 | Tree ring | Tree ring width | SAT | JJA | H (1995) |
| 50 | tree-UT508-grayb | Wild Horse Ridge | -110.10 | 40.00 | 1000 | 1985 | Tree ring | Tree ring width | SAT | Annual | Graybill (1994c) |
| 51 | tree-UT509-grayb | Mammoth Creak | -112.67 | 37.65 | 1000 | 1989 | Tree ring | Tree ring width | SAT | Annual | Graybill (1994d) |
| 52 | tree-albermale-trw | Albermale Sound | -76.00 | 36.00 | 934 | 2005 | Tree ring | Tree ring width | PDSI | July | Stahle et al. (2013) |
| 53 | tree-arjeplog-bi | Arjeplog | 17.90 | 66.50 | 1200 | 2010 | Tree ring | Tree ring BI | SAT | JJA | Björklund et al. (2014) |
| 54 | tree-jamtland-mxd | Jamtland | 15.00 | 63.10 | 800 | 2011 | Tree ring | Tree ring MXD | SAT | AMJJAS | Zhang et al. (2016) |
| 55 | tree-colzad-trw | Col du Zad | -5.10 | 33.00 | 984 | 1984 | Tree ring | Tree ring width | PSI | FMAMJ | Esper et al. (2007) |
| 56 | tree-forf-mxd | Forfjorffalen-x | 15.70 | 68.80 | 978 | 2005 | Tree ring | Tree ring MXD | SAT | AMJJAS | McCarroll et al. (2013) |
| 57 | tree-khibiny-bi | Khibiny | 33.50 | 67.50 | 821 | 2005 | Tree ring | Tree ring BI | SAT | JJA | McCarroll et al. (2013) |
| 58 | tree-laanila-mxd | Laanila | 27.30 | 68.50 | 800 | 2005 | Tree ring | Tree ring MXD | SAT | JJA | McCarroll et al. (2013) |
| 59 | tree-manitoba-trw | S Manitoba | -97.10 | 49.50 | 1409 | 1998 | Tree ring | Tree ring width | Precip. | Annual | George and Nielsen (2002) |
| 60 | tree-mesoamerica-trw | Mesoamerica | -100.00 | 20.00 | 800 | 2008 | Tree ring | Tree ring width | PDSI | June | Stahle et al. (2011) |
| 61 | accr-NGRIP-s | NGRIP-s | -42.00 | 76.00 | 800 | 1995 | Ice core | Snow accumulation | Precip | Annual | Andersen et al. (2007) |
| 62 | tree-potoriv-trw | Potomac River | -77.50 | 39.30 | 950 | 2001 | Tree ring | Tree ring width | Stream flow | MJJAS | Maxwell et al. (2011) |
| 63 | tree-quebecx-mxd | Quebec-x | -77.50 | 39.30 | 1373 | 1988 | Tree ring | Tree ring MXD | SAT | MJJAS | Schneider et al. (2015) |
| 64 | tree-SCengland-trw | SC England | -1.40 | 51.50 | 950 | 2009 | Tree ring | Tree ring width | Precip. | MJJ | Wilson et al. (2013) |
| 65 | tree-sodankyla-thi | Sodankyla | 27.00 | 67.00 | 800 | 2007 | Tree ring | Tree height increment | SAT | JJA | Lindholm and Jalkanen (2011) |
| 66 | tree-southfin-mxd | Southern Finland | 28.50 | 61.50 | 800 | 2000 | Tree ring | Tree ring MXD | SAT | MJJAS | Helama et al. (2014) |
| 67 | tree-SWturkey-trw | SW Turkey | 31.00 | 37.00 | 1339 | 1998 | Tree ring | Tree ring width | Precip. | MJ | Touchan et al. (2003) |
| 68 | tree-tyrol-mxd | Tyrol | 12.50 | 48.00 | 1053 | 2003 | Tree ring | Tree ring MXD | SAT | JAS | Schneider et al. (2015) |
| 69 | d18O-NIshelf-yr | North Icelandic Shelf | 66.53 | -18.20 | 953 | 2000 | Ice core | $\delta^{18}O$ | SAT | Annual | Reynolds et al. (2016) |





**Appendix B: Supplementary table 2 : Ten large volcanic eruption common to four reconstructions and studies**

| Volcano | Location | Gao et al. (2008) | Crowley and Unterman (2013) | Sigl (2014) | Ortega et al. (2015) |
|---|---|---|---|---|---|
| Unknown | Unknown | 1227 | 1229 | 1229 | 1229 |
| Samalas | Indonesia | 1258 | 1258 | 1257 | 1257 |
| Unknown | Unknown | 1284 | 1286 | 1285 | 1285 |
| Huaynaputina | Peru | 1600 | 1600 | 1600 | 1600 |
| Parker | Phillippines | 1641 | 1641 | 1641 | 1640 |
| Serua | Indonesia | 1693 | 1696 | 1694 | 1693 |
| Unknown | Unknown | 1809 | 1809. | 1809 | 1809 |
| Tambora | Indonesia | 1815 | 1816 | 1815 | 1815 |
| Cosiguina | Nicaragua | 1835 | 1835 | 1834 | 1835 |
| Krakatau | Indonesia | 1883 | 1884 | 1884 | 1883 |





## Appendix C: Using of the multi-statistical tool

Data files, source codes, parameter setting files and a description file with all useful informations and examples are available here : LINK. This tool works as a model. First, the files "db_proxy.txt", "params.txt", "run.sh", "runR.txt" and the source code "mov_reconstruction.r" have to be store in the same directory. A file containing the target index has to be added to this folder. It

must contains two informations : the first is observations years and the second is the target index values. "csv" and "txt" format are available for this file. Finally, whatever the name of this file it has to be informed in "params.txt".

The following informations have to be gave by the user in "params.txt" :

1. `Name` : Name of the reconstruction. Gives the name of the folder where results will be output.

2. `y_start` : Year where the reconstruction begins.

3. `y_stop` : Year where the reconstruction stops.

4. `method` : Statistical learning method. Currently available possibilities are :

   – pls, PLS : Partial Least Squares.

   – pcr, PCR : Principal Components Regression.

   – rf, RF : Random Forest.

– enet, ENET : Elastic Net regularization.

5. `R` : Number of repetitions of calibration/validation splits. The computational time hardly depend on this parameter.

6. `train_window` : Length of the training window (as % of the learning set)

7. `tests` : If T (or TRUE), a proxy selection is made. If F (or FALSE), all proxy records available are used for the reconstruction.

8. `conf` : Confidence level of significance correlation test for proxy selection. Must be between 0 and 1 excluded. If tests=F, this argument is ignored.

9. `index_file` : path to the txt or csv file containing the climate index. It has to own two columns: the first has to be the years of observation, and the second, the observations.

10. `seed` : [Optional] set the seed for random sampling. If the user run different methods for reconstructing the same indices,
setting the same seed for the different runs allows to use the same random samples.

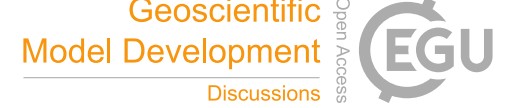

Once the parameter file is set, the user just has to run the script "run.sh". When the run is done, a folder named as it is informed in "params.txt" file is created. Otherwise, a suite of files is given :

1. `final_reconstruction.txt` : A two dimensional array containing two columns : Reconstruction years and reconstructed index.

2. `val_samples.txt` : A two dimensional array containing all the validation year samples by row (R rows).

3. `reconstructions.txt` : A two dimensional array of size $R \times N$ containing the R individual reconstructions on the R samples.

4. `val_cors.txt` : A vector containing validation correlations.

5. `val_rmse.txt` : A vector containing validation root mean squared errors.

6. `proxy_records.txt` : A vector containing the IDs (see, Pages 2K database metadata and supplementary table 1) of the proxy used for reconstruction. These proxy records are those which overlap the reconstruction period given and which are significantly correlated with the target index (if "tests" is T) at the confidence level given in "conf".

The source code is commented such that it can be modified using a few R knowledges.

Important remark: If you aim to find the same results, please use the NAO index provided with codes and data (see code and

data availability). You also need to set the seed at 3, as the results are obtained from random eperiments.



## Appendix D: Statistical test for correlation significance

The statistical test we use in all the study as been firstly proposed by Bretherton et al. (1999) to avoid the individual autocorrelation effects on the correlation between two series. This is done by adjusting the degree of freedom. However, a simplification of this test has been proposed by McCarthy et al. (2015) by only using the first order autocorrelations to modify the degree of
freedom.

Let $X = (X_t)_{t \in T}$ and $Y = (Y_t)_{t \in T}$ two time series of same length. The correlation between the two series is given by :

$$r = cor(X, Y) = \frac{cov(X, Y)}{\sqrt{Var(X)} \cdot \sqrt{Var(Y)}} \tag{D1}$$

We denote $a_1^{(X)}$ and $a_1^{(Y)}$ the first order lag of the respective autocorrelation functions of $X$ and $Y$. The effective number of degrees of freedom [Bretherton et al. (1999)] is then given by :

$$N_{eff} = N_{obs} \cdot \frac{1 - a_1^{(X)} \cdot a_1^{(Y)}}{1 + a_1^{(X)} \dot{a}_1^{(Y)}} \tag{D2}$$

The statistics is then calculated as :

$$t^{stat} = \sqrt{N_{eff}} \cdot \frac{r}{\sqrt{1 - r^2}} \tag{D3}$$

For $\alpha \in ]0, 1[$, the statistic $t^{stat}$ is compared to the $1 - \frac{\alpha}{2}$ order quantile of a Student distribution with $N_eff$ degrees of freedom.

*Author contributions.* Simon Michel implemented the statistical toolbox delivered with this paper. He also ran and compared the different
reconstructions and scores presented in this paper. Simon Michel mainly wrote this paper with the support of the other authors. Didier
Swingedouw supervised the whole redaction and results analyses of this publication and has contributed to the structuration of the general
methodology presented. Juliette Mignot, Myriam Khodri and Pablo Ortega contributed to the redaction of the paper and have also been key
contributors to the results analyses. Marie Chavent has contributed to the redaction of the mathematical formalism of the statistical toolbox
presented section 2 and 3.

*Competing interests.* No competing interests are present.



*Acknowledgements.* This research was partly funded by the Universite de Bordeaux. It is also funded by the LEFE-IMAGO project. To develop the statistical tool and analyse its outputs, this study benefited from the the IPSL Prodiguer-Ciclad facility, supported by CNRS, UPMC Labex L-IPSL. Finally, this study used the PAGES 2K database 2014 verstion, available online and supported by the PAGES group.

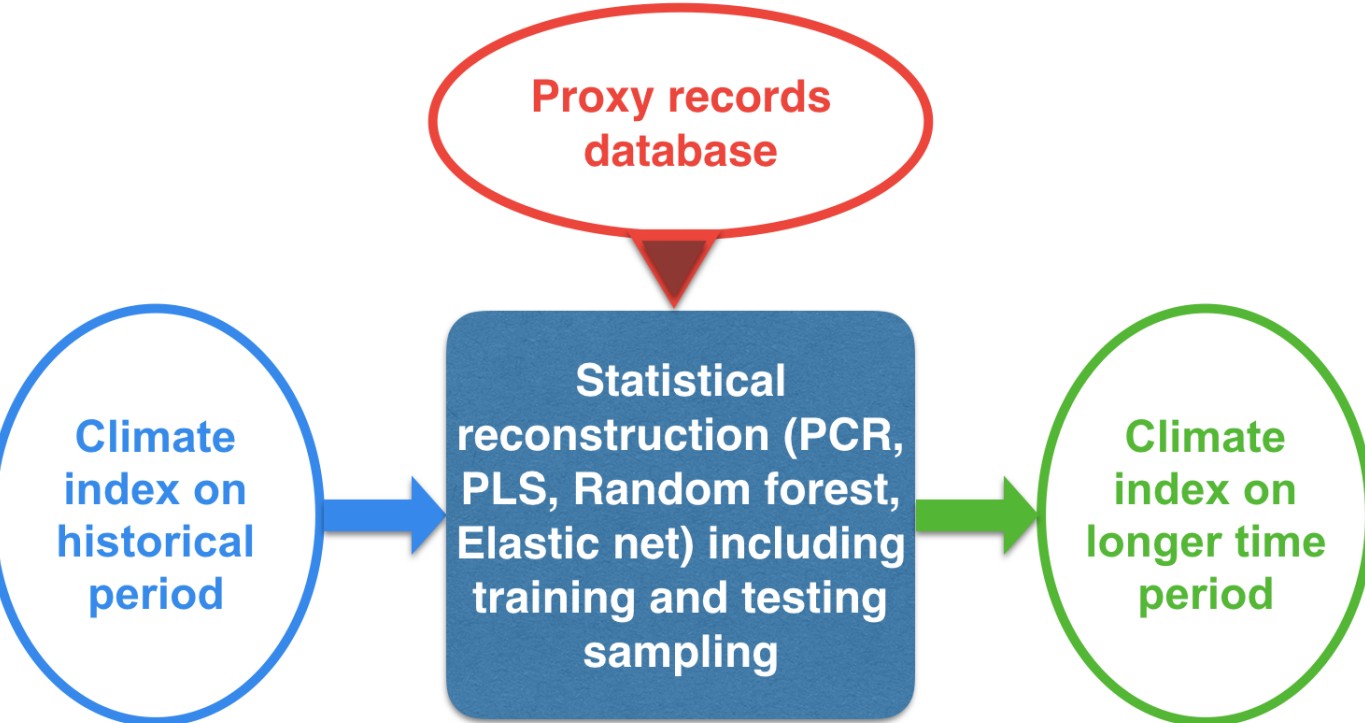

**Figure 1.** Scheme summarising the main features of the proposed statistical toolbox.





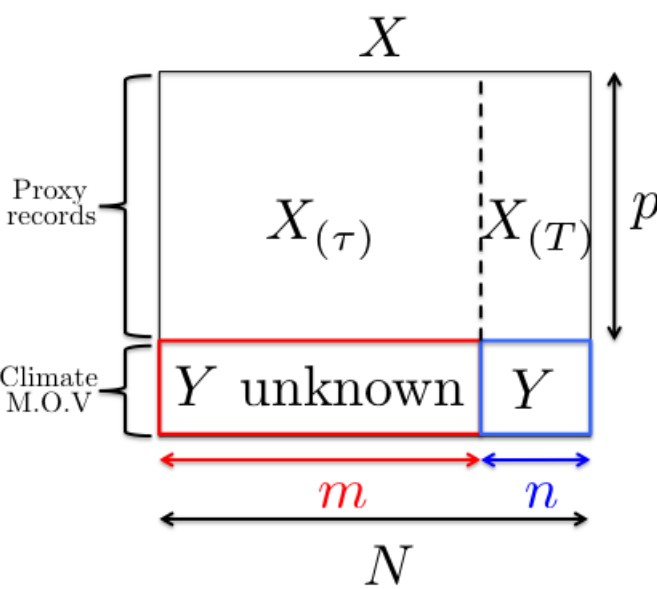

**Figure 2.** Scheme of the initial data. $X$ and $Y$ are respectively the proxy records matrix and the index of the considered mode of variability (M.O.V) index. $N$ is the size of the common period of all proxy records. $n$ is the size of the common period of all proxy records and the index of the mode of variability. $m$ is the size of the common period of all proxy records, where the mode of variability is not known. $p$ is the number of proxy records. $X_{(T)}$ is the sub-matrix of $X$ where the mode of variability is known. $X_{(\tau)}$ is the sub-matrix of $X$ where the mode of variability is not known.



$$X \qquad\qquad Y$$

$$X_{(i)} \quad X_{(-i)} \qquad\qquad Y_{(-i)} \quad Y_{(i)}$$

$$s_i^k = \sqrt{(Y_{(i)} - \mathcal{M}(\{X_{(-i)}, Y_{(-i)}\}, X_{(i)}, \theta_k))^2}$$

$$\Theta = (\theta_1, \ldots, \theta_K) \qquad\qquad \text{For } k \in \{1, \ldots, K\}$$

$$\text{For } i \in \{1, \ldots, n\}$$

$$s_i = (s_i^1, \ldots, s_i^K)$$

$$\bar{s}^k = \frac{1}{n} \sum_{i=1}^{n} s_i^k$$

$$\theta_{LOO} = \underset{k \in \{1, \ldots, K\}}{argmin}\{\bar{s}^k\}$$

**Figure 3.** Scheme of a leave-one-out cross validation process to select the optimal parameter of a specific learning method $\mathcal{M}$. $X$ is the input set of predictors and $Y$ the corresponding variability mode index. $\forall 1 \leq i \leq n$, $\{X_{(i)}, Y_{(i)}\}$ is the $i^{\text{th}}$ observation and $\{X_{(-i)}, Y_{(-i)}\}$ contains all observations except the $i^{\text{th}}$. $\Theta = (\theta_1, \ldots, \theta_K)$ is the ensemble of possible values of $\theta \in \mathbb{R}^q$.



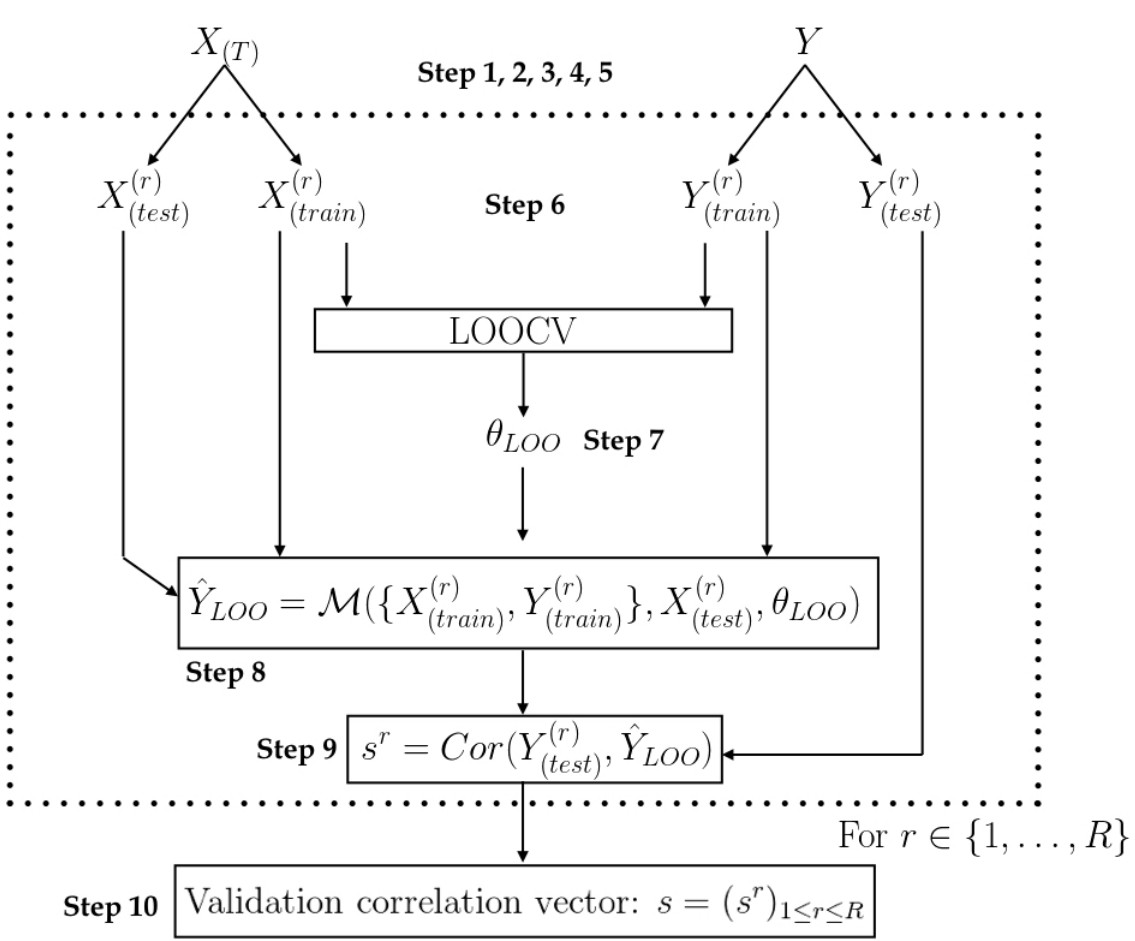

**Figure 4.** Scheme of the whole process for scores calculation for a given method $\mathcal{M}$. $Y$ is the index of the chosen mode of variability. $X_{(T)}$ is the proxy dataset restricted to the period where $Y$ is known. $\{X_{(train)}^{(r)}, Y_{(train)}^{(r)}\}$ is the $r$th training sample and $\{X_{(test)}^{(r)}, Y_{(test)}^{(r)}\}$ is the $r$th testing sample. $\theta_{LOO}$ is the empirically optimal set of parameters obtained by applying the LOOCV (Fig. A3; section 2.5.1)





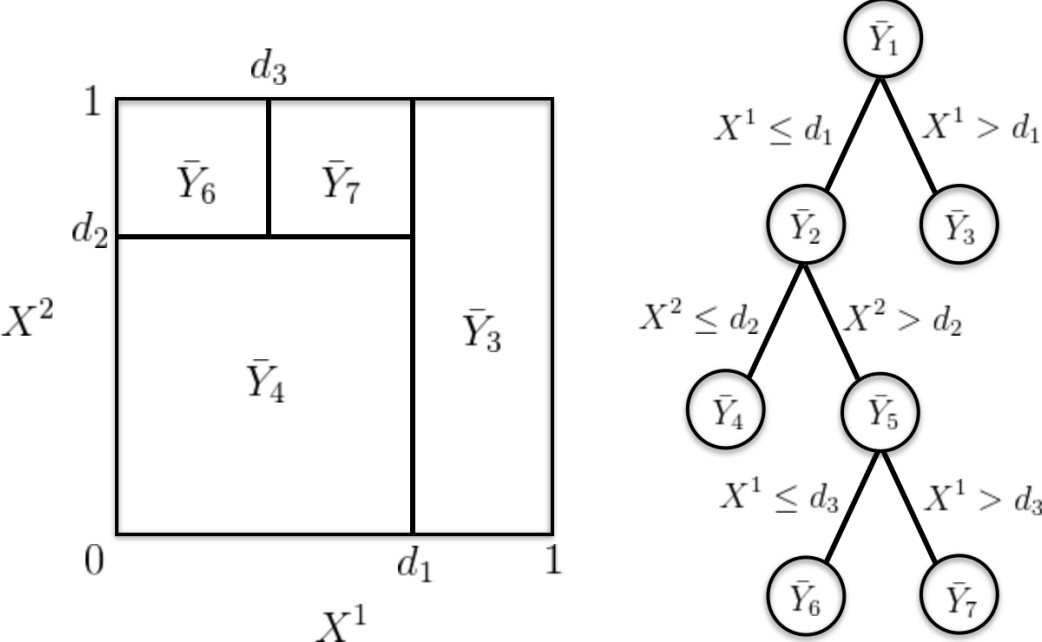

**Figure 5.** Dyadic partition of the unit square (left) and its corresponding regression tree (right). $Y$ is the predictand and $X^1, X^2, X^3$ are the predictors. $d_1$, $d_2$ and $d_3$ are the optimal thresholds of the three steps respectively.



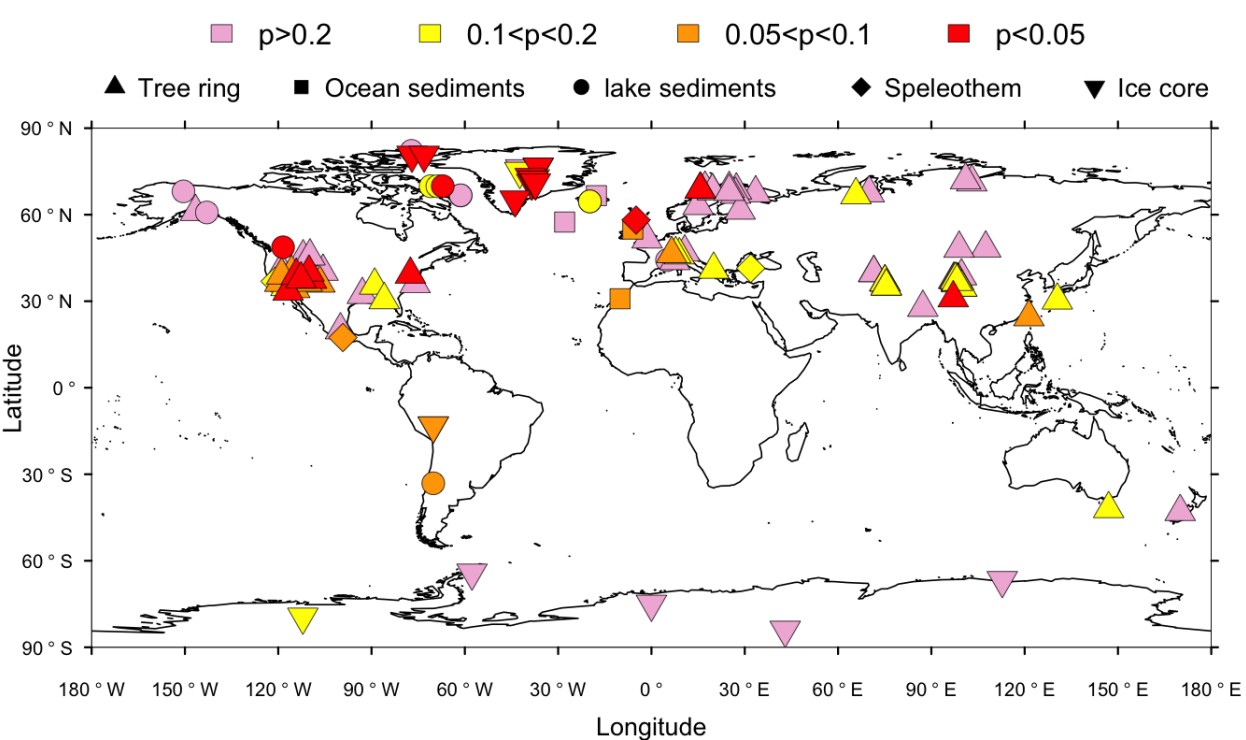

**Figure 6.** Geolocation, types and correlation confidence level between the 122 available proxy records for the period 1000-1970, and the NAO index on the period 1823-1970





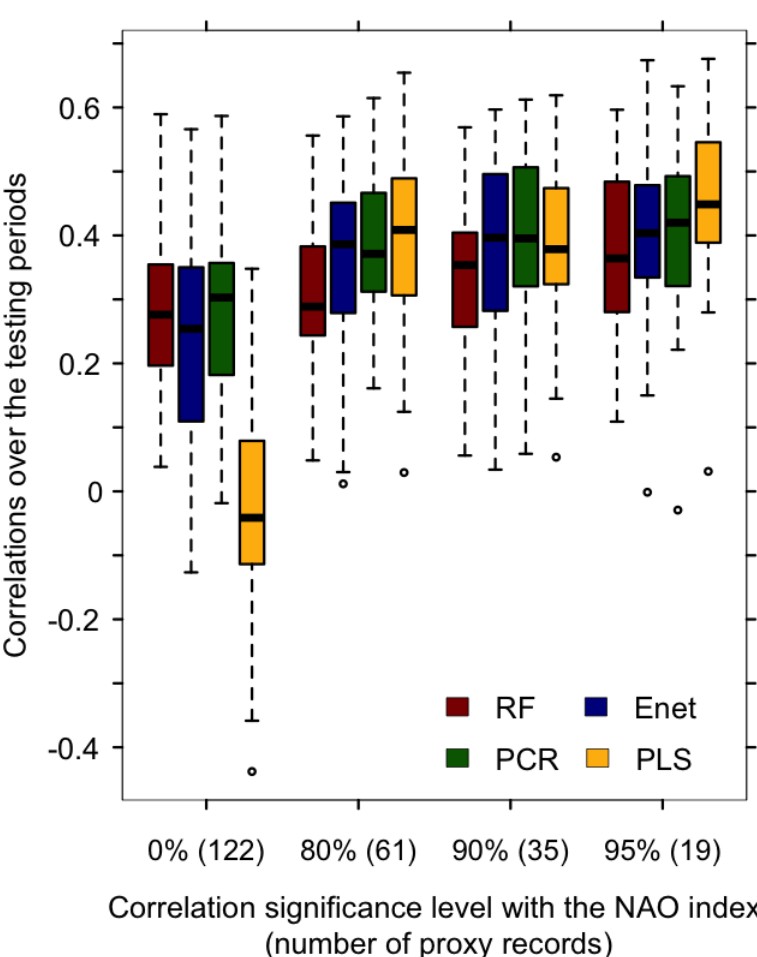

**Figure 7.** Boxplot of validation correlations obtained for the four methods and different groups of proxy records by reconstructing the NAO index on the period 1000-1970 with $R = 50$ validation/calibration samples. Calibration samples size is $n_{train} = 111$, and validation samples size is $n_{test} = 37$. Green boxplots are the validation correlations obtained for the PCR method. Yellow boxplots are the validation correlations obtained for the PLS method. Red boxplots are the validation correlations obtained for the RF method. Blue boxplots are the validation correlations obtained for the Enet method. The first cluster of boxplots is the validation correlations obtained by using all the available proxy records over the period (122 proxy records). The second cluster of boxplots is the validation correlations obtained by using only proxy records significantly correlated with the NAO index at the 80% confidence level (61 proxy records). The third cluster of boxplots is the validation correlations obtained by using only proxy records significantly correlated with the NAO index at the 90% confidence level (35 proxy records). The fourth cluster of boxplots is the validation correlations obtained by using only proxy records significantly correlated with the NAO index at the 95% confidence level (18 proxy records).





**Figure 8.** Validation correlations obtained for different sizes of the calibration samples: from 5% to 95% of the length of the learning period ($n = 148$) with a 5% step. Red boxplots are validation correlations obtained by 100 training/testing sampling using the RF method. Blue boxplots are validation correlations obtained by 100 training/testing sampling using the Enet method. Yellow boxplots are validation correlations obtained by 100 training/testing sampling using the PLS method. Green boxplots are validation correlations obtained by 100 training/testing sampling using the PCR method. All of the reconstructions are made using the reconstruction period 1000-1970 and the 18 proxy records significantly correlated with the NAO index at the 95% confidence level over the learning period 1823-1970.

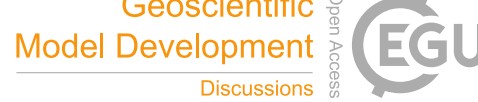



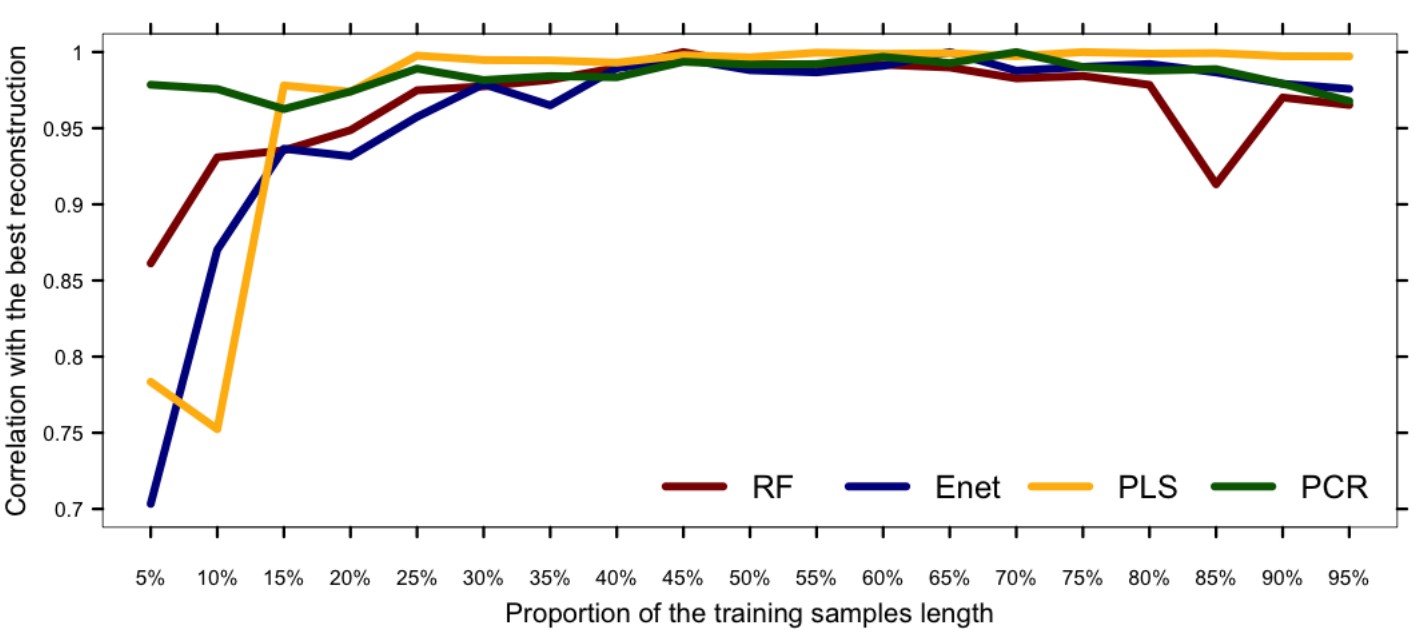

**Figure 9.** Correlations between the best reconstruction of each method given the calibration samples size and those obtained from all of the investigated calibration samples size: from $5\%$ to $95\%$ of the size of the learning period ($n = 148$) with a $5\%$ step. The best PCR proportion for the training samples length is $70\%$ of the length of th learning period ($n_{train} = 104; n_{test} = 44$), the best Enet proportion for the training sample length is $65\%$ of the training period $n_{train} = 96; n_{test} = 52$), the best PLS proportion for the training samples length is $75\%$ of the length of the learning period ($n_{train} = 111; n_{test} = 37$), while the best RF training samples size is $45\%$ of the length of the learning period ($n_{train} = 81; n_{test} = 67$). Red line gives the corresponding correlations for the RF method. Blue line gives the corresponding correlations for the Enet method. Yellow lines gives the corresponding correlations for the PLS method. Green line gives the corresponding correlations for the PCR method.





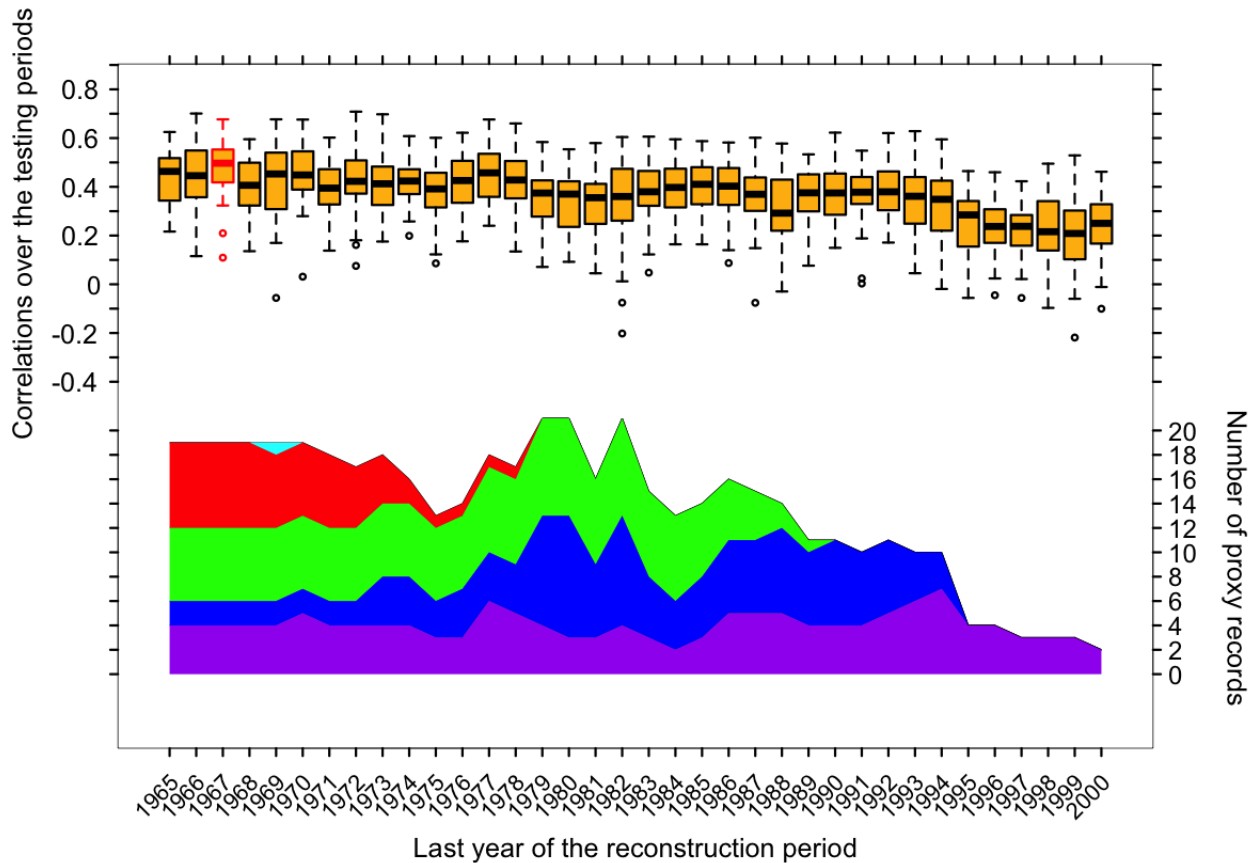

**Figure 10.** All of the reconstructions are made by $R = 50$ sampling calibration/validation, using the PLS method. The proportion of the length of the training samples is fixed to 70% and only the proxy significantly correlated with the NAO index at the 95% confidence level on the learning period are used for reconstruction. The yellow boxplots are the validation correlations obtained for each of the 36 reconstruction period: from 1000-1965 to 1000-2000 by moving the superior born by 1. Filled areas: Evolution of the proxy predictor set. For each reconstruction period, the selected proxy records are those which cover the reconstruction period and are significantly correlated with the NAO index at the 95% confidence level on corresponding the learning period. Cyan area: proxy records finishing before 1970 included. Red area: proxy records finishing after 1970 excluded and before 1980 included. Green area: proxy records finishing after 1980 excluded and before 1990 included. Blue area: proxy records finishing after 1990 excluded and before 2000 included. Purple area: proxy records finishing after 2000 excluded.





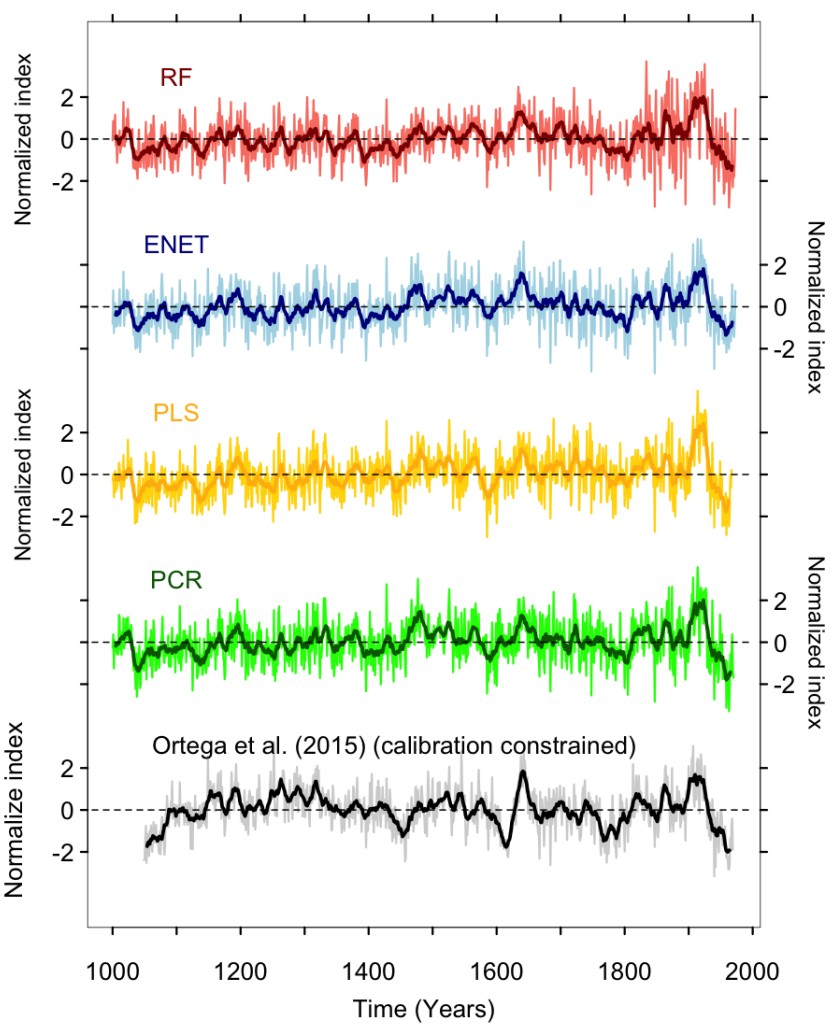

**Figure 11.** Red line: RF reconstruction on the period 1000-1973, using 18 proxy records significantly correlated at the 95% confidence level, with a proportion of the length of the training samples of 45%. Dark red line: ten years low-pass filter of the RF reconstruction. Blue line: Enet reconstruction on the period 1000-1973, using 18 proxy records significantly correlated at the 95% confidence level, with a proportion of the length of the training samples of 65%. Dark blue line: ten years low-pass filter of the Enet reconstruction. Yellow line: PLS reconstruction on the period 1000-1967, using 19 proxy records significantly correlated at the 95% confidence level, with a proportion of the length of the training samples of 70%. Dark yellow line: ten years low-pass filter of the RF reconstruction. Green line: PCR reconstruction on the period 1000-1970, using 19 proxy records significantly correlated at the 95% confidence level, with a proportion of the length of the training samples of 70%. Dark green line: ten years low-pass filter of the RF reconstruction. Grey line: Calibration constrained NAO reconstruction [Ortega et al. (2015)] on the period 1073-1969. Heavy black line: ten years low-pass filter of the calibration constrained NAO reconstruction [Ortega et al. (2015)].



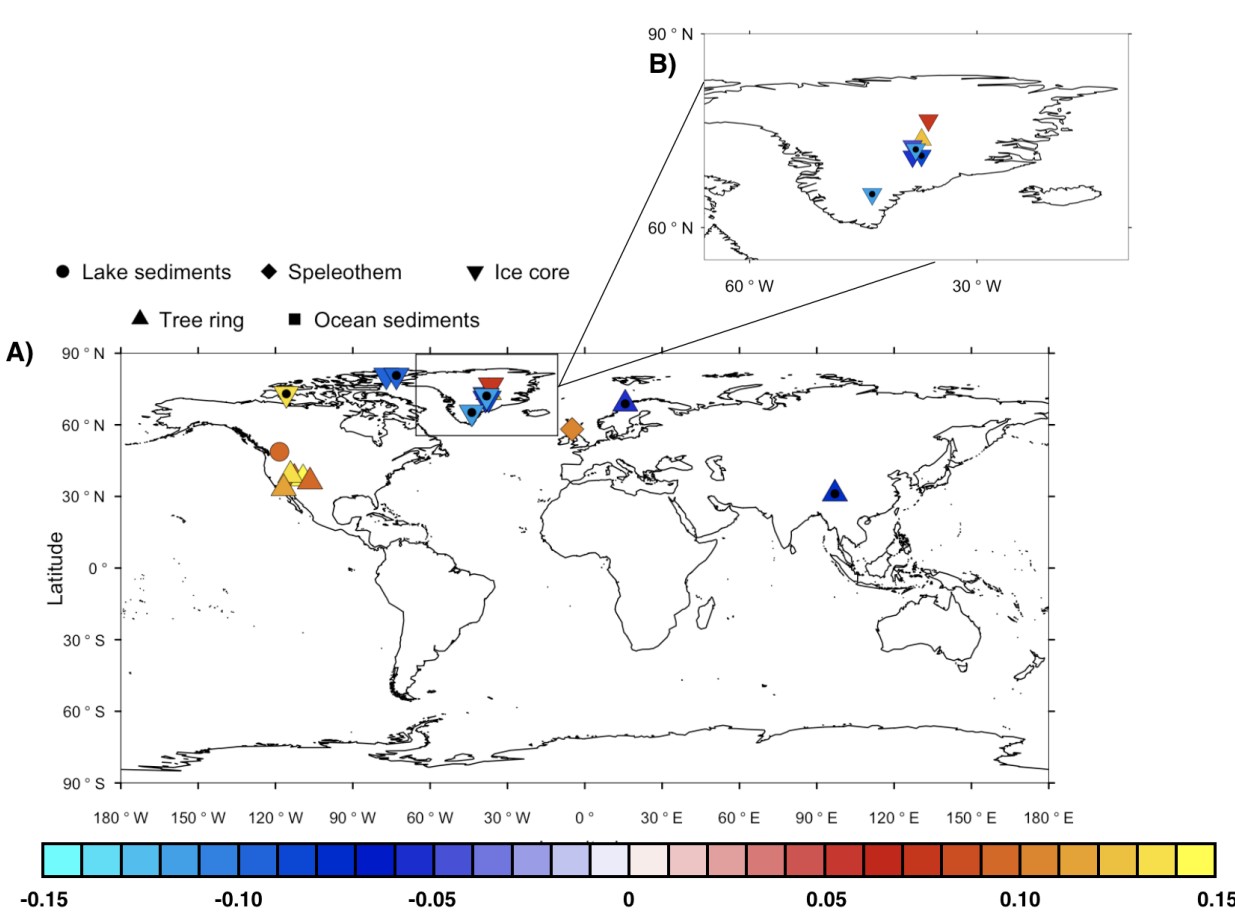

**Figure 12.** Map and weights of the 19 proxy records significantly correlated with the original NAO index on the time window 1000-1967. These weights are obtained from the PLS method and are calculated by projecting regression coefficients on the loadings (see Cook et al. (2002) and section 3.2). The shapes marked by a black circle are the proxy records used in Ortega et al. (2015).





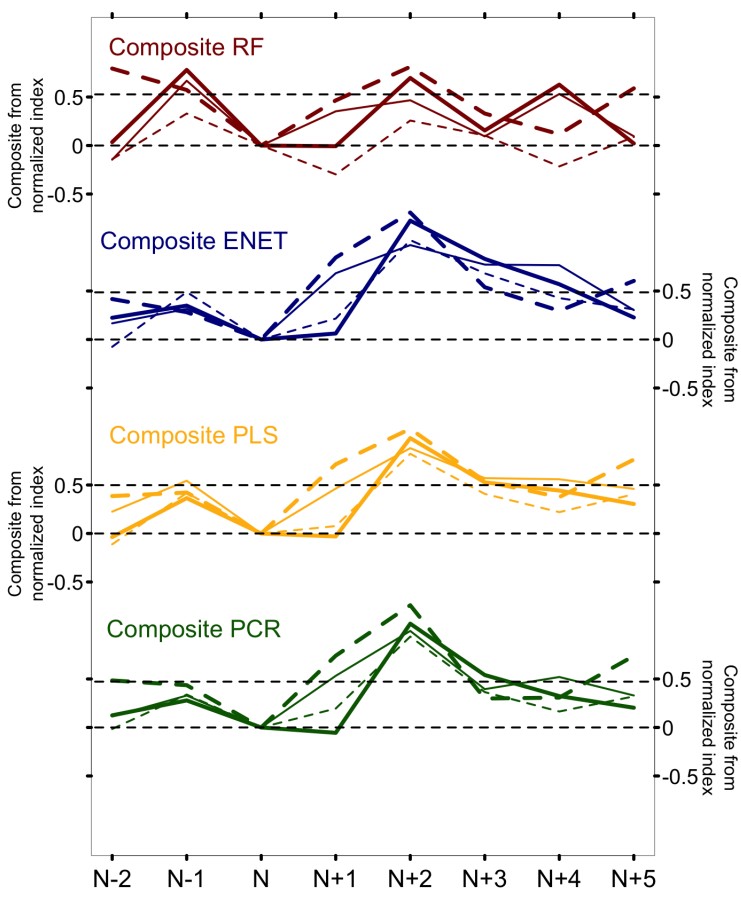

**Figure 13.** Composite of the NAO response from two years (N-2) before to five years after (N+5) ten strong volcanic eruptions considering 4 volcanic activity reconstructions. Red lines: Composites from the RF NAO reconstruction. Blue lines: Composites from the Enet NAO reconstruction. Yellow lines: Composites from the PLS NAO reconstruction. Green lines: Composites from the PCR NAO reconstructions. Light lines: composites determined using the Gao et al. (2008) volcanic activity reconstruction. Light dashed lines: composites determined using the Sigl (2014) volcanic activity reconstruction. Heavy lines: composites determined using the same volcanic activity reconstruction than Ortega et al. (2015). Heavy dashed lines: composites determined using the Crowley and Uterman (2013) volcanic activity reconstruction. All of the composites are centered to their values at the year of the volcanic eruption occurrences. For each method a 99% confidence level have been calculated by Monte-Carlo simulations using 1000 composites of eleven sampled 8 years long sub-series. The confidence born is calculated as the $90^{th}$ percentile of the 1000 differences between the $5^{th}$ and the $3^{rd}$ values of the sample composite series (i.e between N+2 and N). Black dashed lines indicate for each method the 0 level and the 90% confidence level. All of the composite series have been centered to the values at the time N.



|        | RF   | Enet | PLS  | PCR  | Ortega |
|-------:|------|------|------|------|--------|
| RF     | 1.00 | 0.88 | 0.79 | 0.83 | 0.61   |
| Enet   | 0.88 | 1.00 | 0.82 | 0.90 | 0.68   |
| PLS    | 0.79 | 0.82 | 1.00 | 0.88 | 0.52   |
| PCR    | 0.83 | 0.90 | 0.88 | 1.00 | 0.66   |
| Ortega | 0.61 | 0.68 | 0.52 | 0.66 | 1.00   |

**Table 1.** Table of correlations between five reconstructions: Ortega et al. (2015) reconstruction; RF reconstruction on the period 1000-1973 with a proportion of the length of the training samples of 45%; Enet reconstruction on the period 1000-1973 with a proportion of the length of the training samples of 70%; PLS reconstruction on the period 1000-1967 with a proportion of the length of the training samples of 75%; PCR reconstruction on the period 1000-1970 with a proportion of the length of the training samples of 70%.



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
