# Peer review of "Reconstructing climatic modes of variability from proxy records using CliMoRec ClimIndRec version 1.0"

_Geoscientific Model Development, 2018_

## Short Comment (SC1) · 29 Nov 2018

The study uses one metric to evaluate the quality of the reconstruction methods : the correlation between observed and reconstructed index over a test period. However, other properties of the reconstructed indices may also be relevant, for instance, the variance. Many regression-based reconstruction methods underestimate past variability. This can be illustrated in a simple one-dimensional set up. Considering one proxy record P that reacts to variations of the NAO index:

P (t) $= \alpha NAO(t) + \epsilon(t)$

where $\epsilon$ is random noise.

A simple, but widely used, reconstruction method is the statistical regression model:

$$N\hat{A}O(t) = \beta P(t) + \eta(t)$$

where $\eta$ represents the variability not captured by the regression model. Using Ordinary Least Squares regression to estimate $\beta$ leads to underestimation of the true value of $\beta$ and, therefore, of the true NAO variance (see for instance Isobe et al 1990 Linear regression in astronomy for a review of different regression flavours and their properties).

This problem may or not be present in the methods used in this study. It would be useful if the authors could report in Table 4 also the variance of the reconstructed NAO index in the test period wrt. to the observations and also the variance of the reconstructed index over the full period . Also, it would be informative if the time series in figure 11 were not normalized to unit variance (?),but showed the actual reconstructed variability.
* * *

---

## Short Comment (SC2) · 5 Dec 2018

Dear authors,

in my role as Executive editor of GMD, I would like to bring to your attention our Editorial version 1.1:

http://www.geosci-model-dev.net/8/3487/2015/gmd-8-3487-2015.html

This highlights some requirements of papers published in GMD, which is also available on the GMD website in the 'Manuscript Types' section:

http://www.geoscientific-model-development.net/submission/manuscript_types.html

[Figure]

In particular, please note that for your paper, the following requirements have not been met in the Discussions paper:

- "The main paper must give the model name and version number (or other unique identifier) in the title."

In order to simplify reference to your developments, please add the name of your software tool (e.g., "statistical toolbox") and its version number in the title of your article in your revised submission to GMD. The title could be something like "Reconstructing climatic modes of variability from proxy records using the statistical toolbox version 1.0: sensitivity to the methodological approach"

Yours,

Astrid Kerkweg

---

## Referee Comment (RC1) · Anonymous Referee #1 · 19 Dec 2018

Review of "Reconstructing climatic modes of variability from proxy records: sensitivity to the methodological approach"

This paper presents new reconstructions methods and applies them to reconstruct the NAO using data primarily from the PAGES2k database. I think this is a good study that introduces some potentially useful new paleoclimate reconstruction methodologies.

I have a number of comments, corrections, and requests for clarification below:

p.1 l.7-9, p.4 l.18, p.20 l.10 These statements are too strongly worded. Not every mode of variability is reconstructable, some occur on too short of time scales to be captured in the paleoclimate record (e.g., monthly time scales) and some modes are
in locations where there are poor covariances with available proxy records (e.g., the Southern Ocean).

p.2 l.9-11 This sentence is unclearly worded, for example, "non-stationary variability" doesn't "ask" questions, people ask questions.

Introduction: In general, the introduction takes a long time to get to the main points of the study. The authors might consider revising the introduction to cut down the length.

p.5 l.4-5 Linear interpolation of low resolution proxies artificially increases the influence of these records and introduces spectral artifacts in the proxy time series (e.g., Hanhijarvi, Tingley, Korhola 2013, doi: 10.1007/s00382-013-1701-4). This process also ignores dating uncertainty in such low-resolution proxies, which can be a significant source of reconstruction error. Have you accounted for these factors, particularly the dating uncertainty? What is the influence of using only annually resolved data?

Section 2.2 Do the methods estimate uncertainty in the reconstruction or just provide a single reconstruction? Are the ensembles of reconstructions discussed elsewhere a kind of uncertainty estimate of the mean reconstruction? These, or something like them, would be essential to use and display because without reliable uncertainty estimates, paleoclimate reconstructions are not useful.

p.7 l.16-19 Using correlation as the only validation metric is problematic, especially when it comes to comparing reconstruction methodologies. You really must include additional metrics that account not just for the correlation, but the variance and bias as well. If the approaches provide uncertainty estimates, then the skill metrics need to also account for those (using, for example, the continuous ranked probability score).

p.16 l.19-20 This statement is incorrect. Previous reconstructions almost never overlook this issue, but rather proxy network selection is integral to the reconstruction process. It is very rare to have a reconstruction approach, especially one that is regression-based, that does not remove proxies because of insufficient correlation with

the target climate variable.

p.18 l.1-2 Or the "significant" correlation with the NAO could be spurious. Also note that non-stationarity violates one of the fundamental assumptions of these (and nearly all) reconstruction approaches.

p.19 l.12-15 I think this statement is too strongly worded given that you've only validated the reconstructions using correlation and haven't validated reconstruction uncertainties. How do the reconstructions compare given the uncertainties?

———————————————

---

## Referee Comment (RC2) · Anonymous Referee #2 · 3 Feb 2019

**Review of**

**RECONSTRUCTING CLIMATIC MODES OF VARIABILITY FROM PROXY RECORDS: SENSITIVITY TO THE METHODOLOGICAL APPROACH**

**by Michel et al**

February 3, 2019

**Recommendation**: *Major Revisions*

**1 Scientific Comments**

I'll start with what I like about the paper: it applies several methods to the same dataset, and the results are fairly consistent among methods and with another recent reconstruction, in which one of the authors was involved (Ortegal et al, 2015). That's about it.

**1.1 This is no "big data"**

Few things are more irritating than people pretending to do "big data" when they actually don't. The authors only end up using a few dozen proxies, and only reconstruct a single index. Nothing wrong with that, but it's not "big data" by any stretch of the imagination. In fact, except for the random forest method (which is only useful in the presence of hundreds or thousands of predictors, therefore not very useful here), all of the methods described are classic forms of linear regression. Anyone is free to call that "machine learning" (since most ML methods are regression in one form or another), but the larger problem is that this is a modeling journal, and I see very little in the way of statistical modeling here.

**1.2 Suboptimal Methods**

Furthermore, the chosen methods are unable to deal with missing data, forcing the authors to limit the calibration to a set of complete records, thereby jettisoning important information. Meanwhile, at least three methods have been proposed to estimate past climates using discontinuous records:

1. The Expectation-Maximization algorithm (*Dempster et al.*, 1977) and its regularized variants (*Schneider*, 2001; *Guillot et al.*, 2015), as used by *Mann et al.* (2008) to reconstruct the global mean surface temperature, for instance.

2. Bayesian Hierarchical Models, that treat missing observations as extra parameters (*Tingley and Huybers*, 2010a,b; *Tingley et al.*, 2012; *Tingley and Huybers*, 2013; *Barboza et al.*, 2014).

3. Data assimilation approaches, for instance the Last Millennium Reanalysis framework (*Hakim et al.*, 2016; *Singh et al.*, 2018).

All of these methods have code that is publicly archived, often in open-source languages like R.

Restricting themselves to antiquated regression methods forces the authors play a dubious game of optimization on the various training and verification sets, to offset the disadvantage of restricting the network to a gap-less training set. This is suboptimal on methodological and computational grounds.

**1.3 How uncertain?**

An even more serious issue is that the authors do not provide any measure of uncertainty for their reconstructions. They could do so via any defensible method that has been applied in paleoclimate investigations, e.g. parametric or non-parametric bootstrap, jackknife, or maximum-entropy bootstrap (*Vinod and de Lacalle*, 2009).

**1.4 Statistical Models are Models too**

I feel compelled to point out that this is a journal about models, so it would be desirable to discuss the advantages of the methodological choices on modeling grounds: each of them models the data and uncertainties in various ways, and it would seem natural for such modeling assumptions and choices to be discussed here (more so than say, Climate of the Past, where the current manuscript would be a better fit in present form). One implicit modeling assumption they make is that the NAO is a linear combination of the proxy data, whereas the correct etiological relationship is the other way around (proxies react to climate, not climate to proxies). This inevitably leads to important biases (*Frost and Thompson*, 2000).

Again, some of the methods mentioned above can deal with that, and the authors should consider using them.

**1.5   Perfunctory Validation**

Another major problem is that the authors carry out a very perfunctory validation using a metric (correlation) that is known to only reward phase coherence (*Wang et al.*, 2014). At the very least, the authors should explore the Reduction of Error and Coefficient of Efficiently (*Nash and Sutcliffe*, 1970) statistics, which have been used for more than 25 years in the dendrochronological literature (*Cook et al.*, 1994). Another useful measure for point forecasts is the Continuous Ranked Probability Score (*Gneiting and Raftery*, 2007). If the authors were making interval forecasts, which they should, the sharpness of their prediction bands should be evaluated by an Interval Score (*Gneiting and Raftery*, 2007).

Finally, an obligatory measure of any statistical forecasting is to inspect the quality of residuals: since regression relies on residuals being Gaussian, independent and identically distributed, any statistics book (e.g. *Wilks*, 2011) says that the residuals should be tested for these features. This should at least be present in an Appendix.

**1.6   Double dipping**

The authors pre-screen the proxy network for correlation to the NAO index. What isn't clear is whether that is done as part over the model training, or whether this is done over the entire instrumental era (or the parts of it that overlap with each proxy series). If the latter, this is an example of "double-dipping", whereby information from the test set is used as part of training, leading to overoptimistic results. I could not ascertain this from the paper, so a clarification is necessary.

**1.7  Outdated datasets**

Why use the PAGES2k version 1, and not PAGES 2k version 2 (*PAGES 2k Consortium*, 2017)? Also, the forcing of *Gao et al.* (2008) is known to contain many errors, which have been corrected by the vastly more complete dataset of *Sigl et al.* (2014). This could explain the very weak signals observed in the paper's Superposed Epoch Analysis. I recommend using the best available data.

**2  Editorial Comments**

The manuscript reads like a literal translation of a chapter from a French PhD thesis. That means it is 1) overloaded with tedium intended to show that the main author knows what (s)he is talking about; (b) chock full of gallicisms.

**2.1  Tedious writing**

The description of methods is incredibly tedious. Sections 3.1.2, 3.2.1, 3.3.2 explain the obvious step of linear model prediction as a matrix multiplication. None of this is useful in any way as long as the code is shared. Also, an entire appendix is devoted to a user's guide, which should really be a readme file on GitHub. Please do not waste the readers' disk space and printer ink with this.

One of the most tedious parts is that the *PAGES 2k Consortium* (2013) paper is consistently referred to as "the Pages 2K database 2014 version". Since it was published in 2013, why insist on calling it 2014? Also, the consortium's name is "PAGES 2k", not Pages 2K.

In section 3.1.3, several approaches are mentioned to choose the truncation parameter

(none, it should be said, with the aid of any statistical theory), but they are not used. Either leave them unsaid, or mention them and use them (e.g. by comparing what choice is obtained with those methods vs cross-validation).

**2.2 Gallicisms**

The manuscript is generally well organized, but the writing suffers from many gallicisms. Since I happen to know a little French, here is an attempt at translating them:

- page 6, line 11: facilitate → simplify

- page 8, line 11: most performant → best-performing

- page 11, line 16: inversed → inverted

- page 12, line 23: to present frequently a → to often result in a

- page 15, line 15: require to be tuned → require tuning

**2.3 Unavailability**

I understand the need to protect data and code until the paper is published. However, acting like they are public, and linking to a non-functional Zenodo link (https://zenodo.org/record/1403146#.W4UMUGaB2qA) is bad form. Either give a complete link or mention that the data/code will be shared upon publication.

**References**

Barboza, L., B. Li, M. P. Tingley, and F. G. Viens (2014), Reconstructing past temperatures from natural proxies and estimated climate forcings using short- and long-memory models, *Ann. Appl. Stat.*, *8*(4), 1966–2001, doi:10.1214/14-AOAS785.

Cook, E. R., K. R. Briffa, and P. D. Jones (1994), Spatial regression methods in dendroclimatology: A review and comparison of two techniques, *International Journal of Climatology*, *14*, 379–402, doi:10.1002/joc.3370140404.

Dempster, A. P., N. M. Laird, and D. B. Rubin (1977), Maximum likelihood estimation from incomplete data via the EM algorithm (with discussion), *J. Roy. Stat. Soc. B.*, *39*, 1–38.

Frost, C., and S. G. Thompson (2000), Correcting for regression dilution bias: comparison of methods for a single predictor variable, *Journal of the Royal Statistical Society: Series A (Statistics in Society)*, *163*(2), 173–189, doi:10.1111/1467-985X.00164.

Gao, C., A. Robock, and C. Ammann (2008), Volcanic forcing of climate over the past 1500 years: An improved ice core-based index for climate models, *Journal of Geophysical Research (Atmospheres)*, *113*, 23,111–+, doi:10.1029/2008JD010239.

Gneiting, T., and A. E. Raftery (2007), Strictly proper scoring rules, prediction, and estimation, *Journal of the American Statistical Association*, *102*(477), 359–378, doi:10.1198/016214506000001437.

Guillot, D., B. Rajaratnam, and J. Emile-Geay (2015), Statistical paleoclimate reconstructions via Markov random fields, *Ann. Applied. Statist.*, pp. 324–352, doi:10.1214/14-AOAS794.

Hakim, G. J., J. Emile-Geay, E. J. Steig, D. Noone, D. M. Anderson, R. Tardif, N. Steiger, and W. A. Perkins (2016), The last millennium climate reanalysis project: Framework and first results, *Journal of Geophysical Research: Atmospheres*, *121*, 6745 – 6764, doi:10.1002/2016JD024751.

Mann, M. E., Z. Zhang, M. K. Hughes, R. S. Bradley, S. K. Miller, S. Rutherford, and F. Ni (2008), Proxy-based reconstructions of hemispheric and global surface temperature variations over the past two millennia, *Proceedings of the National Academy of Sciences*, *105*(36), 13,252–13,257, doi:10.1073/pnas.0805721105.

Nash, J., and J. Sutcliffe (1970), River flow forecasting through conceptual models part I – A discussion of principles, *J. Hydrol.*, *10*, 282–290, doi:10.1016/0022-1694(70)90255-6.

PAGES 2k Consortium (2013), Continental-scale temperature variability during the past two millennia, *Nature Geosci*, *6*(5), 339–346, doi:10.1038/ngeo1797.

PAGES 2k Consortium (2017), A global multiproxy database for temperature reconstructions of the Common Era, *Scientific Data*, *4*, 170,088 EP, doi:10.1038/sdata.2017.88.

Schneider, T. (2001), Analysis of Incomplete Climate Data: Estimation of Mean Values and Covariance Matrices and Imputation of Missing Values., *J. Clim.*, *14*(5), 853–871, doi:10.1175/1520-0442(2001)014<0853:AOICDE>2.0.CO;2.

Sigl, M., et al. (2014), Insights from antarctica on volcanic forcing during the common era, *Nature Clim. Change*, *4*(8), 693–697.

Singh, H. K. A., G. J. Hakim, R. Tardif, J. Emile-Geay, and D. C. Noone (2018), Insights into atlantic multidecadal variability using the last millennium reanalysis framework, *Climate of the Past*, *14*(2), 157–174, doi:10.5194/cp-14-157-2018.

Tingley, M. P., and P. Huybers (2010a), A Bayesian Algorithm for Reconstructing Climate Anomalies in Space and Time. Part 1: Development and applications to paleoclimate reconstruction problems, *J. Clim.*, *23*, 2759–2781, doi:10.1175/2009JCLI3016.1.

Tingley, M. P., and P. Huybers (2010b), A Bayesian Algorithm for Reconstructing Climate Anomalies in Space and Time. Part 2: Comparison with the Regularized Expectation-Maximization Algorithm, *J. Clim.*, *23*, 2782–2800, doi:2009JCLI3016.1.

Tingley, M. P., and P. Huybers (2013), Recent temperature extremes at high northern latitudes unprecedented in the past 600 years, *Nature*, *496*(7444), 201–205, doi:10.1038/nature11969.

Tingley, M. P., P. F. Craigmile, M. Haran, B. Li, E. Mannshardt, and B. Rajaratnam (2012), Piecing together the past: statistical insights into paleoclimatic reconstructions, *Quaternary Science Reviews*, *35*(0), 1 – 22, doi:10.1016/j.quascirev.2012.01.012.

Vinod, H. D., and J. L. de Lacalle (2009), Maximum entropy bootstrap for time series: The meboot r package, *Journal of Statistical Software*, *29*(5), 1–19.

Wang, J., J. Emile-Geay, D. Guillot, J. E. Smerdon, and B. Rajaratnam (2014), Evaluating climate field reconstruction techniques using improved emulations of real-world conditions, *Climate of the Past*, *10*(1), 1–19, doi:10.5194/cp-10-1-2014.

Wilks, D. S. (2011), *Statistical Methods in the Atmospheric Sciences: an Introduction*, 676 pp., Academic Press, San Diego.
* * *

---

## Editor Comment (EC1) · Lauren Gregoire (Editor) · 15 Feb 2019

Dear authors,

The open discussion has been closed. Based on reviewers comments, the manuscript requires major revisions. I note in particular that reviewers highlight the need for: (i) text editing to improve readability and conform to our rules on titles, (ii) a working link to the data and code, (iii) additional work and/or justification of the methods to ensure robustness of the work. To proceed with the evaluation of your manuscript and its potential publication, please address all of the reviewers' and editors' comments and requests.

Best regards, Lauren Gregoire

---

## Author Comment (AC1) · 22 Mar 2019

The study uses one metric to evaluate the quality of the reconstruction methods : the correlation between observed and reconstructed index over a test period. However, other properties of the reconstructed indices may also be relevant, for instance, the variance. Many regression-based reconstruction methods underestimate past variability. This can be illustrated in a simple one-dimensional set up. Considering one proxy record P that reacts to variations of the NAO index:

$$P(t) = \alpha NAO(t) + \varepsilon(t)$$

where $\varepsilon$ is random noise.

A simple, but widely used, reconstruction method is the statistical regression model:

$$\widehat{NAO}(t) = \beta P(t) + \eta(t)$$

where $\eta$ represents the variability not captured by the regression model. Using Ordinary Least Squares regression to estimate $\beta$ leads to underestimation of the true value of $\beta$ and, therefore, of the true NAO variance (see for instance Isobe et al 1990 Linear regression in astronomy for a review of different regression flavours and their properties).

This problem may or not be present in the methods used in this study. It would be useful if the authors could report in Table 4 also the variance of the reconstructed NAO index in the test period wrt. to the observations and also the variance of the reconstructed index over the full period.

We thank Eduardo Zorita for this constructive and useful comment. In this study we are performing multiple ensemble reconstructions, thereby bringing this result tedious to be presented efficiently for every reconstructions. We decided to add a table and a figure giving this result for the best reconstructions from each method (i.e. the reconstructions presented in figure 11). The variance of the reconstructions is presented for the whole instrumental period, the testing period, the training period, the full reconstruction period and its portion before instrumental observations of the Jones et al. (1997) NAO index (the years before 1856 being excluded). We also add discussions in the main text of the manuscript about this well-known problem in paleoclimate reconstructions.

Also, it would be informative if the time series in figure 11 were not normalized to unit variance (?), but showed the actual reconstructed variability.

Normalizing to unit variance is a useful way to easily quantify NAO variability using standard deviations as unit. Nonetheless, as Eduardo Zorita is mentioning, it is actually hiding important informations about the reconstruction we performed. Thus, we decided to modify figure 11 in order to keep the actual reconstructed variability by our code. +1 and -1 standard deviation levels for each reconstruction have also been added in this figure in addition of their regression uncertainties (see response to 1.3 comment of Anonymous reviewer 2).

---

## Author Comment (AC2) · 22 Mar 2019

Dear authors,

in my role as Executive editor of GMD, I would like to bring to your attention our Editorial version 1.1:

http://www.geosci-model-dev.net/8/3487/2015/gmd-8-3487-2015.html

This highlights some requirements of papers published in GMD, which is also available on the GMD website in the 'Manuscript Types' section:

http://www.geoscientific-model-development.net/submission/manuscript_types.html

In particular, please note that for your paper, the following requirements have not been met in the Discussions paper:

• "The main paper must give the model name and version number (or other unique identifier) in the title."

In order to simplify reference to your developments, please add the name of your software tool (e.g., "statistical toolbox") and its version number in the title of your article in your revised submission to GMD. The title could be something like "Reconstructing climatic modes of variability from proxy records using the statistical toolbox version 1.0: sensitivity to the methodological approach"

We thank the executive Editor Astrid Kerkweg for reminding us the guideline for submission. We decided to attribute the name CLIMOREC (CLIMate MOde REConstruction) to our statistical toolbox and we have changed the name of the manuscript to: "Reconstructing climatic modes of variability from proxy records using CLIMOREC version 1.0: sensitivity to the methodological approach". Also, we have modified the references "statistical toolbox" to "CLIMOREC version 1.0" in the main text of the manuscript.

---

## Author Comment (AC3) · 22 Mar 2019

This paper presents new reconstructions methods and applies them to reconstruct the NAO using data primarily from the PAGES2k database. I think this is a good study that introduces some potentially useful new paleoclimate reconstruction methodologies.

We thank the reviewer for this overall positive evaluation of our work.

I have a number of comments, corrections, and requests for clarification below:

p.1 l.7-9, p.4 l.18, p.20 l.10 These statements are too strongly worded. Not every mode of variability is reconstructable, some occur on too short of time scales to be captured in the paleoclimate record (e.g., monthly time scales) and some modes are in locations where there are poor covariances with available proxy records (e.g., the Southern Ocean).

We agree with the reviewer that this claim was too strong. This statement is modified in the corrected manuscript  to clarify that our method is not able to reconstruct every climate index but only the ones for which sufficient covariances between large-scale modes and proxy records are found and for which proxy records exhibit fine enough time resolution to resolve the main time scale of the considered variability mode. Furthermore, we will also highlight that our approach can be used to reconstruct other kind of climate variable time-series such as temperatures or precipitations for a given location.

p.2 l.9-11 This sentence is unclearly worded, for example, "non-stationary variability" doesn't "ask" questions, people ask questions.

We agree with the reviewer on this statement. We replaced "asks the questions of" by "highlights".

Introduction: In general, the introduction takes a long time to get to the main points of the study. The authors might consider revising the introduction to cut down the length.

The introduction has been largely cut down by only keeping the most important informations relative to the topic of the manuscript.

p.5 l.4-5 Linear interpolation of low resolution proxies artificially increases the influence of these records and introduces spectral artifacts in the proxy time series (e.g., Hanhijarvi, Tingley, Korhola 2013, doi: 10.1007/s00382-013-1701-4). This process also ignores dating uncertainty in such low-resolution proxies, which can be a significant source of reconstruction error. Have you accounted for these factors, particularly the dating uncertainty? What is the influence of using only annually resolved data?

Reviewer 2 also highlights this issue. He also highlights that the database of proxy records that we use (the 2014 version of the Pages 2k database plus 69 additional proxy records) has been recently updated in 2017. Following this comment we have updated our code, manuscript and data with the use of the 2017 version of the Pages 2k database. Then, using this new proxy database, and in order to address this comment, we decided to remove the proxy records that are not annually resolved. Indeed, we found that using interpolated low resolution proxy records results in overestimating their weights in our reconstruction because

of the falsely high correlations they have with the NAO index. This is largely due to their respective high auto-correlations at the annual time-scale. Hence, as mentioned by the reviewer, using this kind of proxy record indeed brings a lot of reconstruction errors due to overestimated weights, dating uncertainties, but also, because they induce erroneous validation scores as the link between these proxy records and the NAO index is overestimated.

Concerning the dating uncertainty, it is also present in annually-resolved proxy records and this aspect is not accounted for in the present version of the reconstruction toolbox. Nevertheless, this is certainly something to be considered in the next version of the code. We thus add a short discussion on this aspect in the discussion section, concerning potential outlooks for the next versions.

Section 2.2 Do the methods estimate uncertainty in the reconstruction or just provide a single reconstruction? Are the ensembles of reconstructions discussed elsewhere a kind of uncertainty estimate of the mean reconstruction? These, or something like them, would be essential to use and display because without reliable uncertainty estimates, paleoclimate reconstructions are not useful.

This was actually a major omission in the former version of the paper and we thank the reviewer to report it. The uncertainties we now provide are calculated as in Ortega et al. (2015) using the residuals calculated over the 50 training periods. These uncertainties are represented by the standard errors (s.e.) of the regression, calculated as the root of the sum of the squared residuals divided by the degree of freedom over the training periods divided by the degree of freedom:

$$s.e = \sqrt{\frac{\sum\limits_{i=i}^{n_{train}} (Y_{train} - \widehat{Y}_{train})}{n_{train} - 2}}$$

Where $n_{train}$ is the length of the training sample, $Y_{train}$ the true values of the NAO index over the training period, and $\widehat{Y}_{train}$ the fitted NAO by the regression model over the training period.

An uncertainty band 2*s.e. is calculated for each of the 50 individual reconstructions and the envelope of this 2*s.e. uncertainty bands is our estimate of the total uncertainty range of the final reconstruction.

We added regression uncertainties in a table and on the figures where the reconstructions are shown. Also, the code we deliver provide standard errors for each member of a given final reconstruction.

p.7 l.16-19 Using correlation as the only validation metric is problematic, especially when it comes to comparing reconstruction methodologies. You really must include additional metrics that account not just for the correlation, but the variance and bias as well. If the

approaches provide uncertainty estimates, then the skill metrics need to also account for those (using, for example, the continuous ranked probability score).

This comment was also highlighted by the other reviewer as well as in the short comment of Eduardo Zorita. We totally agree with this comment and we decided to add both the root mean squared errors and the Nash-Sutcliffe Coefficient of Efficiency (NSCE) as additional metrics. The NSCE calculates the ratio of the averaged quadratic distance between the reconstruction and the observations and the quadratic distance between the mean of the observations and the observations. This metric, defined between $-\infty$ and 1 indicates that the reconstruction is robust when NSCE>0. Otherwise,  lower values mean that using the mean of the testing series is more robust than performing a reconstruction using the statistical model.

We thus believe that these two metrics adequately account for the bias and variance in the reconstruction, which should then improve the conservation of these properties in our reconstruction. The whole new manuscript now accounts for these two metrics and use the NSCE as main decision metric.

p.16 l.19-20 This statement is incorrect. Previous reconstructions almost never overlook this issue, but rather proxy network selection is integral to the reconstruction process. It is very rare to have a reconstruction approach, especially one that is regression-based, that does not remove proxies because of insufficient correlation with the target climate variable.

For climate index reconstructions we found at least two major studies that have not used proxy network selection to perform their reconstruction : Cook et al 2002 (NAO reconstruction) and Wang et al 2017 (AMV reconstruction). For the latter, a table of the proxy records used is presented in supplementary information. According to this table, we found that this study has used proxy records with correlations close to 0 and non-significant between some of the proxy records and the targeted AMV index. Nevertheless, we indeed found that these studies are particular case and we modified this statement to clarify that we were referring mainly to these two studies.

p.18 l.1-2 Or the "significant" correlation with the NAO could be spurious. Also note that non-stationarity violates one of the fundamental assumptions of these (and nearly all) reconstruction approaches.

Indeed, we also ask ourselves if the significant correlations we found could be spurious but it is relatively difficult to determine whether they are or not. An indirect way to "verify" this significance of correlation is the location of the proxy records that have high correlations with the NAO. Indeed, the proxy records we use are located in the well-known center of actions of the NAO, which, in a sense, shows that the corresponding correlations are not fully spurious but may be related with well-known climatological fingerprints of the NAO (e.g. Casado et al. 2013). The second comment about non-stationarity indeed highlights a problem that not only questions our study, but also all of the proxy based reconstructions studies. We believe that this sentence was not at the right place in the submitted manuscript, since this type of caveat has to be included in the discussion section. This has been done in the revised version.

p.19 l.12-15 I think this statement is too strongly worded given that you've only validated the reconstructions using correlation and haven't validated reconstruction uncertainties. How do the reconstructions compare given the uncertainties?

As mentioned above, in the revised version we use the coefficient of efficiency to validate our reconstructions.

---

## Author Comment (AC4) · 22 Mar 2019

**1 Scientific Comments**

I'll start with what I like about the paper: it applies several methods to the same dataset, and the results are fairly consistent among methods and with another recent reconstruction, in which one of the authors was involved (Ortegal et al, 2015). That's about it.

We thank the reviewer for this positive comment. Nevertheless, as a general response to the main reviewer's criticisms below, we would like to highlight that our study is proposing novel regression methods that have, to our knowledge, not yet been applied to climate signal reconstructions. In addition, we found in previous studies cited in this manuscript (that concerns the reconstruction of climate modes, but not of climate fields), several issues in the classical methodological approaches. Our objective here is to assist paleoclimate experts in making the best out of their proxy databases with valid and robust statistical assessments. More specifically, using a new metric that we discuss below, we show how to evaluate different reconstructions of the same climate index but with different methodological choices (regression method, proxy network, length of the period on which the regression model is built). The wide range covered by the scores shows that the selection of these inputs is an important step to obtain a reconstruction as robust as possible. Furthermore, to make the production of such reconstructions more straightforward and facilitate its use to potential users, we have developed a code that simply requires a few parameters as input and that provides a reconstruction of a given climate index for a given proxy record database. In addition, the code provides an ensemble of scores that evaluates the reconstruction. By varying the different methodological choices, the user of the code can then perform several reconstructions and pick the one that has the best scores. This is why we do not submit this paper to Climate of The Past, as we would like to make climate signal reconstructions more transparent and easily accessible and verified by the community.

Furthermore, we believe this statistical toolbox could be improved in the future by including further refinements in follow up versions which constitute an additional reason for which we prefer to submit this paper to GMD. Last but not least, we believe that providing sufficient level of details concerning the mathematical rationale behind our methods is very useful, while they are hidden in the appendix in journals like Climate of the Past, which are more focused on the scientific results.

1.1 This is no "big data"

Few things are more irritating than people pretending to do "big data" when they actually don't. The authors only end up using a few dozen proxies, and only reconstruct a single index. Nothing wrong with that, but it's not "big data" by any stretch of the imagination. In fact, except for the random forest method (which is only useful in the presence of hundreds or thousands of predictors, therefore not very useful here), all of the methods described are classic forms of linear regression. Anyone is free to call that "machine learning" (since most ML methods are regression in one form or another), but the larger problem is that this is a modeling journal, and I see very little in the way of statistical modeling here.

We entirely agree that what is done in this paper is not "big data" and we didn't intend to claim we did it. The word "big data" was mentioned twice in the submitted text with the only

aim of providing a context, once in the abstract (line 6) and once in the introduction (page 4, line 8). We are actually claiming that the emergence of big data that followed the innovation in technologies and data storage has led to the development of new regression methods in the 2000's, in particular elastic net regression and Random Forest (Breiman 2001; Zou and Hastie 2005). Those methods have indeed been developed in order to address high-dimensional problems (p>n), that Principal Components Regression and Partial Least Squares poorly deal with. However, since the word "big data" can be misleading, we have decided to remove it in the revised version.

Random Forests are indeed particularly useful for high dimensional data with numerous predictors such as boosting gradients or neural networks. However, in the new version of the code, by using the Nash-Sutcliffe Coefficient of Efficiency, we have found significantly better results for the Random Forest and the Elastic-net methods than for the PLS and the PCR methods (this is illustrated in the Fig. R1 that will replace Fig.5 of the previous manuscript), which shows that adding these methods even in a low-dimension study such as in ours can be more efficient than using classical forms of linear regression. Additionally the code we provide allows to choose the network of proxy records that is used for the reconstruction. As the number of available paleoclimate data is constantly growing (even if it does not reach hundreds of thousands yet), we claim that regression methods adapted to high-dimensional problems such as Random Forests will sooner or later, become particularly useful for climate index reconstructions. We have added a few words on this subject in the discussion of the manuscript.

[Figure]

Fig. R1: Nash-Sutcliffe Coefficient Efficiency (NSCE) scores obtained for each method for the reconstruction period 1000-1970 and for different significance for the correlation test performed on the training periods: 95%, 90%, 80% and 0%. Red boxplots give the NSCE scores for the Random Forest method. Blue boxplots give the NSCE scores for the Elastic-net method. Green boxplots give the NSCE scores for the Principal Components Regression method. Yellow boxplots give the NSCE scores for Partial Least Squares method.

1.2 Suboptimal Methods

Furthermore, the chosen methods are unable to deal with missing data, forcing the authors to limit the calibration to a set of complete records, thereby jettisoning important information.

Meanwhile, at least three methods have been proposed to estimate past climates using discontinuous records:

1. The Expectation-Maximization algorithm (Dempster et al., 1977) and its regularized variants (Schneider, 2001; Guillot et al., 2015), as used by Mann et al. (2008) to reconstruct the global mean surface temperature, for instance.
2. Bayesian Hierarchical Models, that treat missing observations as extra parameters (Tingley and Huybers, 2010a,b; Tingley et al., 2012; Tingley and Huybers, 2013; Barboza et al., 2014).

3. Data assimilation approaches, for instance the Last Millennium Reanalysis framework (Hakim et al., 2016; Singh et al., 2018).

All of these methods have code that is publicly archived, often in open-source languages like R. Restricting themselves to antiquated regression methods forces the authors play a dubious game of optimization on the various training and verification sets, to offset the disadvantage of restricting the network to a gap-less training set. This is suboptimal on methodological and computational grounds.

In this study, we focus on climate variability modes, which is only a part of the global climate. We applied dedicated methods aiming at improving the reconstruction of these modes. Our techniques can certainly be further improved, but as it stands, we believe that they add new potentialities to the regression approaches currently at use.

Many paleoclimate studies (cited in the manuscript), such as our study, are focusing on reconstructions of global climate indices in time while others (e.g. the studies the reviewer mentioned here) reconstruct a particular climate variable (usually temperatures) in space and time (e.g. Climate Field Reconstruction methods). This paper is actually clarifying and adding methodological clue and gives an accessible tool to help paleoclimatologists to build more robust climate index reconstructions.

Although both approaches aim at reconstructing past climate, the question and focus of the paper is not to show if one is better than the other, but to try to further develop one of them.

Concerning data assimilation methods, we certainly agree that these are very useful methods, but we do not believe that these methods, difficult to develop within paleoclimates, necessarily discard other more simple statistical models.

We believe that science can benefit from a variety approaches, since it contributes to build robustness of its results. Therefore, we acknowledge the existence of the three methods depicted by the reviewer, and discuss them shortly in our manuscript, but we do not think there are decisive arguments showing that our approach is necessarily weaker, although this is not the scope of this paper to prove it at this stage.

1.3 How uncertain?

An even more serious issue is that the authors do not provide any measure of uncertainty for their reconstructions. They could do so via any defensible method that has been applied in paleoclimate investigations, e.g. parametric or non-parametric bootstrap, jackknife, or maximum-entropy bootstrap (Vinod and de Lacalle, 2009).

We thank the reviewer for pointing out this major omission (also mentioned by Anonymous Reviewer 1): that is the importance of assessing the reliability of our reconstruction.

The uncertainties we now provide are calculated as in Ortega et al. (2015) using the residuals calculated over the 50 training periods. These regression uncertainties are represented by the standard errors (s.e.) of the regression, calculated as the root of the sum of the squared residuals divided by the degree of freedom over the training periods divided by the degree of freedom:

$$s.e = \sqrt{\frac{\sum\limits_{i=i}^{n_{train}} (Y_{train} - \widehat{Y}_{train})}{n_{train} - 2}}$$

Where $n_{train}$ is the length of the training sample, $Y_{train}$ the true values of the NAO index over the training period, and $\widehat{Y}_{train}$ the fitted NAO by the regression model over the training period.

An uncertainty band 2*s.e. is calculated for each of the 50 individual reconstructions and the envelope of this 2*s.e. uncertainty bands is our estimate of the total uncertainty range of the final reconstruction (as a sum of the regression uncertainty plus the parameter uncertainty).

We added regression uncertainties in a table and on the figures where the reconstructions are shown. Also, the code we deliver provide standard errors for each member of a given final reconstruction.

1.4 Statistical Models are Models too

I feel compelled to point out that this is a journal about models, so it would be desirable to discuss the advantages of the methodological choices on modeling grounds: each of them models the data and uncertainties in various ways, and it would seem natural for such modeling assumptions and choices to be discussed here (more so than say, Climate of the Past, where the current manuscript would be a better fit in present form). One implicit modeling assumption they make is that the NAO is a linear combination of the proxy data, whereas the correct etiological relationship is the other way around (proxies react to climate, not climate to proxies). This inevitably leads to important biases (Frost and Thompson, 2000). Again, some of the methods mentioned above can deal with that, and the authors should consider using them.

We have explained before our motivation for submitting the paper to this journal rather than to *Climate of the Past*: the idea is to propose a statistical modelling tool, which will be available to the community and could be further developed in a transparent way, rather than to only propose a new NAO reconstruction. We have been encouraged for this by the editorial guidelines of GMD which include 'statistical models'. Nevertheless, we leave it to the editor to decide whether our study is suited for GMD or not.

Regarding the modelling assumption: stating that the NAO is a linear combination of the proxy data is something about which we have been unclear in the manuscript but this is not what we have meant literally. "NAO index can be reconstructed from a linear combination" would be a more suited sentence. We hope that the reviewer agrees with this one and we have revised the manuscript so as to avoid such shortcuts.

1.5 Perfunctory Validation

Another major problem is that the authors carry out a very perfunctory validation using a metric (correlation) that is known to only reward phase coherence (Wang et al., 2014). At the very least, the authors should explore the Reduction of Error and Coefficient of Efficiently (Nash and Sutcliffe, 1970) statistics, which have been used for more than 25 years in the

dendrochronological literature (Cook et al., 1994). Another useful measure for point forecasts is the Continuous Ranked Probability Score (Gneiting and Raftery, 2007).

We agree that the results may be sensitive to the choice of the calibration/validation metric. Thus, we have also calculated Root Mean Squared Errors as a new validation score. It gives very similar results than correlations. We thank the reviewer to suggest this more sophisticated statistics that will be added and used as the main metric in the manuscript on top of the correlations and RMSE: The Nash-Sutcliffe Coefficient of Efficiency (NSCE). The NSCE scores is indeed helping us in many ways. It shows that all the reconstruction made using the Vinther et al (2003) NAO index are not reliable since their NSCE scores are not significantly different to 0 (following student test on the scores obtained from the individual reconstructions). However, using the Jones et al (1999) index (which is exactly the same as Vinther et al (2003) index on their common period) we obtain more robust validation scores (i.e. significantly higher than 0 at 95% ).

If the authors were making interval forecasts, which they should, the sharpness of their prediction bands should be evaluated by an Interval Score (Gneiting and Raftery, 2007).

We thank the reviewer for this interesting comment. Nevertheless we should confess that what the reviewer is requesting here is not very clear to us even after carefully reading the reference mentioned. As a response, we can say that in the revised version of the manuscript, we are now properly computing uncertainties (cf. point 1.3) and notably for the validation scores, which correspond to the "forecast section" from our methodology.

Finally, an obligatory measure of any statistical forecasting is to inspect the quality of residuals: since regression relies on residuals being Gaussian, independent and identically distributed, any statistics book (e.g. Wilks, 2011) says that the residuals should be tested for these features. This should at least be present in an Appendix.

We agree that this is an important assumption to check. We have then check this assumption for the best reconstruction of each method (presented Fig. 11 of the previous manuscript) and we have added a figure showing the p-values of Shapiro-Wilk tests obtained for the 50 individual reconstructions for each of them (which have the best NSCE scores on average). Also, we have updated the code to provide the p-values of this test as an additional output.

1.6 Double dipping

The authors pre-screen the proxy network for correlation to the NAO index. What isn't clear is whether that is done as part over the model training, or whether this is done over the entire instrumental era (or the parts of it that overlap with each proxy series). If the latter, this is an example of "double-dipping", whereby information from the test set is used as part of training, leading to overoptimistic results. I could not ascertain this from the paper, so a clarification is necessary.

This comment is very useful firstly because we have been unclear on this point and secondly because it helps us to actually find out that we were doing double-dipping. Indeed, as the proxy records are selected over the entire instrumental era, the model built over the training

period uses proxy records that are, at least partially, coherent with the NAO index over the testing period, which is supposed to be independent. To correct this issue, we decided that the subselection of proxy records based on correlation test with the NAO has to be made always on training period, which means that there is no *a priori* information about the coherence between the NAO index and the selected proxy records made over the overlap period with the NAO. We have modified the code and all the results following this improvement in our approach. This does not affect much our results in the end, but is clearly an improvement in the coherence and rigor of our method, for which we thank the reviewer again.

Why use the PAGES2k version 1, and not PAGES 2k version 2 (PAGES 2k Consortium, 2017)?
Pages 2k version 2 was not available when we started this study. We thank the reviewer for highlighting the updated version, which is now used in the new version of the manuscript.

Also, the forcing of Gao et al. (2008) is known to contain many errors, which have been corrected by the vastly more complete dataset of Sigl et al. (2014). This could explain the very weak signals observed in the paper's Superposed Epoch Analysis. I recommend using the best available data.
We thank the reviewer for pointing us to the potential errors present in the Gao et al. (2008) reconstruction. Indeed, this reconstruction is now quite old, and we agree that the more recent reconstructions may have corrected some of the errors from former ones. Thus, we have removed the use of the Gao et al. (2008) reconstruction and only kept Sigl et al (2014) and Crowley et al. (2013) in the analysis of the manuscript. The inclusion of Gao et al. (2008) in the submitted manuscript was aiming to better explore potential uncertainties, but we agree with the reviewer that since Sigl et al. (2014) built on the reconstruction of Gao et al (2008) trying to improve it, this latter one has been superseded.

**2 Editorial Comments**
The manuscript reads like a literal translation of a chapter from a French PhD thesis. That means it is 1) overloaded with tedium intended to show that the main author knows what (s)he is talking about; (b) chock full of gallicisms.
We have worked hard for improving the language in this revised version and a native english colleague has agreed to review it before submission.
As mentioned by Anonymous reviewer 1, the introduction of this paper was very heavy and difficult to read, with a lot of technical details that were not always useful. The introduction of the paper has been largely reduced.

2.1 Tedious writing
The description of methods is incredibly tedious. Sections 3.1.2, 3.2.1, 3.3.2 explain the obvious step of linear model prediction as a matrix multiplication. None of this is useful in any way as long as the code is shared. Also, an entire appendix is devoted to a user's guide,

which should really be a readme file on GitHub. Please do not waste the readers' disk space and printer ink with this.

While writing the first version of the manuscript we indeed hesitated to put section 3 in the main text and not in the appendix. We believe that it may be useful to have all the necessary details in the main text, which was one of the reasons why we choose GMD. We have asked the editor about this issue and she supports our choice since it may improve clarity for people that are non-expert in statistical models. Indeed, we acknowledge that the reviewer is a great expert in statistical modelling, but our aim here is to gather a larger audience, and notably the paleoclimate record experts, who may be interested in having further details to precisely follow our methodology. Thus, we believe that this level of details is useful and this explain why we chose GMD instead of Climate of the Past.

Following the reviewer's advice, the user's guide has been removed from the appendix and is now available in a readme file on GitHub, where codes and data are also available (see section 2.3 of this response).

One of the most tedious parts is that the *PAGES 2k Consortium* (2013) paper is consistently referred to as "the Pages 2K database 2014 version". Since it was published in 2013, why insist on calling it 2014? Also, the consortium's name is "PAGES 2k", not Pages 2K.

As we have updated the database in our code, we now call it the "P2k-2017 database" in the new version of the manuscript.

In section 3.1.3, several approaches are mentioned to choose the truncation parameter (none, it should be said, with the aid of any statistical theory), but they are not used. Either leave them unsaid, or mention them and use them (e.g. by comparing what choice is obtained with those methods vs cross-validation).

We have actually tested them for the Principal Components Regression because they only are specific to this method. Results show that cross-validation gives better results but we decided not to show it in the manuscript as it was already quite dense. Nevertheless, we agree that it should be shown or mentioned. Thus, we have added a figure and a supplementary table in order to show that the use of cross validation provides better results than previous methods (only for PCR).

2.2 Gallicisms

The manuscript is generally well organized, but the writing suffers from many gallicisms. Since I happen to know a little French, here is an attempt at translating them:

• page 6, line 11: facilitate → simplify

• page 8, line 11: most performant → best-performing

• page 11, line 16: inversed → inverted

• page 12, line 23: to present frequently a → to often result in a • page 15, line 15: require to be tuned → require tuning

We thank the reviewer for these corrections that have been added in the manuscript.

2.3 Unavailability

I understand the need to protect data and code until the paper is published. How- ever, acting like they are public, and linking to a non-functional Zenodo link (https: //zenodo.org/record/1403146#.W4UMUGaB2qA) is bad form. Either give a complete link or mention that the data/code will be shared upon publication.

When we submitted the paper we tested the Zenodo link, and it worked well. We figured out, thanks to this comment, that it is now broken, and we do not know since when. We did not mean to protect our code nor our data and we actually are glad to share it as we have worked hard to build it. Codes and data can now be found on the following GitHub link: https://github.com/SimMiche/CLIMOREC

---

## Author Comment (AC5) · 28 Mar 2019

Final author response for the manuscript gmd-2018-211: "Reconstructing climatic modes of variability from proxy records: sensitivity to the methodological approach" by Michel et al.

This final author response is organised for each comment as follows:

(1) Comment from referee/public
(2) Author's response
(3) Author's change in the manuscript

**- Response to Anonymous Referee #1:**

(1) This paper presents new reconstructions methods and applies them to reconstruct the NAO using data primarily from the PAGES2k database. I think this is a good study that introduces some potentially useful new paleoclimate reconstruction methodologies.
(2) (3) We thank the reviewer for this overall positive evaluation of our work.

I have a number of comments, corrections, and requests for clarification below:

(1) p.1 l.7-9, p.4 l.18, p.20 l.10 These statements are too strongly worded. Not every mode of variability is reconstructable, some occur on too short of time scales to be captured in the paleoclimate record (e.g., monthly time scales) and some modes are in locations where there are poor covariances with available proxy records (e.g., the Southern Ocean).

(2) We agree with the reviewer that this claim was too strong.

(3) This statement is modified in the corrected manuscript to clarify that our method is not able to reconstruct every climate index but only the ones for which sufficient covariances between large-scale modes and proxy records are found and for which proxy records exhibit fine enough time resolution to resolve the main time scale of the considered variability mode. Furthermore, we will also highlight that our approach can be used to reconstruct other kind of climate variable time-series such as temperatures or precipitations for a given location.

(1) p.2 l.9-11 This sentence is unclearly worded, for example, "non-stationary variability" doesn't "ask" questions, people ask questions.

(2) We agree with the reviewer on this statement.

(3) We replaced "asks the questions of" by "highlights".

(1) Introduction: In general, the introduction takes a long time to get to the main points of the study. The authors might consider revising the introduction to cut down the length.

(2) (3) The introduction has been largely cut down by only keeping the most important informations relative to the topic of the manuscript.

(1) p.5 l.4-5 Linear interpolation of low resolution proxies artificially increases the influence of these records and introduces spectral artifacts in the proxy time series (e.g., Hanhijarvi, Tingley, Korhola 2013, doi: 10.1007/s00382-013-1701-4). This process also ignores dating uncertainty in such low-resolution proxies, which can be a significant source of reconstruction error. Have you accounted for these factors, particularly the dating uncertainty? What is the influence of using only annually resolved data?

(2) Indeed, we found that using interpolated low resolution proxy records results in overestimating their weights in our reconstruction because of the falsely high correlations they have with the NAO index. This is largely due to their respective high auto-correlations at the annual time-scale. Hence, as mentioned by the reviewer, using this kind of proxy record indeed brings a lot of reconstruction errors due to overestimated weights, dating uncertainties, but also, because they induce erroneous validation scores as the link between these proxy records and the NAO index is overestimated. Concerning the dating uncertainty, it is also present in annually-resolved proxy records and this aspect is not accounted for in the present version of the code.

(3) Following this comment we have updated our code, manuscript and data with the use of the 2017 version of the Pages 2k database as suggested by Reviewer 2. Then, using this new proxy database, and in order to address this comment, we decided to remove the proxy records that are not annually resolved. For dating uncertainties, this is certainly something to be considered in the next version of the code. We thus add a short discussion on this aspect in the discussion section, concerning potential outlooks for the next versions.

(1) Section 2.2 Do the methods estimate uncertainty in the reconstruction or just provide a single reconstruction? Are the ensembles of reconstructions discussed elsewhere a kind of uncertainty estimate of the mean reconstruction? These, or something like them, would be essential to use and display because without reliable uncertainty estimates, paleoclimate reconstructions are not useful.

(2) This was actually a major omission in the former version of the paper and we thank the reviewer to report it. The uncertainties we now provide are calculated as in Ortega et al. (2015) using the residuals calculated over the 50 training periods. These uncertainties are represented by the standard errors (s.e.) of the regression, calculated as the root of the sum of

the squared residuals divided by the degree of freedom over the training periods divided by the degree of freedom:

$$s.e = \sqrt{\frac{\sum\limits_{i=i}^{n_{train}} (Y_{train} - \widehat{Y}_{train})}{n_{train}-2}}$$

Where $n_{train}$ is the length of the training sample, $Y_{train}$ the true values of the NAO index over the training period, and $\widehat{Y}_{train}$ the fitted NAO by the regression model over the training period.

An uncertainty band 2*s.e. is calculated for each of the 50 individual reconstructions and the envelope of this 2*s.e. uncertainty bands is our estimate of the total uncertainty range of the final reconstruction.

(3) We added regression uncertainties in a table and on the figures where the reconstructions are shown. Also, the code we deliver provide standard errors for each member of a given final reconstruction.

(1) p.7 l.16-19 Using correlation as the only validation metric is problematic, especially when it comes to comparing reconstruction methodologies. You really must include additional metrics that account not just for the correlation, but the variance and bias as well. If the approaches provide uncertainty estimates, then the skill metrics need to also account for those (using, for example, the continuous ranked probability score).

(2) This comment was also highlighted by the other reviewer as well as in the short comment of Eduardo Zorita. We totally agree with this comment and we decided to add both the root mean squared errors and the Nash-Sutcliffe Coefficient of Efficiency (NSCE) as additional metrics. The NSCE calculates the ratio of the averaged quadratic distance between the reconstruction and the observations and the quadratic distance between the mean of the observations and the observations. This metric, defined between $-\infty$ and 1 indicates that the reconstruction is robust when NSCE>0. Otherwise, lower values mean that using the mean of the testing series is more robust than performing a reconstruction using the statistical model.
We thus believe that these two metrics adequately account for the bias and variance in the reconstruction, which should then improve the conservation of these properties in our reconstruction.

(3) The whole new manuscript now accounts for these two metrics and use the NSCE as the main decision metric.

(1) p.16 l.19-20 This statement is incorrect. Previous reconstructions almost never overlook this issue, but rather proxy network selection is integral to the reconstruction process. It is

very rare to have a reconstruction approach, especially one that is regression-based, that does not remove proxies because of insufficient correlation with the target climate variable.

(2) For climate index reconstructions we found at least two major studies that have not used proxy network selection to perform their reconstruction : Cook et al 2002 (NAO reconstruction) and Wang et al 2017 (AMV reconstruction).

(3) Nevertheless, we indeed found that these studies are particular cases and we modified this statement to clarify that we were referring mainly to these two studies.

(1) p.18 l.1-2 Or the "significant" correlation with the NAO could be spurious. Also note that non-stationarity violates one of the fundamental assumptions of these (and nearly all) reconstruction approaches.

(2) Indeed, we also ask ourselves if the significant correlations we found could be spurious but it is relatively difficult to determine whether they are or not. An indirect way to "verify" this significance of correlation is the location of the proxy records that have high correlations with the NAO. A way to rule out spurious correlation is the use of pseudo-proxies like in Ortega et al. (2015), but handling pseudo-proxies from different datasets was an arduous task for this multimethod paper. Nevertheless, the fact that most proxy records selected for the highest levels of correlation significance (i.e. Greenland, Arctic Canada, North America and Europe. See Fig. 6 in the last version of the manuscript) are located in the centers of action of the NAO (which has not been imposed *a priori*) (e.g. Casado et al. 2013) is a good indicator that most proxy records won't be spurious NAO predictors. The second comment about non-stationarity indeed highlights a problem that not only questions our study, but also all of the proxy based reconstructions studies.

(3) In the new version of the manuscript we remove the sentence concerning non-stationarity since this type of caveat has to be included in the discussion section. We also highlight that the location of most of the proxy records selected shows that our method seems to adequately select reliable predictors.

(1) p.19 l.12-15 I think this statement is too strongly worded given that you've only validated the reconstructions using correlation and haven't validated reconstruction uncertainties. How do the reconstructions compare given the uncertainties?

(2) (3) As mentioned above, in the revised version we use the coefficient of efficiency to validate our reconstructions and we include and discuss regression uncertainties in our main text and dedicated figures.

**- Response to Anonymous Referee #2:**

**1 Scientific Comments**

(1) I'll start with what I like about the paper: it applies several methods to the same dataset, and the results are fairly consistent among methods and with another recent reconstruction, in which one of the authors was involved (Ortegal et al, 2015). That's about it.

(2) (3) We thank the reviewer for this positive comment. Nevertheless, as a general response to the main reviewer's criticisms below, we would like to highlight that our study is proposing novel regression methods that have, to our knowledge, not yet been applied to climate signal reconstructions. In addition, we found in previous studies cited in this manuscript (that concerns the reconstruction of climate modes, but not of climate fields), several issues in the classical methodological approaches. Our objective here is to assist paleoclimate experts in making the best out of their proxy databases with valid and robust statistical assessments. More specifically, using a new metric that we discuss below, we show how to evaluate different reconstructions of the same climate index but with different methodological choices (regression method, proxy network, length of the period on which the regression model is built). The wide range covered by the scores shows that the selection of these inputs is an important step to obtain a reconstruction as robust as possible.

Furthermore, to make the production of such reconstructions more straightforward and facilitate its use to potential users, we have developed a code that simply requires a few parameters as input and that provides a set of different alternative reconstructions of a given climate index for a given proxy record database. In addition, the code provides an ensemble of scores that evaluate the different reconstructions, each produced with different methodological choices. Thus the user of CliMoRec (see "Response to the short comment from Astrid Kerkweg: 'Executive Editor comment on gmd-2018-21' ") can finally pick the one that has the best scores. This is why we do not submit this paper to Climate of The Past, as we would like to make climate signal reconstructions more transparent and easily accessible and verified by the community. Furthermore, we believe that CliMoRec could be improved in the future by including further refinements in follow up versions which constitute an additional reason for which we prefer to submit this paper to GMD. Last but not least, we believe that providing sufficient level of details concerning the mathematical rationale behind our methods is very useful,  an information that is hidden in the appendix in journals like Climate of the Past, which are more focused on the scientific results.

1.1 This is no "big data"
(1) Few things are more irritating than people pretending to do "big data" when they actually don't. The authors only end up using a few dozen proxies, and only reconstruct a single index. Nothing wrong with that, but it's not "big data" by any stretch of the imagination. In

fact, except for the random forest method (which is only useful in the presence of hundreds or thousands of predictors, therefore not very useful here), all of the methods described are classic forms of linear regression. Anyone is free to call that "machine learning" (since most ML methods are regression in one form or another), but the larger problem is that this is a modeling journal, and I see very little in the way of statistical modeling here.

(2) (3) We entirely agree that what is done in this paper is not "big data" and we didn't intend to claim we did it. The word "big data" was mentioned twice in the submitted text with the only aim of providing a context, once in the abstract (line 6) and once in the introduction (page 4, line 8). We are actually claiming that the emergence of big data that followed the innovation in technologies and data storage has led to the development of new regression methods in the 2000's, in particular elastic net regression and Random Forest (Breiman 2001; Zou and Hastie 2005). Those methods have indeed been developed in order to address high-dimensional problems (p>n), that Principal Components Regression and Partial Least Squares poorly deal with. However, since the word "big data" can be misleading, we have decided to remove it in the revised version. Random Forests are indeed particularly useful for high dimensional data with numerous predictors such as boosting gradients or neural networks. However, in the new version of the code, by using the Nash-Sutcliffe Coefficient of Efficiency, we have found significantly better results for the Random Forest and the Elastic-net methods than for the PLS and the PCR methods (this is illustrated in the Fig. R1 that will replace Fig.7 of the previous manuscript), which shows that adding these methods even in a low-dimension study such as in ours can be more efficient than using classical forms of linear regression. Additionally the code we provide allows to choose the network of proxy records that is used for the reconstruction. As the number of available paleoclimate data is constantly growing (even if it does not reach hundreds of thousands yet), we claim that regression methods adapted to high-dimensional problems such as Random Forests will sooner or later, become particularly useful for climate index reconstructions. We have added a few words on this subject in the discussion of the manuscript.

[Figure]

Fig. R1: Nash-Sutcliffe Coefficient Efficiency (NSCE) scores obtained for each method for the reconstruction period 1000-1970 and for different significance for the correlation test performed on the training periods: 95%, 90%, 80% and 0%. Red boxplots give the NSCE scores for the Random Forest method. Blue boxplots give the NSCE scores for the Elastic-net method. Green boxplots give the NSCE scores for the Principal Components Regression method. Yellow boxplots give the NSCE scores for Partial Least Squares method.

1.2 Suboptimal Methods

(1) Furthermore, the chosen methods are unable to deal with missing data, forcing the authors to limit the calibration to a set of complete records, thereby jettisoning important information.

Meanwhile, at least three methods have been proposed to estimate past climates using discontinuous records:

1. The Expectation-Maximization algorithm (Dempster et al., 1977) and its regularized variants (Schneider, 2001; Guillot et al., 2015), as used by Mann et al. (2008) to reconstruct the global mean surface temperature, for instance.
2. Bayesian Hierarchical Models, that treat missing observations as extra parameters (Tingley and Huybers, 2010a,b; Tingley et al., 2012; Tingley and Huybers, 2013; Barboza et al., 2014).

3. Data assimilation approaches, for instance the Last Millennium Reanalysis framework (Hakim et al., 2016; Singh et al., 2018).

All of these methods have code that is publicly archived, often in open-source languages like R. Restricting themselves to antiquated regression methods forces the authors play a dubious game of optimization on the various training and verification sets, to offset the disadvantage of restricting the network to a gap-less training set. This is suboptimal on methodological and computational grounds.

(2) In this study, we focus on climate variability modes, which is only a part of the global climate. We applied dedicated methods aiming at improving the reconstruction of these modes. Our techniques can certainly be further improved, but as it stands, we believe that they add new potentialities to the regression approaches currently at use. This paper is actually clarifying and adding methodological clue and gives an accessible tool to help paleoclimatologists to build more robust climate index reconstructions. Although our approach and the approaches mentioned by the reviewer aim at reconstructing past climate, the question and focus of the paper is not to show if one is better than the other, but to try to further develop one of them. Concerning data assimilation methods, we certainly agree that these are very useful methods, but we do not believe that these methods, difficult to implement and thus not accessible to all paleoclimatologists, necessarily discard other more simple statistical models. We believe that science can benefit from a variety of approaches, all together contributing to identify robust results.

(3) Therefore, we acknowledge the existence of the three methods depicted by the reviewer, and discuss them shortly in our manuscript, but we do not think there are decisive arguments showing that our approach is necessarily weaker, although this is not the scope of this paper to prove it at this stage.

1.3 How uncertain?
(1) An even more serious issue is that the authors do not provide any measure of uncertainty for their reconstructions. They could do so via any defensible method that has been applied in paleoclimate investigations, e.g. parametric or non-parametric bootstrap, jackknife, or maximum-entropy bootstrap (Vinod and de Lacalle, 2009).

(2) We thank the reviewer for pointing out this major omission (also mentioned by Anonymous Reviewer 1): that is the importance of assessing the reliability of our reconstruction. The uncertainties we now provide are calculated as in Ortega et al. (2015) using the residuals calculated over the 50 training periods. These regression uncertainties are represented by the standard errors (s.e.) of the regression, calculated as the root of the sum of the squared residuals over the training periods divided by the degree of freedom:

$$s.e = \sqrt{\frac{\sum\limits_{i=i}^{n_{train}} (Y_{train} - \widehat{Y}_{train})}{n_{train} - 2}}$$

Where $n_{train}$ is the length of the training sample, $Y_{train}$ the true values of the NAO index over the training period, and $\widehat{Y}_{train}$ the fitted NAO by the regression model over the training period.

An uncertainty band 2*s.e. is calculated for each of the 50 individual reconstructions and the envelope of this 2*s.e. uncertainty bands is our estimate of the total uncertainty range of the final reconstruction (as a sum of the regression uncertainty plus the parameter uncertainty).

(3) We added regression uncertainties in a table and on the figures where the reconstructions are shown (Fig. 11 of the last version of the manuscript). Also, the code we deliver provide standard errors for each member of a given final reconstruction.

1.4 Statistical Models are Models too

(1) I feel compelled to point out that this is a journal about models, so it would be desirable to discuss the advantages of the methodological choices on modeling grounds: each of them models the data and uncertainties in various ways, and it would seem natural for such modeling assumptions and choices to be discussed here (more so than say, Climate of the Past, where the current manuscript would be a better fit in present form). One implicit modeling assumption they make is that the NAO is a linear combination of the proxy data, whereas the correct etiological relationship is the other way around (proxies react to climate, not climate to proxies). This inevitably leads to important biases (Frost and Thompson, 2000). Again, some of the methods mentioned above can deal with that, and the authors should consider using them.

(2) We have explained before our motivation for submitting the paper to this journal rather than to *Climate of the Past*: the idea is to propose a statistical modelling tool, which will be available to the community and could be further developed in a transparent way, rather than to only propose a new NAO reconstruction. We have been encouraged for this by the editorial guidelines of GMD which include 'statistical models'. Nevertheless, we leave it to the editor to decide whether our study is suited for GMD or not. Regarding the modelling assumption: stating that the NAO is a linear combination of the proxy data is something about which we have been unclear in the manuscript but this is not what we have meant literally. "NAO index can be reconstructed from a linear combination" would be a more suited sentence.

(3) We have revised the manuscript so as to avoid such shortcuts following proposition described above.

1.5 Perfunctory Validation

(1) Another major problem is that the authors carry out a very perfunctory validation using a metric (correlation) that is known to only reward phase coherence (Wang et al., 2014). At the very least, the authors should explore the Reduction of Error and Coefficient of Efficiently (Nash and Sutcliffe, 1970) statistics, which have been used for more than 25 years in the dendrochronological literature (Cook et al., 1994). Another useful measure for point forecasts is the Continuous Ranked Probability Score (Gneiting and Raftery, 2007).

(2) (3) We agree that the results may be sensitive to the choice of the calibration/validation metric. Thus, we have also calculated Root Mean Squared Errors as a new validation score. We thank the reviewer to suggest this more sophisticated metrics that have been added and used as the main metric in the manuscript on top of the correlations and RMSE: The Nash-Sutcliffe Coefficient of Efficiency (NSCE). The NSCE scores is indeed helping us in many ways. It shows that all the reconstruction made using the Vinther et al (2003) NAO index are not reliable since their NSCE scores are not significantly different to 0 (following student test on the scores obtained from the individual reconstructions). However, using the Jones et al (1999) index (which is exactly the same as Vinther et al (2003) index on their common period) we obtain more robust validation scores (i.e. significantly higher than 0 at 95% ).

(1) If the authors were making interval forecasts, which they should, the sharpness of their prediction bands should be evaluated by an Interval Score (Gneiting and Raftery, 2007).

(2) (3) We thank the reviewer for this interesting comment. Nevertheless we should confess that what the reviewer is requesting here is not very clear to us even after carefully reading the reference mentioned. As a response, we can say that in the revised version of the manuscript, we are now properly computing uncertainties (cf. point 1.3) and notably for the validation scores, which correspond to the "forecast section" from our methodology.

(1) Finally, an obligatory measure of any statistical forecasting is to inspect the quality of residuals: since regression relies on residuals being Gaussian, independent and identically distributed, any statistics book (e.g. Wilks, 2011) says that the residuals should be tested for these features. This should at least be present in an Appendix.

(2) We agree that this is an important assumption to check.

(3) We have then check this assumption for the best reconstruction of each method (presented Fig. 11 of the previous manuscript) and we have added a figure showing the p-values of Shapiro-Wilk tests obtained for the 50 individual reconstructions for each of them (which

have the best NSCE scores on average). Also, we have updated the code to provide the p-values of this test as an additional output.

1.6 Double dipping

(1) The authors pre-screen the proxy network for correlation to the NAO index. What isn't clear is whether that is done as part over the model training, or whether this is done over the entire instrumental era (or the parts of it that overlap with each proxy series). If the latter, this is an example of "double-dipping", whereby information from the test set is used as part of training, leading to overoptimistic results. I could not ascertain this from the paper, so a clarification is necessary.

(2) This comment is very useful firstly because we have been unclear on this point and secondly because it helps us to actually find out that we were doing double-dipping. Indeed, as the proxy records are selected over the entire instrumental era, the model built over the training period uses proxy records that are, at least partially, coherent with the NAO index over the testing period, which is supposed to be independent.

(3) To correct this issue, we decided that the subselection of proxy records based on correlation test with the NAO has to be made always on training period, which means that there is no *a priori* information about the coherence between the NAO index and the selected proxy records made over the overlap period with the NAO. We have modified the code and all the results following this improvement in our approach. This does not affect much our results in the end, but is clearly an improvement in the coherence and rigor of our method, for which we thank the reviewer again.

(1) Why use the PAGES2k version 1, and not PAGES 2k version 2 (PAGES 2k Consortium, 2017)?

(2) Pages 2k version 2 was not available when we started this study.

(3) We thank the reviewer for highlighting the updated version, which is now used in the new version of the manuscript.

(1) Also, the forcing of Gao et al. (2008) is known to contain many errors, which have been corrected by the vastly more complete dataset of Sigl et al. (2014). This could explain the very weak signals observed in the paper's Superposed Epoch Analysis. I recommend using the best available data.

(2) We thank the reviewer for pointing us to the potential errors present in the Gao et al. (2008) reconstruction. Indeed, this reconstruction is now quite old, and we agree that the more recent reconstructions may have corrected some of the errors from former ones.

(3) Thus, we have removed the use of the Gao et al. (2008) reconstruction and only kept Sigl et al (2014) and Crowley et al. (2013) in the analysis of the manuscript. The inclusion of Gao et al. (2008) in the submitted manuscript was aiming to better explore potential uncertainties, but we agree with the reviewer that since Sigl et al. (2014) built on the reconstruction of Gao et al (2008) trying to improve it, this latter one has been superseded.

**2 Editorial Comments**
(1) The manuscript reads like a literal translation of a chapter from a French PhD thesis. That means it is 1) overloaded with tedium intended to show that the main author knows what (s)he is talking about; (b) chock full of gallicisms.

(2) We have worked hard for improving the language in this revised version and a native english colleague has agreed to review it before submission.

(3) As mentioned by Anonymous reviewer 1, the introduction of this paper was very heavy and difficult to read, with a lot of technical details that were not always useful. The introduction of the paper has been largely reduced.

2.1 Tedious writing
(1) The description of methods is incredibly tedious. Sections 3.1.2, 3.2.1, 3.3.2 explain the obvious step of linear model prediction as a matrix multiplication. None of this is useful in any way as long as the code is shared. Also, an entire appendix is devoted to a user's guide, which should really be a readme file on GitHub. Please do not waste the readers' disk space and printer ink with this.

(2) While writing the first version of the manuscript we indeed hesitated to put section 3 in the main text and not in the appendix. We believe that it may be useful to have all the necessary details in the main text, which was one of the reasons why we choose GMD. We have asked the editor about this issue and she supports our choice since it may improve clarity for people that are non-expert in statistical models. Indeed, we acknowledge that the reviewer is a great expert in statistical modelling, but our aim here is to gather a larger audience, and notably the paleoclimate record experts, who may be interested in having further details to precisely follow our methodology. Thus, we believe that this level of details is useful and this explain why we chose GMD instead of Climate of the Past.

(3) Following the reviewer's advice, the user's guide has been removed from the appendix and is now available in a readme file on GitHub, where codes and data are also available (see section 2.3 of this response).

1) One of the most tedious parts is that the *PAGES 2k Consortium* (2013) paper is consistently referred to as "the Pages 2K database 2014 version". Since it was published in 2013, why insist on calling it 2014? Also, the consortium's name is "PAGES 2k", not Pages 2K.

(2) (3) As we have updated the database in our code, we now call it the "P2k-2017 database" in the new version of the manuscript.

(1) In section 3.1.3, several approaches are mentioned to choose the truncation parameter (none, it should be said, with the aid of any statistical theory), but they are not used. Either leave them unsaid, or mention them and use them (e.g. by comparing what choice is obtained with those methods vs cross-validation).

(2) We have actually tested them for the Principal Components Regression because they only are specific to this method. Results show that cross-validation gives better results but we decided not to show it in the manuscript as it was already quite dense. Nevertheless, we agree that it should be shown or mentioned.

(3) Thus, we have added a supplementary figure and a supplementary table in order to show that the use of cross validation provides better results than previous methods (only for PCR).

2.2 Gallicisms
(1) The manuscript is generally well organized, but the writing suffers from many gallicisms. Since I happen to know a little French, here is an attempt at translating them:
• page 6, line 11: facilitate → simplify
• page 8, line 11: most performant → best-performing
• page 11, line 16: inversed → inverted
• page 12, line 23: to present frequently a → to often result in a • page 15, line 15: require to be tuned → require tuning

(2) (3) We thank the reviewer for these corrections that have been added in the manuscript.

2.3 Unavailability
(1) I understand the need to protect data and code until the paper is published. However, acting like they are public, and linking to a non-functional Zenodo link (https:

//zenodo.org/record/1403146#.W4UMUGaB2qA) is bad form. Either give a complete link or mention that the data/code will be shared upon publication.

(2) (3) When we submitted the paper we tested the Zenodo link (as advised by GMD), and it worked well. We figured out, thanks to this comment, that it is now broken, and we do not know since when neither why. We did not mean to protect our code nor our data and we actually are glad to share it as we have worked hard to build it. Codes and data can now be found on the following GitHub link: https://github.com/SimMiche/CliMoRec

**- Response to the short comment from Eduardo Zorita: "Reconstruction Variance?"**

(1) The study uses one metric to evaluate the quality of the reconstruction methods : the correlation between observed and reconstructed index over a test period. However, other properties of the reconstructed indices may also be relevant, for instance, the variance. Many regression-based reconstruction methods underestimate past variability. This can be illustrated in a simple one-dimensional set up. Considering one proxy record P that reacts to variations of the NAO index:

$P(t) = \alpha NAO(t) + \varepsilon(t)$

where $\varepsilon$ is random noise.

A simple, but widely used, reconstruction method is the statistical regression model:

$N\hat{A}O(t) = \beta P(t) + \eta(t)$

where $\eta$ represents the variability not captured by the regression model. Using Ordinary Least Squares regression to estimate $\beta$ leads to underestimation of the true value of $\beta$ and, therefore, of the true NAO variance (see for instance Isobe et al 1990 Linear regression in astronomy for a review of different regression flavours and their properties).

This problem may or not be present in the methods used in this study. It would be useful if the authors could report in Table 4 also the variance of the reconstructed NAO index in the test period wrt. to the observations and also the variance of the reconstructed index over the full period.

(2) We thank Eduardo Zorita for this constructive and useful comment.

(3) We decided to add a table showing the variance for the best reconstructions from each method (i.e. the reconstructions presented in figure 11). The variance of the reconstructions is presented for the whole instrumental period, the testing period, the training period, the full reconstruction period and its portion before instrumental observations of the Jones et al. (1997) NAO index (the years before 1856 being excluded). We also add discussions in the main text of the manuscript about this well-known problem in paleoclimate reconstructions.

(1) Also, it would be informative if the time series in figure 11 were not normalized to unit variance (?), but showed the actual reconstructed variability.

(2) Normalizing to unit variance is a useful way to easily quantify NAO variability using standard deviations as unit. Nonetheless, as Eduardo Zorita is mentioning, it is actually hiding important informations about the reconstruction we performed.

(3) Thus, we decided to modify figure 11 in order to keep the actual reconstructed variability by our code. +1 and -1 standard deviation levels for each reconstruction have also been added in this figure in addition of their regression uncertainties (see response to 1.3 comment of "Response to Anonymous Referee #2").

- **'Response to the short comment from Astrid Kerkweg: "Executive Editor comment on gmd-2018-211"'**

(1) Dear authors,
in my role as Executive editor of GMD, I would like to bring to your attention our Editorial version 1.1:
http://www.geosci-model-dev.net/8/3487/2015/gmd-8-3487-2015.html

This highlights some requirements of papers published in GMD, which is also available on the GMD website in the 'Manuscript Types' section:
http://www.geoscientific-model-development.net/submission/manuscript_types.html

In particular, please note that for your paper, the following requirements have not been met in the Discussions paper:
• "The main paper must give the model name and version number (or other unique identifier) in the title."

In order to simplify reference to your developments, please add the name of your software tool (e.g., "statistical toolbox") and its version number in the title of your article in your revised submission to GMD. The title could be something like "Reconstructing climatic modes of variability from proxy records using the statistical toolbox version 1.0: sensitivity to the methodological approach"

(2) We thank the executive Editor Astrid Kerkweg for reminding us the guideline for submission.

(3) We decided to attribute the name CliMoRec (Climate Mode Reconstruction) to our statistical toolbox and we have changed the name of the manuscript to: "Reconstructing

climatic modes of variability from proxy records using CliMoRec version 1.0: sensitivity to the methodological approach". Also, we have modified the references "statistical toolbox" to "CliMoRec version 1.0" in the main text of the manuscript.

---

## Referee Report (RR1)

**Review of**
**RECONSTRUCTING CLIMATIC MODES OF VARIABILITY FROM PROXY RECORDS USING CLIMOREC VERSION 1.0**
**by Michel et al**

June 3, 2019

**Recommendation**: *Minor Revisions*

**Summary**: *This is a revised version of a previous manuscript. The authors have addressed some, but not all, of the point I had raised, and some serious methodological points remain. The suggested revisions are minor in comparison to the first round, but I strongly urge the authors to either implement them or substantiate their position with a convincing rebuttal.*

**1    Scientific Comments**

**1.1    Paleo Mumbo Jumbo**

A common feature of paleoclimate statistics is the introduction of methodological twists that have little or no theoretical justification. Witness the tortured Principal Component regression method of *Mann et al.* (1998), which has created endless backlash from statisticians for little gain. Other examples abound. The lesson is that, unless there is a clear theoretical or heuristic justification for modifying a tried and true method, one had best stick to the tried and true method.

While the paper carefully describes the classic regression flavors or learning algorithms used here (PCR, elastic net, Random Forest, PLS), it also wraps them into an extremely unconventional form of bootstrapping (subselecting parts of the training period in an unspecified way) and averaging over this sample to obtain their "best" reconstructions (Section 2.2, point #6). I know of no justification for doing this, and it seems highly redundant with the cross-validation approach. I thought I had pointed this out in my original review, but I cannot find it there, as it got overshadowed by other considerations. It is now time to address this serious issue.

My recommendations are:

- Provide a theoretical justification

- Demonstrate using simulated data that this is a sensible (I suspect this won't work, but I'm open to surprises).

- Clearly explain the rationale in the text.

- Make sure users can easily turn off this feature, in case they want to stick to tried and true methods.

I cannot recommend the publication of this toolbox unless these conditions are met.

**1.2 Statistical Models are Models too**

I must reiterate the point that GMD is a journal about models, so it would be desirable to discuss the advantages of the methodological choices on modeling grounds: each of the regression methods models the data and uncertainties in various ways, and it would seem natural for such modeling assumptions and choices to be discussed here. One implicit modeling assumption they make is that the NAO is a linear combination of the proxy data, whereas the correct etiological relationship is the other way around (proxies react to climate, not climate to proxies). This inevitably leads to important biases (*Frost and Thompson*, 2000), as pointed out by Eduardo Zorita in his comment. Since the paper describes a toolbox, it is important that users be made aware of these caveats.

I also second Eduardo Zorita's suggestion of including metrics of variance in the validation, as this is easier to do for indices than fields. This and other diagnostics (e.g. R, CE) should be included on Fig 10, for instance, or in a Table.

**1.3 Perfunctory Validation**

Validation has improved in this version with the addition of more metrics, and an analysis of residuals, which will be very reassuring to informed readers. A more fundamental issue is the sampling: the authors currently do not specify how they perform cross-validation, with substantial implications for the estimation of generalization error.

Put simply, cross-validation is a way to estimate generalization error, that is, the error that one would make by estimating values of the target (here, the NAO index) that lie outside the range of the training interval (*Arlot and Celisse*, 2010). That is ostensibly the goal of reconstructing pre-instrumental values of a climate index, so one wants a way to estimate this generalization error using instrumental values. Cross-validation does this by selectively removing a subset of the training interval, and using it to compute validation metrics. If one does this in a sensible manner over suitable permutations, one can show that CV provides a good estimate of the actual generalization error. Here is the rub: a lot depends on the sampling mechanism. There are basically two choices: removing points at random, or removing blocks of consecutive points. The first looks like Venetian blinds in the data matrix, so it is sometimes called "blinds-style cross-validation"; the other is called "block-style cross-validation", for obvious reasons. This makes little difference when the data are independent; but in climate timeseries, autocorrelation is almost always incredibly large, so that one gets skill from persistence alone. Thus, if you remove the year 1911 from your training sample but have both 1910 and 1912, you can produce a skillful estimate of the NAO index in 1911 without having any proxies at all! You will get enough information from past and future values of the index to produce a reasonable estimate of the index at that withheld point. In climate timeseries it is essential that cross-validation be done in the block style. Is this what was done here?

Another issue is the value of $K$: large ones lead to more variable estimates, whereas small ones lead to more bias. In the case of the block-style cross-validation that needs to be applied here, $K = 5$ is usually found a good compromise, but obviously the code can be made flexible enough to adjust this. The other limit is leave-one-out cross-validation, where $K = n$. Evaluating the sensitivity of parameter tuning to this choice would be important. Note that this would fall under section 4.1.2, as under block-style validation, the length of the validation period depends on $K$.

**1.4 Regression Methods vs Inferential Framework**

I commend the authors for including a discussion of methodologies to deal with missing data (Section 5.1). One aspect that does not come out as clearly as it should is that the inference framework (e.g. the Expectation-Maximization algorithm, or a Bayesian hierarchical model) is distinct from the modeling choices, a point made eloquently by *Tingley et al.* (2012). Thus, all the methods used herein could be used in the framework of RegEM for instance (they would have to be embedded within), or in a Bayesian framework.

**1.5 Modes vs indices**

I understand that the paper's motivation is the reconstruction of climate modes, particularly the NAO. However, as the authors seem fond of overwhelming readers with superfluous details, it is fair to provide a detail-oriented review. Strictly speaking, CliMoRec enables the reconstruction of indices, not "modes". The authors should explain that it would also work on ANY timeseries, including hemispheric averages (e.g. Northern Hemisphere Temperature). Accordingly, they might also consider rebranding it: ClimIndRec, perhaps?

**1.6 Forcing attribution**

The authors used the common method of Superposed Epoch Analysis to evaluate the response of the NAO to volcanic forcing, but do not quantify uncertainties. Obviously, not all of the wiggles are meaningful, and some methods exist to tell which ones are (e.g. Rao et al 2019, 10.1016/j.dendro.2019.05.001). As it stands, the authors mention significance, but I could not get details on how that was established. Without proper uncertainty quantification, one cannot rule out the possibility that none of the wiggles stand out of the noise.

**1.7 Feature selection**

The conclusion states: *We have shown that for Enet, PLS and particularly PCR which is frequently used in paleclimatology, selecting proxy records with a strong correlation with the index to be reconstructed over the training periods is a good way to improve the NSCE scores, and hence it allows more reliable reconstructions (section 4.1.1). Contrarily, RF gives more reliable reconstructions using the whole set of records (section 4.1.1).* This is entirely unsurprising for PCR and PLS, as they do are not designed to achieve *feature selection* (*Friednman, Hastie & Tibshirani*, 2008, chap 18). However, that is one of the purposes of RF, so it is entirely expected that it would not need additional screening prior to application. I'm a little more perplexed that Enet benefits from screening, as its L1 penalty encourages zero coefficients that effectively turn off features (here, proxies) that don't help prediction. I suspect things might be different if the LASSO is used first for feature selection, and then ridge regression applied to minimize prediction error (with the correct parameter choice), as opposed to apply both at once. The extreme variance suppression of the Enet estimate in Fig 10 suggests that the parameter choice is not optimal, in this case at least.

**1.8 Uncertainties**

Buried in the supplement is the definition of how uncertainties are calculated with CliMoRec; it turns out to be the standard error of residuals, a perfectly reasonably choice when the number of predictors stays constant over time, but an otherwise suboptimal one. Indeed, as proxy density decreases back in time, so does the information available, and therefore error bars should widen back

in time. While it is true that this point remains depressingly under-appreciated in the paleoclimate community, some methods can deal with this, like BARCAST (see *Tingley and Huybers*, 2013, , Fig. 1) and LMR (*Hakim et al.*, 2016). In the regression context, this can be taken care of with frozen network analysis, or bootstrapping, as done in *PAGES 2k Consortium* (2017) (see their Figs. 7 and 8). At the very least, the authors should flag that this choice neglects changes in proxy availability over time, highlighting potential improvements for future versions of the toolbox.

**2 Editorial Comments**

Gallicisms have almost entirely disappeared; nice job! One remains: "contrarily" should be replaced by "on the contrary".

As pointed out before: *The description of methods is incredibly tedious. Sections 3.1.2, 3.2.1, 3.3.2 explain the obvious step of linear model prediction as a matrix multiplication. None of this is useful in any way as long as the code is shared.*

I maintain that the mathematical details of these regression methods is of limited use: people with a statistical background already know them, and people without a statistical background are unlikely to read them. This section should be moved to an appendix, so that the few readers who really need the details can find them, but it doesn't clutter the narrative. What *would* be interesting is to discuss the *modeling* assumptions underlying these methods (as requested above), but that is not what is done here.

Progress has also been made in that the authors are now using the up-to-date version of the PAGES 2k database. Yet, they insist on calling it Pages 2K. Not a deal-breaker, but it would be nice to use the correct spelling (PAGES 2k).

Re: code, the GitHub link works, but Zenodo registration is still a good idea to encourage code citation.

**References**

Arlot, S., and A. Celisse (2010), A survey of cross-validation procedures for model selection, *Statistics Surveys*, 4, 40–79, doi:10.1214/09-SS054.

Hastie, T., R. Tibshirani, and J. Friedman (2008), The elements of statistical learning: data mining, inference and prediction, 2 ed., *Springer Verlag*.

Frost, C., and S. G. Thompson (2000), Correcting for regression dilution bias: comparison of methods for a single predictor variable, *Journal of the Royal Statistical Society: Series A (Statistics in Society)*, *163*(2), 173–189, doi:10.1111/1467-985X.00164.

Hakim, G. J., J. Emile-Geay, E. J. Steig, D. Noone, D. M. Anderson, R. Tardif, N. Steiger, and W. A. Perkins (2016), The last millennium climate reanalysis project: Framework and first results, *Journal of Geophysical Research: Atmospheres*, *121*, 6745 – 6764, doi:10.1002/2016JD024751.

Mann, M. E., R. S. Bradley, and M. K. Hughes (1998), Global-scale temperature patterns and climate forcing over the past six centuries, *Nature*, 392, 779–787, doi:10.1038/33859.

PAGES 2k Consortium (2017), A global multiproxy database for temperature reconstructions of the Common Era, *Scientific Data*, *4*, 170,088 EP, doi:10.1038/sdata.2017.88.

Tingley, M. P., and P. Huybers (2010a), A Bayesian Algorithm for Reconstructing Climate Anomalies in Space and Time. Part 1: Development and applications to paleoclimate reconstruction problems, *J. Clim.*, *23*, 2759–2781, doi:10.1175/2009JCLI3016.1.

Tingley, M. P., and P. Huybers (2010b), A Bayesian Algorithm for Reconstructing Climate Anomalies in Space and Time. Part 2: Comparison with the Regularized Expectation-Maximization Algorithm, *J. Clim.*, *23*, 2782–2800, doi:2009JCLI3016.1.

Tingley, M. P., and P. Huybers (2013), Recent temperature extremes at high northern latitudes unprecedented in the past 600 years, *Nature*, *496*(7444), 201–205, doi:10.1038/nature11969.

Tingley, M. P., P. F. Craigmile, M. Haran, B. Li, E. Mannshardt, and B. Rajaratnam (2012), Piecing together the past: statistical insights into paleoclimatic reconstructions, *Quaternary Science Reviews*, *35*(0), 1 – 22, doi:10.1016/j.quascirev.2012.01.012.

---

## Referee Report (RR2)

The authors provide an overview of four statistical methods together with code for the development of proxy reconstructions. I agree with the first reviewer that the paper (and code) provide a valuable contribution to the literature, particularly by making the statistical tools easily accessible and comparing their performance. However, I also share the second reviewer's concerns about the content, presentation, and format of the paper. In particular, the authors need to highlight the value-added of their manuscript compared to existing work already documenting the implementation of existing software packages.

**Main Comments**

1. Calling R-code a 'computer device' seems strange. Why not release a formal R-package on CRAN? Or alternatively, call it 'software' or just 'R-code'.

2. Please clarify more carefully what the value-added of the paper is. There already exist R-packages (which are used within CliMoRec) that run PCR, Lasso, Elastic Net etc. What additional benefits does CliMoRec provide over the existing packages and their standard implementations? This would be important to highlight for potential users who have to choose between just using e.g. the 'glmnet' package for Elastic Net, vs. CliMoRec.

3. Please provide a bit more detail on important tuning parameters. For example, the section on choosing the penalty terms lambda and alpha (in Elastic Net) is very brief and it is not clear for practitioners whether the particular form of cross-validation employed is generally applicable.

4. More generally, please clarify why the particular model selection procedures are chosen? Why PCR, Elastic Net and Random Forests? There are many other alternatives (see e.g. adaptive Lasso, general-to-specific selection, etc.). The manuscript would benefit from being placed in the wider context of other methods being available as alternatives.

5. CliMoRec appears to use R-code based on existing packages implementing PCR, Lasso, Elastic Net, etc. rather than developing these functions itself. These existing packages have to be cited and credited in the methods sections (such as 'glmnet' in R for elastic net). Not citing the software packages is poor practise and particularly important for a paper that discusses software. Most packages used in CliMoRec have a corresponding *Journal of Statistical Software* (JSS) paper that should be cited.

   For example, 'glmnet' used by the authors (in the first line of code of CliMoRec on Github) is documented in Friedman, Hastie, and Tibshirani (2010) https://www.jstatsoft.org/article/view/v033i01 and should be cited as such.

   If there is no JSS paper available, then the R-packages should be cited through CRAN.

6. On Lasso and Elastic Net: Lasso is not a consistent model selection method with oracle properties, instead, the authors may want to refer readers to the Adaptive Lasso and Elastic Net. See e.g. Zou, H. (2006). The adaptive lasso and its oracle properties. *Journal of the American Statistical Association*, 101(476), 1418-1429., or Zou, H., & Zhang, H. H. (2009). On the adaptive elastic-net with a diverging number of parameters. Annals of statistics, 37(4), 1733.

**Minor Comments**

1. P14. Line 1: "most simple" replace with "simplest"
2. Section 2.3: the title "Mathematical Formalism" seems strange and not entirely clear.
3. P 13, the sentence "For Enet method" is missing a word, maybe "For *the* Enet method"?

---

## Referee Report (RR3)

[referee-annotated manuscript omitted]

---

## Author Response (AR2)

**Response to review comment 1**

This author response is organised for each comment as follows:

- (1) Comment from referee/public
- (2) Author's response
- (3) Author's change in the manuscript and code

Summary: This is a revised version of a previous manuscript. The authors have addressed some, but not all, of the point I had raised, and some serious methodological points remain. The suggested revisions are minor in comparison to the first round, but I strongly urge the authors to either implement them or substantiate their position with a convincing rebuttal.

**1** Scientific Comments**

**1.1 Paleo Mumbo Jumbo**

A common feature of paleoclimate statistics is the introduction of methodological twists that have little or no theoretical justification. Witness the tortured Principal Component regression method of Mann et al. (1998), which has created endless backlash from statisticians for little gain. Other examples abound. The lesson is that, unless there is a clear theoretical or heuristic justification for modifying a tried and true method, one had best stick to the tried and true method. While the paper carefully describes the classic regression flavors or learning algorithms used here (PCR, elastic net, Random Forest, PLS), it also wraps them into an extremely unconventional form of bootstrapping (subselecting parts of the training period in an unspecified way) and averaging over this sample to obtain their "best" reconstructions (Section 2.2, point #6). I know of no justification for doing this, and it seems highly redundant with the cross-validation approach. I thought I had pointed this out in my original review, but I cannot find it there, as it got overshadowed by other considerations. It is now time to address this serious issue.

We thank the reviewer for highlighting this problem as we wish to develop a useful tool for paleoclimatologists while being verified by the statistical community. We decided to use the methodology proposed by Ortega et al (2015) and extend it to other statistical methods by varying different methodological parameters such as the learning period, proxy selection or the way train/test sampling is carried out (Section 4 of the submitted paper). The blind trust in the methodology of Ortega et al (2015) led us to this methodological approximation. Indeed, following this commentary of the reviewer, we have searched the bibliography for an explanation of this unconventional bootstrap approach and have found nothing convincing. Nevertheless, the methodology used to calculate scores and the cross-validation approach only applied to sample training for hyperparameter optimization is well known and frequently used in statistical learning.

Following the reviewer's suggestions we decided to keep an almost identical algorithm regardless of the method used in order to calculate the scores of a reconstruction in the same way as in the previous version. For the final reconstruction, rather than aggregating the individual reconstructions as before, the method is applied to the entire learning dataset with here also a cross-validation to optimize hyperparameters.

My recommendations are:

• Provide a theoretical justification

We have not found any theoretical justification and therefore decide to stick to tried and true methods.

• Demonstrate using simulated data that this is a sensible (I suspect this won't work, but I'm open to surprises).

• Clearly explain the rationale in the text.

• Make sure users can easily turn off this feature, in case they want to stick to tried and true methods.

We have now changed the way the final reconstruction given by the toolbox in the code is calculated and this is specified in the main text of the manuscript. There will therefore be no option for the users, who will directly obtain the result obtained by the tried and true methods (once their hyperparameters have been optimised using cross-validation and scores have been calculated using training/testing sampling). In addition of the scores calculated over testing samples, it now provides statistics (correlation, RMSE, CE) calculated for the final model that uses the whole years of observations for X and Y.

I cannot recommend the publication of this toolbox unless these conditions are met.

**1.2 Statistical Models are Models too**

I must reiterate the point that GMD is a journal about models, so it would be desirable to discuss the advantages of the methodological choices on modeling grounds: each of the regression methods models the data and uncertainties in various ways, and it would seem natural for such modeling assumptions and choices to be discussed here. One implicit modeling assumption they make is that the NAO is a linear combination of the proxy data, whereas the correct etiological relationship is the other way around (proxies react to climate, not climate to proxies). This inevitably leads to important biases (Frost and Thompson, 2000), as pointed out by Eduardo Zorita in his comment. Since the paper describes a toolbox, it is important that users be made aware of these caveats.

We understood from Eduardo Zorita's comment and Frost and Thompson's (2000) work that the bias is rather due to the uncertainties associated with predictors, in this case proxies. The uncertainties associated to biases due to biological/geological signals, other climatic influences, seasonal effects, etc... It means that the climate variable associated with the proxy is biased and thus the statistical link between the proxy and the climate index is underestimated, leading to biases in the reconstruction. Following the reviewer's comment, we decided to add a paragraph of discussions in section 2 dedicated to the limits of our approach and therefore of the tool we propose.

We therefore address this fundamental problem by specifying in the main text that since climate variations affect variations in proxies, we can then attempt to estimate past climate variations using the statistical methods proposed. We also discuss the problem that proxies uncertainties due to measurement and transfer methods lead to an underestimation of the link between climate variables translated by proxies and climate variations.

I also second Eduardo Zorita's suggestion of including metrics of variance in the validation, as this is easier to do for indices than fields. This and other diagnostics (e.g. R, CE) should be included on Fig 10, for instance, or in a Table.

In the second version of the manuscript we already added Tab. 4 in supplementaries. This table addresses Eduardo Zorita's comment as it presents the variance of reconstructions for different periods

or groups of periods: Training periods, testing periods, reconstruction period and learning period. To address this comment.

These statistics R, CE and RMSE are now included in Fig 10 in the next version of the manuscript.

**1.3 Perfunctory Validation**

Validation has improved in this version with the addition of more metrics, and an analysis of residuals, which will be very reassuring to informed readers. A more fundamental issue is the sampling: the authors currently do not specify how they perform cross-validation, with substantial implications for the estimation of generalization error. Put simply, cross-validation is a way to estimate generalization error, that is, the error that one would make by estimating values of the target (here, the NAO index) that lie outside the range of the training interval (Arlot and Celisse, 2010). That is ostensibly the goal of reconstructing pre-instrumental values of a climate index, so one wants a way to estimate this generalization error using instrumental values. Cross-validation does this by selectively removing a subset of the training interval, and using it to compute validation metrics. If one does this in a sensible manner over suitable permutations, one can show that CV provides a good estimate of the actual generalization error. Here is the rub: a lot depends on the sampling mechanism. There are basically two choices: removing points at random, or removing blocks of consecutive points. The first looks like Venetian blinds in the data matrix, so it is sometimes called "blinds-style cross-validation"; the other is called "block-style cross-validation", for obvious reasons. This makes little difference when the data are independent; but in climate timeseries, autocorrelation is almost always incredibly large, so that one gets skill from persistence alone. Thus, if you remove the year 1911 from your training sample but have both 1910 and 1912, you can produce a skillful estimate of the NAO index in 1911 without having any proxies at all! You will get enough information from past and future values of the index to produce a reasonable estimate of the index at that withheld point. In climate timeseries it is essential that cross-validation be done in the block style. Is this what was done here?

We currently use purely random sampling (i.e. the blinds-style CV) cross-validation as well as train/test sampling. We would like to specify that it is by using the "hold out" approach (e.g. train/test sampling) that we can calculate the generalization error while cross-validation is used to optimize the hyperparameters of the associated regression method. The hold-out approach differs slightly from cross-validation in that no different blocks are built, each of which is set aside at each iteration. This involves setting aside part of the sample (test sample) to finally estimate the quality of the statistical model.

However, we understand the reviewer's comment and have decided to now use the block-style approach rather than blinds-style approach for both the hold-out sampling and the K-fold cross-validation sampling and we now use the reviewer's argument in the main text..

We emphasize that the block-style approach results in a finite number of samples regardless of the size of the train sample chosen. This therefore leads to an estimate of the generalization error or optimal hyperparameters no longer dependent on sampling and are therefore unique for a given K. Thus, if, for example, one year of instrumental measurement is incorrect (say twenty years), a block-style approach would suffer much more than a blind-style approach to bias since these data will pollute the calculation of scores and the calibration of data on a permanent basis. The blind-style approach would not completely eliminate bias, of course, but by randomly distributing potentially poorly measured data across the different samples, it can reduce bias. Block-style vs. Blind-style approaches is now discussed in the paper but only the block-style one is used for the study. It should be stressed that if a block-style splits is performed for hold-out, the number of training splits, R, is now determined by the size of the testing (or training) samples relative to the size of the whole learning sample. For ClimIndRec users, if a block style hold-out is performed, R input is ignored and the real value of R is determined. Following the block-style splitting and in order to produce the maximum splits as possible, the first testing period encompasses the first  $n_{test}$  time steps. The second testing period is then the shifted by one time step version of the first testing period. And so on until each data of the learning period has been used at least once. These informations are specified and explained in the new version of the manuscript.

Important note for the reviewer: We understand what is embarrassing for the reviewer as our approach seems to him to apply a double validation. We insist that what we call "hold-out" (Sammut et al. 2009, *Encyclopedia of Machine Learning, p.507*) is the validation while KFCV is the method we use to tune parameters. We did not find a way to combine both and we were seriousely working on for this round of review. If this can help for understanding, in Ortega et al. (2015), Nat. Geosci., they use the Preisendörfer's rule N (Presiendörfer, 1988) to tune the number of Principal Components used over their calibration (here training) samples. As this method is only applicable for PCR, we chose the KFCV method to tune parameters as it can be used for any regression method. Hence it is very important to see KFCV not as a validation procedure but as a tuning parameter method such as the Preisendörfer's rule N is for PCR. As we use ClimIndRec for other studies, we are actually searching for an approach to overcome the double sampling we actually do. Hence, if the approach we use in this version of the manuscript appears still irritating for the reviewer we are fully open to discuss and find a compromise with the reviewer about this, around another round of review if needed for instance. We emphasize that we do not think we are doing wrong but we know that this might not be an optimal approach.

Another issue is the value of K: large ones lead to more variable estimates, whereas small ones lead to more bias. In the case of the block-style cross-validation that needs to be applied here, K = 5 is usually found a good compromise, but obviously the code can be made flexible enough to adjust this. The other limit is leave-one-out cross-validation, where K = n. Evaluating the sensitivity of parameter tuning to this choice would be important. Note that this would fall under section 4.1.2, as under block-style validation, the length of the validation period depends on K.

This interesting comment highlighted to us that we chose K in a completely arbitrary way. In the first version of the code and paper implemented, we had a cross-validation leave-one-out. The problem we had encountered was the execution time since with R=100 and K=n doing so as  $100^{n}$  models were built for each reconstruction studied in the paper when we had to respect the deadline of the journal. This is why in the current version we have switched to a 10-fold CV method that is less expensive in terms of computing time. We performed for each reconstruction method 3 reconstructions over the reconstruction period 1000-1970 with different values for K=5,10 and n. NSCE scores shown Fig S1 indicates that the choice of K is not affecting scores. In addition reconstructions for a same regression method but for different choices of K never have a correlation lesser than 0.

Figure S1: NSCE scores obtained by reconstructing the NAO over the period 1000-1970 with each regression method for different values for K.

Hence, in view of the reviewer's comment, we have decided to study by default the case K=5 which is now discussed in the paper. In the next version of ClimIndRec, the choice of K can be determined by the user. It should be noted that if the user chooses a block-style approach for train/test sampling, then R is ignored, and  $n_{train}$  is used to determine the samples.

**1.4 Regression Methods vs Inferential Framework**

I commend the authors for including a discussion of methodologies to deal with missing data (Section 5.1). One aspect that does not come out as clearly as it should is that the inference framework (e.g. the Expectation-Maximization algorithm, or a Bayesian hierarchical model) is distinct from the modeling choices, a point made eloquently by Tingley et al. (2012). Thus, all the methods used herein could be used in the framework of RegEM for instance (they would have to be embedded within), or in a Bayesian framework.

In view of the large number of changes we had to make in the last round of reviews, we failed to highlight this important problem with our toolbox. We do not exclude the possibility that in the future, this type of Bayesian approach could be implemented in the tool, which would lead to a major improvement in the exhaustiveness of the use of the proxies database. Unfortunately, we think, because we are currently limited in our theoretical and technical knowledge of this type of approach, we will not have time to look at this aspect for this round of review.

We will add a discussion largely based on these limitations of the toolbox in the paragraph that discuss the rationale of ClimIndRec, its limitations and its added-value to the classical R packages (see section 2 of response to reviewer 3) (i.e. section 2.1 in the new version of the manuscript).

**1.5 Modes vs indices**

I understand that the paper's motivation is the reconstruction of climate modes, particularly the NAO. However, as the authors seem fond of overwhelming readers with superfluous details, it is fair to provide a detail-oriented review. Strictly speaking, CliMoRec enables the reconstruction of indices, not "modes". The authors should explain that it would also work on ANY timeseries, including hemispheric averages (e.g. Northern Hemisphere Temperature). Accordingly, they might also consider rebranding it: ClimIndRec, perhaps?

Very good catch. We already highlighted that the toolbox can reconstruct any climate timeseries so that we indeed choose an inappropriate name for it.

The reviewer's suggestion being very relevant, we decided to change the name of the toolbox by ClimIndRec which is more appropriated.

**1.6 Forcing attribution**

The authors used the common method of Superposed Epoch Analysis to evaluate the response of the NAO to volcanic forcing, but do not quantify uncertainties. Obviously, not all of the wiggles are meaningful, and some methods exist to tell which ones are (e.g. Rao et al 2019, 10.1016/j.dendro.2019.05.001). As it stands, the authors mention significance, but I could not get details on how that was established. Without proper uncertainty quantification, one cannot rule out the possibility that none of the wiggles stand out of the noise.

We put the method we use to calculate this significance in the supplement. Actually, we use a very similar approach than Rao et al. 2019 (for volcanic response but not fire response in their paper) which is the method used by Ortega et al. (2015), a Monte-Carlo approach. We first randomly select 1000 sets of 11 "fake" volcanic eruptions and each is centered to 0 for the year N (year of the eruption). For each, a superposed epoch analysis of the 11 fake eruptions is performed. We then retain the 90% level of the N+1 response among the 1000 11-length composite of fake eruptions as the significance level. In Rao et al. (2019), their approach is more sophisticated as the significance level is calculated in the same way but for each time lag of eruption, which we are not doing here.

According to this interesting comment we decided to use and develop in the supplement the Rao et al. (2019) approach as suggested by the reviewer.

**1.7 Feature selection**

The conclusion states: We have shown that for Enet, PLS and particularly PCR which is frequently used in paleoclimatology, selecting proxy records with a strong correlation with the index to be reconstructed over the training periods is a good way to improve the NSCE scores, and hence it allows more reliable reconstructions (section 4.1.1). Contrarily, RF gives more reliable reconstructions using the whole set of records (section 4.1.1). This is entirely unsurprising for PCR and PLS, as they do are not designed to achieve feature selection (Friednman, Hastie & Tibshirani, 2008, chap 18). However, that is one of the purposes of RF, so it is entirely expected that it would not need additional screening prior to application. I'm a little more perplexed that Enet benefits from screening, as its L1 penalty encourages zero coefficients that effectively turn off features (here, proxies) that don't help prediction.

I suspect things might be different if the LASSO is used first for feature selection, and then ridge regression applied to minimize prediction error (with the correct parameter choice), as opposed to apply both at once. The extreme variance suppression of the Enet estimate in Fig 10 suggests that the parameter choice is not optimal, in this case at least.

Reviewer 3 also highlighted that he has some doubts about our choice of Elastic Net. He suggested to apply an Adaptive Lasso approach (Zou and Zhang, 2009, see comment 6 of reviewer 3). This response is then likely similar to the one we provided too reviewer 3 (see response to comment 1.7 of reviewer 3).

If Lasso+Ridge had provided better results than the methods presented in the former version of the manuscript, we would have certainly added it, and modified the figures accordingly, but given the short time available for resubmission, and the negative results, we decided not to. Fig S2 and S3 presented below show the results obtained for Fig 6 an Fig 7, but where PCR and PLS (outperformed by Enet and RF in this case) CE scores are respectively replaced by those obtained for Lasso+Ridge and adaptive Lasso CE scores.

---

## Author Response (AR3)

1 Scientific Comments

For a study leveraging statistical methods, the manuscript betrays great naïveté about statistics. For instance, the authors use "hyperparameters" (which are specific to Bayesian inference) to mean "control parameters". This is misleading, and needs to be corrected least users be confused into thinking that the methods applied in this toolbox are Bayesian.

We had read in different publications the use of "hyperparameter" to mention control parameters from machine learning technics:
- Duan et al., 2003, https://doi.org/10.1016/S0925-2312(02)00601-X
- Bernard et al., 2009, https://doi.org/10.1007/978-3-642-02326-2_18
- Probst et al., 2019, https://doi.org/10.1002/widm.1301

And there are others.

Although "control parameter" seems to be more accurate as it is less confusing with "hyperparameter" from Bayesian inference, we don't think that its use in the last version of the manuscript is a misleading. We nevertheless changed the term into "control parameter" in the new version of the manuscript.

They also mention that "updates of ClimIndRec will be dedicated to propose other regression methods such as lasso regression", apparently ignorant of the fact that it is a special case of the Elastic Net regression they are using.

In the supplementaries of the last version manuscript, we included the mathematical details of regression methods used and clearly explained that Elastic Net is a combination of ridge and lasso regression. We also explain that if alpha=1 is equivalent to a ridge regression and alpha=0 is equivalent to a lasso regression. However, as Elastic-Net is actually used in ClimIndRec, it optimises both lambda and alpha. While implementing lasso regression in ClimIndRec might consist in only optimising lambda while alpha is set to 0. The sentence cited by the reviewer is actually a mistake from us. But we mentioned that lasso is a special case of Elastic Net in both the supplementaries and in the main text of the two previous versions of the manuscript. So we think that calling us "ignorants" and using this argument to say that readers might be warned that we are, is maybe a little bit unfair. We hope you understand we are a bit upset.

I am not sure that much can be done about the problem at this stage, but I do think it is a warning to all users that those who designed this toolbox are not on top of their statistical game.

We are sorry to hear that as both reviewer's arguments mentioned above do not seem really fair for us.

2 Editorial Comments

The writing is still rather tedious. Please see annotated PDF for suggestions.

We thank very much the reviewer for all these useful suggestions that we added in the new version of the manuscript.

Also note that we had a "bug" in the code when producing Shapiro-Wilks tests for Fig 11 that were not convincing at all. These bugs have been corrected in the code provided on Zenodo and Fig 11 has been updated.

It is the third time that I have to point out that "Pages 2K" should be "PAGES 2k" and I am getting really, really tired of it. For heaven's sake, please show respect for this working group and correct the mistake!

We are really sorry for this mistake we changed it everywhere in the main text but we forgot to modify our .bib file so that you find (another time) this mistake when we cited PAGES groups' papers.